# Surface ozone at Nam Co in the inland Tibetan Plateau: variation, synthesis comparison and regional representativeness

Xiufeng Yin [1, 2, 3], Shichang Kang [1, 3, 4], Benjamin de Foy [5], Zhiyuan Cong [2, 4], Jiali Luo [6], Lang Zhang [2], Yaoming Ma [2, 3, 4], Guoshuai Zhang [2], Dipesh Rupakheti [2, 3], Qianggong Zhang [2, 4]

[1]State Key Laboratory of Cryospheric Sciences, Northwest Institute of Eco-Environment and Resources, Chinese Academy of Sciences, Lanzhou, 730000, China

[2]Key Laboratory of Tibetan Environment Changes and Land Surface Processes, Institute of Tibetan Plateau Research, Chinese Academy of Sciences, Beijing, 100101, China

[3]University of Chinese Academy of Sciences, Beijing, 100039, China

[4]CAS Center for Excellence in Tibetan Plateau Earth Sciences, Beijing, 100085, China

[5]Department of Earth and Atmospheric Sciences, Saint Louis University, St. Louis, MO, 63108, USA

[6]Key Laboratory of Semi-Arid Climate Change, Ministry of Education, Lanzhou, 730000, China

*Correspondence to:* Qianggong Zhang (qianggong.zhang@itpcas.ac.cn) and Shichang Kang (shichang.kang@lzb.ac.cn)

**Abstract:**

Ozone is an important pollutant and greenhouse gas, and tropospheric ozone variations are generally associated with both natural and anthropogenic processes. As one of the most pristine and inaccessible regions in the world, the Tibetan Plateau has been considered as an ideal region for studying processes of the background atmosphere. Due to the vast area of the Tibetan Plateau, sites in the southern, northern and central regions exhibit different patterns of variation in surface ozone. Here, we present continuous measurements of surface ozone mixing ratios at Nam Co Station over a period of ~5 years (January 2011 to October 2015), which is a background site in the inland Tibetan Plateau. An average surface ozone mixing ratio of 47.6±11.6 ppb (mean ± standard deviation) was recorded, and a large annual cycle was observed with maximum ozone mixing ratios in the spring and minimum ratios during the winter. The diurnal cycle is characterized by a minimum in the early morning and a maximum in the late afternoon. Nam Co Station represents a background region where surface ozone receives negligible local anthropogenic emissions inputs, and the anthropogenic contribution from South Asia in spring and China in summer may affect Nam Co Station occasionally. Surface ozone at Nam Co Station is mainly dominated by natural processes involving photochemical reactions, vertical mixing and downward transport of stratospheric air mass. Model results indicate that the study site is affected differently by the surrounding areas in different seasons: air masses from the southern Tibetan Plateau contribute to the high ozone levels in the spring and enhanced ozone levels in the summer are associated with air masses from the northern Tibetan Plateau. By comparing measurements at Nam Co Station with those from other sites in the Tibetan Plateau, we aim to expand the understanding of ozone cycles and transport processes over the Tibetan Plateau. This work may provide a reference for future model simulations.

## 1 Introduction

The concentration of ozone in the troposphere showed sustained growth during the 20th century due to the increased emissions of anthropogenic precursors (Cooper et al., 2014). High levels of surface ozone are currently a major environmental concern because of the harm ozone poses to health and vegetation at the surface (LRTAP, 2015; REVIHAAP, 2013; US EPA, 2013; Mauzerall and Wang, 2001; Desqueyroux et al., 2002). In addition, ozone is a major precursor of hydroxyl (OH) and hydroperoxy ($HO_2$) radicals and it controls the oxidation capacity of the atmosphere (Brasseur et al., 1999). Furthermore, as the third most important greenhouse gas (after carbon dioxide ($CO_2$) and methane ($CH_4$)), tropospheric ozone contributes to global warming and has an estimated global average radiative forcing of $0.40 \pm 0.20$ W m$^{-2}$ at a high confidence level (Myhre et al., 2013). Average monthly and annual concentrations of tropospheric ozone are used to assess and improve global modeling results (Wild and Prather, 2006; Roelofs et al., 2003).

The origin of tropospheric ozone and its temporal variation vary from site to site. Historically, the stratosphere was initially thought to be the main source of surface (tropospheric) ozone, and a network of surface ozone monitoring sites was proposed (Junge, 1962). In the 1970s and 1980s, the effect of photochemical reactions in the troposphere on surface ozone became well recognized (Chameides and Walker, 1973; Crutzen, 1974), and photochemistry was identified as the dominant source of tropospheric ozone, as supported by models (Wu et al., 2007). Background sites can represent areas with surface ozone concentrations that are under the control of synoptic systems and are minimally affected by local anthropogenic sources. The study of surface ozone at background sites may enrich the understanding of surface ozone variation patterns.

The Tibetan Plateau (27°N–45°N, 70°E–105°E, average elevation ~ 4 km) is the highest and most extensive highland in the world and has been called the 'Third Pole' (Yao et al., 2012). Due to its small human population and low level of industrialization, the Tibetan Plateau is an ideal natural laboratory for studying surface ozone across remote regions of the Eurasian continent. Long-term surface ozone measurements over the Tibetan Plateau have been conducted at Mt. Waliguan (northeastern edge of the Tibetan Plateau) since 1994 (Xu et al., 2016), the Nepal Climate Observatory at Pyramid (NCO-P) which has operated on the southern slope of the Himalayan region since 2006 (Cristofanelli et al., 2010) and the Xianggelila Regional Atmosphere Background Station at the southeastern rim of the Tibetan Plateau, which has been in operation since 2007 (Ma et al., 2014). Analysis of long-term ozone mixing ratios at Waliguan Station has revealed steadily increasing concentrations over the past two decades (Xu et al., 2016) and has shown that maximum surface ozone occurs during the summer (Zhu et al., 2004). At NCO-P and Xianggelila, maximum surface ozone was observed in the spring (Cristofanelli et

al., 2010; Ma et al., 2014). It is noteworthy that these three monitoring sites are on the boundaries of the Tibetan Plateau. In the vast inland area of the Tibetan Plateau, surface ozone measurements were only reported from Lhasa and Dangxiong for one year and two years, respectively. These measurements might be less representative of regional surface ozone variation due to their proximity to human settlements and the relatively short duration of the measurements (Ran et al., 2014; Lin et al. 2015). The paucity of long-term surface ozone observations in the Tibetan Plateau, especially in the inland region, limits our understanding of the regional background ozone level and the factors that influence it in the Tibetan Plateau.

Surface ozone mixing ratios were monitored for ~5 years (January 2011 to October 2015) at Nam Co Station on the shore of Nam Co Lake (30°30′-30°56′N, 90°16′-91°03′E). In this study, we investigated the seasonal and diurnal variations of surface ozone and its influential factors. We then evaluated surface ozone variability using combined observations over the Tibetan Plateau and beyond. Finally, we discuss the potential representativeness of surface ozone at Nam Co Station as the regional background of surface ozone in the inland Tibetan Plateau. This study expands the understanding of the concentration and variations in the surface ozone concentration and the transport processes that influence tropospheric ozone in the inland Tibetan Plateau.

## 2 Measurements and Methods

### 2.1 Measurement site

The Nam Co Comprehensive Observation and Research Station (hereafter referred to as the Nam Co Station, 30°46.44′N, 90°59.31′E, 4730 m a.s.l.) is a high-altitude scientific research center located between the southeastern shore of Nam Co Lake (1 km from the station) and the foothills of the northern Nyainqêntanglha Mountains (15 km from the station) in the southern-central region of the Tibetan Plateau (Fig. 1). Nam Co Station was established in September 2005 to monitor atmospheric conditions and enabled research of the atmospheric environment in the inland Tibetan Plateau (Kang et al., 2011). Nam Co Station is located in a natural flat field (220 × 100 m) and records meteorological, ecological, and atmospheric data, including surface ozone mixing ratios (Cong et al., 2007; Li et al., 2007; Huang et al., 2012; Liu et al., 2015; de Foy et al., 2016a). The climate at Nam Co Station is dry and cold, representing a typical climate regime in a high mountain region. The solar radiation at Nam Co Station is stronger than that at other sites at the same latitude due to the high altitude and thin air. Three synoptic systems influence the atmosphere at Nam Co Station: the South Asian anticyclone (which controls the 100-hPa upper layer), a subtropical high-pressure system, and southeasterly warm and wet airflow (during the monsoon season) (Qiao and Zhang, 1994). No major anthropogenic sources of atmospheric emissions exist near Nam Co Station. The urban area closest to the station is Dangxiong County, which is located on the southern slopes of the Nyainqêntanglha Mountain Range approximately

60 km south of Nam Co. Dangxiong is lower in elevation than Nam Co Station by more than 500 m. No large industries are located within 100 km of Nam Co Station. Local traffic is limited to a small number of vehicles traveling through the area during the tourism season.

## 2.2 Measurements: surface ozone and meteorology

The surface ozone mixing ratios were measured using a UV photometric instrument (Thermo Environmental Instruments, USA, Model 49i), which uses the absorption of radiation at 254 nm and has a dual cell design. The ambient air inlet (Teflon tube) was 1.5 m above the roof and 4 m above the ground. The instrument has zero noise, 0.25 parts per billion (ppb) RMS (root mean square error) (60 s average time), a low detection limit of 0.5 ppb, a precision of 1 ppb and a response time of 20 s (10 s lag time). The instrument was calibrated using a 49i-PS calibrator (Thermo Environmental Instruments, USA) before measurements and during the monitoring periods and yearly instrument calibrations were performed against the Standard Reference Photometer (SRP) maintained by the WMO World Calibration Centre in Switzerland (EMPA). Field operators checked the instruments and created a monitoring log file every day. Due to the extreme winter weather that occurs at Nam Co Station, measurements were intermittently interrupted because of unstable power supply (due to damage to the electrical wires caused by strong winds) and equipment maintenance. All data displayed in this study are in UTC+8 format (Beijing Time, BJT), and solar noon at Nam Co Station occurs at 13:56 UTC+8.

Measurements of temperature, relative humidity, wind speed, wind direction and downward shortwave radiation (SWD) were conducted at Nam Co Station using an automatic weather station system (Milos520, Vaisala) and a radiation measurement system (CNR-1) (Ma et al., 2008).

## 2.3 Meteorological simulations

Backward trajectories and clusters were calculated using NOAA-HYSPLIT (HYbrid Single-Particle Lagrangian Integrated Trajectory) model (Draxler and Rolph, 2003, http://ready.arl.noaa.gov/HYSPLIT.php) using TrajStat, which is a free software plugin of MeteoInfo (Wang, 2014). Gridded meteorological data for backward trajectories in HYSPLIT and the planetary boundary layer height (PBLH) were obtained from the Global Data Assimilation System (GDAS-1) operated by the U.S. National Oceanic and Atmospheric Administration (NOAA) with 1°×1° latitude and longitude horizontal resolution and 23 vertical levels from 1000 hPa to 20 hPa (http: // www. arl. noaa. gov/ gdas1. php). The backward trajectory arrival height in HYSPLIT was set at 500 m (500 , 1000 and 1500 m were tested as arrival heights and there was no obvious difference in the results) above the surface and the total run times was 120 hours for each backward trajectory with a time intervals of 3

hours throughout the whole measurement period. The vertical motion was calculated using the default model selection, which used the meteorological model's vertical velocity fields. Angle distance (Sirois and Bottenheim, 1995) was selected to calculate clusters in this study.

To identify the impact of different air masses in a multiple linear regression model, WRF-FLEXPART (Stohl et al., 2005; Brioude et al., 2013) was used to obtain the clusters of particle trajectories reaching the Nam Co Station. 1000 particles were released per hour in the bottom 100 m surface layer above Nam Co Station and were tracked in backward mode for 4 days (de Foy et al., 2016a). Residence time analysis (RTA) (Ashbaugh et al., 1985) was used to create gridded fields representing the dominant transport paths of air masses impacting the measurement site (Wang et al., 2016; Wang et al., 2017). A k-means algorithm was used to classify the transport patterns into clusters (Wang et al., 2016). Six clusters were found to represent the dominant flow patterns to the Nam Co Station simulated using WRF-FLEXPART.

A tracer for stratospheric ozone incursions at the measurement site was obtained using the CAMx (Comprehensive Air-quality Model with eXtensions) v6.30 model (Ramboll Environ, 2016). The model initial and boundary conditions were obtained from ERA-Interim ozone fields, retaining only concentrations above 80 ppb and higher than 400 hPa. CAMx simulations were performed using the WRF medium and fine domains (domains 2 and 3) in nested mode for the full 4 year time series. In order to serve as a tracer for direct transport, there was no chemistry in the model and ozone was treated as a passive tracer. The resulting time series of the tracer concentration at the measurement site was used as input in the multi-linear regression model. This is similar to the procedure described in de Foy et al. (2014) to estimate the impact of the free troposphere on surface reactive mercury concentrations.

The ECMWF ERA-Interim data (Dee et al., 2011) were used to analyze the upper troposphere and lower stratosphere structures of the meridional cross-section over Nam Co Station.

**2.4 Multiple Linear Regression Model**

A multiple linear regression (MLR) model was used in this study to quantify the main factors affecting the hourly surface ozone concentrations. The method follows the description provided in de Foy et al. (2016b and 2016c). The inputs to the MLR model include meteorological parameters (wind speed, temperature and humidity), inter-annual variation factors, seasonal factors, diurnal factors, WRF boundary layer heights, WRF-FLEXPART trajectory clusters and the CAMx stratospheric ozone tracer.

Tests were performed with different variables and averaging times for the meteorological parameters, including hourly

data, running averages of 3, 8 and 24 hours, and smoothed variables using Kolmogorov-Zurbenko filters (Rao et al., 1997). The variables to be included in the regression were obtained iteratively. At each iteration, the variable leading to the greatest increase in the square of Pearsons correlation coefficient was added to the inputs as long as the increase was greater than 0.005.

The regression model is described by the following equations (Eq. 1, 2, 3):

$$\log(O_3 + 10) = \sum_{yr=2011}^{2014} \alpha_{yr} t_{yr} + f(seasons) + f(diurnal) + \sum_{i=1}^{6} \alpha_{cl_i} t_{cl_i} + \sum_{i=1}^{5} \alpha_{PBLH_i} t_{PBLH_i} + \sum_{j=1}^{4} \sum_{i=5}^{5} \alpha_{WS_{i,j}} t_{WS_{i,j}}$$
$$+ \alpha_{st} \log(Strat.Tracer) + \alpha_{bkg} + \epsilon \qquad (1)$$

$$f(seasons) = \alpha_{T_{SS}} T'_{SS} + \alpha_{T_{SS}} q'_{SS} + \sum_{j=1}^{2} \alpha_{sj} \sin\left(\frac{2\pi jt}{365.25}\right) + \alpha_{cj} \cos\left(\frac{2\pi jt}{365.25}\right) \qquad (2)$$

$$f(diurnal) = \alpha_{T_{DU}} T'_{DU} + \alpha_{q_{DU}} q'_{DU} + \sum_{i=1-11,14-24} \alpha_{hr_i} t_{hr_i} \qquad (3)$$

$\alpha$ represents the regression coefficients that are determined by the model. $O_3$ is the hourly surface ozone concentration. Time vectors $t$ are used to represent different temporal terms and vary between 0 and 1. For example, $t_{yr}$ represents the variation from year to year. This means that $t_{2011}$ is 1 during 2011 and 0 otherwise. The air mass clusters derived from WRF-FLEXPART are represented using $t_{cl}$: $t_{cl_1}$ is 1 during cluster 1 and 0 otherwise.

The impact of winds and mixing heights on $O_3$ is expected to be non-linear. We therefore separated the data into quartiles
and included a regression factor for each of the 5 points separating the quartiles (0%, 25%, 50%, 75% and 100%). For the boundary layer height, we have 5 time series represented by $t_{PBLHi}$. Piecewise linear interpolation is used to give them a value of 1 at the edge of the quartile which then decreases to 0 by the next quartile. As an example, $t_{PBLH_1}$ is 1 for the times when the mixing height is at the data minimum and it is 0 when the mixing height is above the first quartile. At times when the mixing height is between these two levels, $t_{PBLHi}$ varies linearly.

For wind speed and direction, we performed the piecewise interpolation in 2 dimensions, with 5 factors for the wind speed and 4 factors for the wind direction (because the two extremes are the same: 0° and 360°). $t_{WS_{1,1}}$ represents times when the wind speeds are in the bottom quartile ($i = 1$) and when the wind direction is from the north ($j = 1$).

We used the natural logarithm of $O_3$ concentrations offset by 10 ppb to approximate a normal distribution while reducing the long tail in the transformed variable that would be caused by taking the logarithm of low $O_3$ concentrations. The background
term is included as $\alpha_{bkg}$ and the residual between the regression model and the measurement time series is $\epsilon$.

T is the temperature and q is the specific humidity. Both are normalized linearly as follows: $T' = (T - \mu(T))/\sigma(T)$. Because we were interested in identifying separately the seasonal and diurnal variation, we used a Kolmogorov-Zurbenko filter to separate T and q into a seasonal component ($T_{SS}$ and $q_{SS}$) and a diurnal component ($T_{DU}$ and $q_{DU}$) (Rao et al., 1997). The seasonal component used 5 passes of a 13-point moving average, and the diurnal component was the difference between the hourly and the seasonal time series. The seasonal terms also include harmonic terms with a 12 month and 6 month period as described in de Foy et al. (2016c).

The diurnal term includes diurnal temperature and specific humidity. In addition, we included a time series term for each hour of the day ($t_{hri}$) except for 12:00 and 13:00 which are taken to be reference hours.

The "Strat. Tracer" was simulated using the CAMx model as described in section 2.3. During testing, the best fit in the regression model was found using the seasonal component of the stratospheric tracer obtained from the Kolmogorov-Zurbenko filter in the same way as $T_{SS}$ and $q_{SS}$.

Because the results of least-squares methods are sensitive to outliers, an iteratively reweighted least squares (IRLS) procedure was used to screen them out. Measurement times when the model residual was greater than two standard deviations of all the residuals were excluded from the analysis. This was repeated iteratively until the method converged on a stable set of outliers (de Foy et al., 2016a).

To estimate the uncertainty in the results, we used block-bootstrapping with a 24 hour block length, as described in de Foy et al. (2015). 100 realizations of the final model were performed. For each model realization, days were selected randomly with replacement from the full set of days until a dataset containing the same total number of days was obtained. This accounts for both measurement errors and model errors as the instrument errors are assumed to be uncorrelated and model errors are unlikely to be correlated beyond a couple of days. The uncertainty in the results can then be estimated by calculating the standard deviation of the parameters obtained from the 100 model realizations.

When presenting results of the regression analysis, we obtained a single time series for each group by summing the individual parts. For example to estimate the impact of the WRF-FLEXPART clusters we make a sum of the 6 $t_{cli}$ time series multiplied by the corresponding regression factors $\alpha_{cli}$. For a log-transformed regression, the time series of each group can be interpreted as a scaling factor on the background term ($\alpha_{bkg}$) and expressed as a percentage change. For a linear regression model, the time series can be expressed as a linear departure from the mean in the units of the measurements (ppb).

In all, there were 67 free parameters ($\alpha$) in the regression model that were used to fit 27,310 data points.

**2.5 Potential Source Contribution Function**

The potential source contribution function (PSCF) assumes that back-trajectories arriving at times of higher mixing ratios

likely point to the more significant pollution directions (Ashbaugh et al., 1985). PSCF has been applied in previous studies to

locate air masses associated with high levels of surface ozone for different sites (Kaiser et al., 2007; Dimitriou and Kassomenos,

2015). In this study, PSCF was calculated using trajectories that were calculated by HYSPLIT. The top of the model was set

to 10000 m. The PSCF values for the grid cells in the study domain were based on a count of the trajectory segment (hourly

trajectory positions) that terminated within each cell (Ashbaugh et al., 1985). Let $n_{ij}$ be the total number of endpoints that fall

in the $ij$th cell during whole the simulation period. Let $m_{ij}$ represents the number of points in the same cell with arrival times

at the sampling site that correspond to surface ozone mixing ratios higher than a set criterion. In this study, we calculated the

PSCF based on trajectories corresponding to concentrations that exceeded the mean level of surface ozone during measurement.

The PSCF value for the $ij$th cell was then defined as:

$$PSCF_{ij} = \frac{m_{ij}}{n_{ij}}$$

The PSCF value can be interpreted as the conditional probability that the ozone mixing ratios at the measurement site are

greater than the mean mixing ratios if the air parcel passes though the $ij$th cell before arriving at the measurement site. Cells

with high PSCF values are associated with the arrival of air parcels at the receptor site that have pollutant mixing ratios that

exceed the criterion value. These cells are indicative of areas of 'high potential' contributions for the chemical constituent.

Identical $PSCF_{ij}$ values can be obtained from cells with very different counts of back-trajectory points (e.g., grid cell A

with m$ij$=5000 and n$ij$=10000 and grid cell B with m$ij$ = 5 and n$ij$ = 10). In this extreme situation grid cell A has 1000 times

more air parcels passing through than grid cell B. Because of the sparse particle count in grid cell B, the PSCF values are more

uncertain. To account for the uncertainty due to low values of n$ij$, the PSCF values were scaled by a weighting function $W_{ij}$

(Polissar et al., 1999). The weighting function reduced the PSCF values when the total number of endpoints in a cell was less

than approximately three times the average value of the end points per cell. In this case, $W_{ij}$ was set as follows:

$$Wij \begin{cases} 1.00 & n_{ij} > 3N_{ave} \\ 0.70 & 3N_{ave} > n_{ij} > 1.5N_{ave} \\ 0.42 & 1.5N_{ave} > n_{ij} > N_{ave} \\ 0.05 & N_{ave} > n_{ij} \end{cases}$$

where $N_{ave}$ represents the mean $n_{ij}$ of all grid cells. The weighted PSCF values were obtained by multiplying the original

PSCF values by the weighting factor.

## 3 Surface ozone behavior at Nam Co Station

### 3.1 Mean mixing ratio

The mean surface ozone mixing ratio at Nam Co Station during the entire observational period was 47.6 ± 11.6 ppb (mean ± standard deviation), and the yearly average surface ozone mixing ratio was between 46.0 and 48.9 ppb (Table 1). During the whole monitoring period, the lowest hourly mixing ratio at Nam Co Station was 10.1 ppb, which was observed on December 3rd, 2011; and the highest hourly mixing ratio was 94.7 ppb, which was recorded on June 11th, 2011, resulting in a range of ~85 ppb.

The mean surface ozone mixing ratio at Nam Co Station was within the reference range reported for the Himalayas and Tibetan Plateau; it was higher than the ratios for the two nearest urban sites: Lhasa (Ran et al., 2014) and Dangxiong (Lin et al., 2015) and was comparable to of two sites on the edge of the Tibetan Plateau: Waliguan Station (Xu et al., 2011) and NCO-P (5079 m) (Cristofanelli et al., 2010) (see Fig. 1 for station locations). Surface ozone mixing ratios at Nam Co as well as ay other sites over the Tibetan Plateau were generally higher than the range of 20-45 ppb measured at background sites in the mid-latitudes of the Northern Hemispheres. This was in agreement with the higher concentrations typically observed at sites located in the free troposphere (Vingarzan, 2004).

### 3.2 Seasonal pattern

Every month considered in this study had more than 400 hours of available data (valid data for each month >56%). The surface ozone concentrations at Nam Co Station experienced similar annual cycles during each of the 5 years of measurements with slight variations (Fig. S1). The monthly average mixing ratios of ozone from 2011 to 2015 at Nam Co Station showed clear seasonal features (Fig. 2): 1) remarkably high values in the late spring-early summer; 2) low values in the winter; 3) little fluctuation during the remainder of the year except for the late spring-early summer and 4) a small peak around October in the second half of the year. Three winter months (December, January and February) had the lowest monthly mean surface ozone mixing ratios (41.0±7.6 ppb – 41.5±7.0 ppb) of the year, with variations smaller than 0.5 ppb. The mean monthly surface ozone mixing ratios increased from February to March by ~3.5 ppb, and a sharp increase from 44.5±10.4 ppb to 54.7±11.6 ppb occurred in March-April. The monthly mean mixing ratios remained above 54 ppb for the next 3 months (April, May and June), with the highest monthly mean mixing ratios occurring in May (58.6±12.2 ppb). After a large decrease in June-July (from 55.5±12.7 ppb to 44.9±11.9 ppb), the monthly mean mixing ratios of surface ozone during the second half of the year remained at low levels (ranging from 41.5±7.0 ppb to 48.0±8.6 ppb), albeit with a small increase in October.

### 3.3 Diurnal variation

The diurnal cycles at Nam Co Station showed low ozone mixing ratios at night and high ozone mixing ratios during the day, with a unimodal pattern. After a rapid increase during the morning (8:00-11:00) of 6 ppb, the surface ozone mixing ratio at Nam Co continued to increase until reaching a maximum at 18:00 (53.2±10.9 ppb); it then decreased continuously to its lowest level at 8:00 the next day. Field observations revealed that the ozone mixing ratios reached an average of 50.6±10.9 ppb during the day (9:00-20:00) and an average of 44.6±11.2 ppb during the night and early morning (21:00-8:00).

All seasons displayed similar diurnal ozone mixing ratio cycles at Nam Co Station (Fig. 3). Mixing ratios went from low levels at night to high levels during the daytime. The diurnal profile was generally characterized by a later shift from low to high concentrations in the winter than in the rest of the year, most likely as a result of the later time of sunrise. Relatively large diurnal amplitudes were observed in the spring, with much smaller diurnal amplitudes observed during the summer, the autumn and the winter.

### 4 Factors affecting surface ozone variation at Nam Co Station

### 4.1 Impact factors on seasonal variation

The regression model had 27,310 hourly data points of which 26,005 were retained by the IRLS procedure, see Table 2. The correlation coefficient (r) was 0.77 for the entire time series and 0.81 without the outliers. The time series of the model is shown in Fig. 4 and scatter plots between the measurements and the model are shown in Fig. S2. Note that because stratospheric intrusion contributions are seasonal, there was covariance between the stratospheric tracer and the seasonal signal. Uncertainties in the estimate of the contribution from one of these therefore impacted the estimate from the other.

As described in section 2.4, we combined the time series in the regression model into 7 distinct groups and present the results for these merged factors. In the log-transformed model, the time series for each group corresponds to a scaling factor that is applied to the baseline ozone concentration. This is shown in Fig. 4 for the stratospheric tracer and for the seasonal scaling. The regression model suggests that both stratospheric transport and seasonal variation can lead to enhancements in hourly concentrations of up to 20% of the baseline level of ozone.

The contribution of each group to the O3 variation can be calculated as the variance of the time series for that group divided by the sum of the variances for all the groups shown in Table 2 and described in Eqs. 1-3. The uncertainty of the results was calculated using the standard deviation of the 100 realizations of the model using block-bootstrapping. The stratospheric ozone tracer from the CAMx model contributed 18.2±2.6% of the ozone variance at the site and the WRF-FLEXPART wind

transport clusters (Fig. S3) contributed 6.5±1.7%. Local winds accounted for 31.0±1.8%, seasonal variations (including the 12 and 6-month sine and cosine terms, and the seasonal temperature and humidity terms) accounted for 35.3±3.0%, diurnal signals (including the hourly terms and the diurnal temperature and humidity signals) accounted for 7.4±0.8%, the annual signal for 1.5±0.5% and the WRF boundary layer height for 0.1±0.1% of the variance. The uncertainties in the model regression results are shown graphically in Fig. S4. The histograms show variation in the contribution to ozone variance for each group based on the 100 realizations of the model. Taking the CAMx tracer as an example, the model suggests that this term contributes 17.7% of the O3 variance on average, but the results range from 12 to 24% and have a standard deviation of 2.6%. The scatterplots show the covariance between the model estimates for different groups. Most groups do not covary and hence there is no correlation between the x and y axis and the correlation coefficients are low. For example, the contribution from the WRF-FLEXPART clusters does not covary with the CAMx tracer, and $r^2 = 0.03$. In contrast, the seasonal signal covaries with the CAMx tracer ($r^2 = 0.5$) which suggests that there is an increased uncertainty in the estimates for these terms and that changes in estimates for the seasonal signal will lead to changes in the estimates for the CAMx tracer. In this case, this is because stratospheric intrusions occur in spring and hence there is an inescapable correlation between the two groups.

As a separate test, the regression model was performed with linear transformations instead of log-transformations. The model is similar to Eq. 1 but with the logarithm terms replaced by a linear normalization for the ozone concentration and for the stratospheric tracer. The advantage of this model is that the results can be interpreted as ozone enhancements from each group in ppb. The results are shown in Table 2. Although the fit was not as good, the results were remarkably similar. The contribution of the stratospheric tracer was lower, mainly because there were individual peaks which had a larger influence in the linearly transformed model than in the log-transformed model. Fig. S5 (corresponding to Fig. 4) shows the linear results. Although the mean contribution of the stratospheric tracer to surface ozone concentrations was only 1 ppb over the entire time series, it can reach above 20 ppb during specific events in the spring.

Potential vorticity from the ERA-Interim model at 500 hPa, which was near the surface at Nam Co, was not found to contribute to the simulated ozone time series. However, at 350 hPa a positive correlation was found. The correlation was even larger if we took the potential vorticity at 350 hPa above the Himalayas. Total column ozone correlated more weakly with surface ozone than potential vorticity and was not found to improve the regression model. As for potential vorticity, the correlation coefficient for total column ozone was higher above the Himalayas than at the measurement site. Fig. S6 shows the 24-hour running average of the surface ozone and the stratospheric tracer at the measurement site, and the total column ozone and the potential vorticity from ERA-Interim above the Himalayas.

We performed a separate model run where we replaced the stratospheric tracer with the potential vorticity time series at

350 hPa above the Himalayas. The model found the best fit using the Kolmogorov-Zurbenko seasonally filtered time series of

potential vorticity. The model had a slightly lower correlation coefficient, and lower contribution of the potential vorticity

tracer (5.8%) than the model using the CAMx stratospheric tracer. This suggests that the CAMx stratospheric tracer is a better

indicator of stratospheric ozone incursions than the time series of potential vorticity.

The regression model was also performed by season, as shown in Table S1. This results show that the largest stratospheric

incursions occurred in the spring (Mar, Apr, May) with 20% contribution to ozone variation, and did not impact surface ozone

in the fall (Sep, Oct, Nov). The air mass transport clusters accounted for nearly 10% of the ozone variation in the summer (Jun,

Jul, Aug) but very little otherwise.

To visualize the transport of ozone from the stratosphere to the troposphere, we analyzed the upper troposphere and lower

stratosphere structures of the meridional cross-section of monthly mean ERA-Interim data above Nam Co Station (Fig. 5). In

the spring (Mar, Apr and May), the dynamical tropopause (identified by the isolines of 1 and 2 potential vorticity unit) exhibited

a folded structure over the Tibetan Plateau. This tropopause folding can lead to a downward transport of ozone from the

stratosphere to the troposphere. Tropopause folding occurred in the southern Tibetan Plateau and closer to Nam Co Station in

the spring. Cosmogenic $^{35}$S results (Lin et al., 2016) also indicated that in the spring, Nam Co was affected by aged stratospheric

air originating over the Himalayas rather than being affected by transport from fresh stratospheric air masses above Nam Co

Station. The larger diurnal amplitude of surface ozone in the spring than in other seasons (Fig. 3, mentioned in section 3.3)

may be related to four factors: (1) position of the STE (stratosphere–troposphere exchange) hot spot; (2) frequency of STE; (3)

PBLH at Nam Co Station and (4) solar radiation at Nam Co Station. In the spring, plots of tropopause folding suggest that

STE mostly occurs in the southern Tibetan Plateau which is close to Nam Co Station. Furthermore, PBLH at Nam Co Station

was higher in the spring than during the rest of the year. The higher PBLH in the spring facilitated the impact of downward

transport from the stratosphere to Nam Co Station. The spring also has more intense solar radiation than the summer because

the monsoon leads to increased cloudiness in the summer. Pearson's correlation coefficient between monthly SWD and surface

ozone was ~0.93 in 2012 (2012 was selected because it had a more complete dataset than the other years) (Fig. 6), indicating

that monthly surface ozone variability at Nam Co Station was associated with solar radiation. This was expected as increased

solar radiation promotes the photochemical production of surface ozone in the spring, which is similar to the mechanism at

other background sites (Monks 2000). Consequently, more photochemical production of ozone is expected in the spring. In

the summer (Jun, Jul and Aug), the jet core moved to the northern Tibetan Plateau and tropopause folding was relatively farther

from Nam Co Station than those in the spring. Consequently, there was a smaller impact of stratospheric air at Nam Co Station. With tropopause folding further north in the summer, the air masses from the northern Tibetan Plateau may contribute more to the surface ozone levels at Nam Co Station than the air masses from the southern Tibetan Plateau. Ojha et al. (2017) found that the potential vorticity layer in the summer was weaker than during the late winter and in the spring in the central Himalayan region which is to the south of Nam Co Station. In the autumn (Sep, Oct and Nov) and the winter (Dec, Jan and Feb), the mixing heights at Nam Co Station were much lower than those in the spring and the summer. Furthermore, SWD in the autumn and the winter was weaker than those in the spring and the summer. These factors contributed to the relatively low level of surface ozone at Nam Co Station in the autumn and the winter.

### 4.2 Impacts of vertical mixing and photochemical production on diurnal variation

Wind speed and PBLH are generally regarded as the main factors influencing the diurnal cycle of surface ozone. High wind speed was found to covary with turbulent downward mixing in previous studies in the Tibetan Plateau (Tang et al., 2002; Ma et al., 2014; Lin et al., 2015). A lake-land breeze influenced Nam Co Station, and the wind speed in the daytime was higher than that at night (Fig. S7). The hourly average wind speed and PBLH at Nam Co Station showed a positive correlation with the hourly average surface ozone (Fig. 7). The correlation coefficient between hourly average surface ozone and hourly average wind speed was 0.95 and the correlation coefficient between the hourly average surface ozone and the hourly average PBLH was 0.92. These results indicate that high levels of surface ozone are associated with high wind speeds and high mixing heights. In addition, local photochemical production may also contribute to the higher concentration of surface ozone at Nam Co Station in the daytime.

### 5 Synthesis comparison of surface ozone variation across the Tibetan Plateau and beyond

### 5.1 Diurnal variation

Diurnal surface ozone patterns varied among sites across the Tibetan Plateau (Fig. 8). Nam Co Station, Xianggelila, Lhasa and Dangxiong showed similar diurnal surface ozone patterns as discussed in section 4.2.

Diurnal surface ozone at NCO-P showed different patterns in different seasons (Fig. 8), and thermal circulation was the most influential factor (Cristofanelli et al., 2010). The surface ozone mixing ratio at Waliguan experienced a minimum around noon and a maximum at night (Fig. 8), which is indicative of a mountain-valley breeze (local anabatic and catabatic winds) (Xue et al., 2011). Specifically, more boundary layer air affected Waliguan and resulted in lower surface ozone at noon; whereas at night, more air masses from the free troposphere increased the surface ozone level (Xu et al., 2011). It should be noted that the amplitudes in the diurnal variations at Waliguan were much smaller than those at other sites.

In general, diurnal surface ozone variations across the Tibetan Plateau were typically controlled by site-specific meteorological conditions and photochemical production. Sites located in plains or valleys exhibited daytime ozone maxima associated with vertical mixing and photochemical production, whereas mountain-top sites exhibited daytime ozone minima associated with up-slope flow of low-ozone air.

**5.2 Seasonal variation**

The seasonal variation of surface ozone mixing ratios at different sites around the world is influenced by many factors including stratospheric intrusion, photochemical production, long-range transport of ozone or its precursors, local vertical mixing and even deposition (Vingarzan, 2004; Ordónez et al., 2005; Tang et al., 2009; Reidmiller et al., 2009; Cristofanelli et al., 2010; Langner et al., 2012; Ma et al., 2014; Lin et al., 2015; Ran et al., 2014; Xu et al., 2011; Macdonald et al., 2011;

Pochanart et al., 2003; Derwent et al., 2016; Lin et al., 2014; Tarasova et al., 2009; Gilge et al., 2010; Wang et al., 2011; Wang et al., 2009; Zhu et al., 2004; Zhang et al, 2015; Nagashima et al., 2010). The seasonal variation of ozone at sites across the Tibetan Plateau and at the ridge of the Himalayas can be divided into the summer-maximum and spring-maximum type based on the location of the sites:

A) The northern Tibetan Plateau: summer-maximum type.

In the northern Tibetan Plateau (Waliguan site), surface ozone experienced a maximum in the summer and a minimum in the winter (Fig. 9A). The summer maximum of surface ozone at Waliguan was linked to the impact of a high ozone band between 35°N-45°N over 70°E-125°E (Zhu et al., 2004). Similarly, the Qinghai Lake site also experienced a maximum in the summer (Shen et al., 2014). Horizontal and vertical wind transport have been regarded as major contributor to surface ozone at these two sites (Zhu et al., 2004; Shen et al., 2014).

B) The central Tibetan Plateau: spring-maximum type.

Sites in the central Tibetan Plateau including Nam Co Station experienced maximum ozone levels during the late spring-early summer and relatively low levels during the remainder of the year (Fig. 9B), corresponding to the spring-maximum type. Compared with the surface ozone levels at Nam Co Station, those at Lhasa and Dangxiong were much lower. It is possible that titration of ozone by NOx at the urban sites as well as the differences in altitude and meteorology may lead to differences

between the surface ozone concentrations at Nam Co Station and those at Lhasa and Dangxiong. A study at Dangxiong revealed that the higher rainfall in the summer caused the surface ozone levels to remain relatively low during the warm period (July-September) (Lin et al., 2015). At Lhasa, photochemistry was the main factor affecting surface ozone in the spring and summer,

whereas transport largely contributed to the observed ozone mixing ratios in the autumn and the winter (Ran et al., 2014). The large-scale background of surface ozone in the spring is considered an important influence on Dangxiong and Lhasa during this season (Lin et al., 2015; Ran et al., 2014).

C) The southern Tibetan Plateau and the southern ridge of the Himalayas: spring-maximum type.

In the southern Tibetan Plateau and the southern ridge of the Himalayas, Xianggelila and NCO-P each had a single surface ozone peak in the spring (pre-monsoon) and a minimum in the summer (monsoon) with the difference between the two exceeding 30 ppb. This pattern is different from those of the northern and central Tibetan Plateau (Fig. 9C). At NCO-P, frequent stratospheric intrusions were recorded in all seasons except during the monsoon season (Cristofanelli et al., 2010). A similar frequency of downward transport was identified at Xianggelila, including less frequent intrusions in the summer (Ma et al., 2014).

**5.3 Backward trajectories and PSCF results of surface ozone at Nam Co Station**

Backward trajectories and PSCF were utilized to identify the air masses associated with high levels of surface ozone at Nam Co Station and to assess the regional representativity of surface ozone at Nam Co Station. In the spring, the air masses that arrived at Nam Co Station were predominantly from the west and the south, and the 3-D clusters indicated that the air masses traveled through the Himalayas before reaching Nam Co Station (Fig. 10). Cristofanelli et al. (2010), Putero et al. (2016) and Chen et al. (2011) found that the frequency of stratospheric intrusions in the Himalayas was high in the spring; and slightly lower than during the winter. This was confirmed by analysis of the ERA-Interim data set, which showed that the seasonal average ozone flux from the stratosphere to the troposphere in the Himalayas was high in the spring (Škerlak et al., 2014). The contribution of polluted air masses in driving ozone variability at the southern ridge of the Himalayas was remarkable in the spring and it may also have had an effect on the level of surface ozone at Nam Co Station through transport. In the summer, there are more backward trajectories originating from the northern Tibetan Plateau than in the other seasons (Fig. 10). During the summer, the northern Tibetan Plateau is the hot spot of stratosphere-to-troposphere ozone flux; and during autumn this flux remains higher than the one in the southern Tibetan Plateau (Škerlak et al., 2014). The summer peak of surface ozone at Waliguan also suggests that the northern Tibetan Plateau and northwestern China (a band between 35°N-45°N over 70°E-125°E) have their highest level of surface ozone in the summer (Zhu et al., 2004).

HYSPLIT backward trajectories arriving at Nam Co Station in the spring and summer were classified into 6 clusters respectively (Fig. 11). In the spring, clusters that originated from the southern Tibetan Plateau had higher mean surface ozone

levels than clusters that originated from the northern Tibetan Plateau. Air masses transported from the Himalayas therefore led to higher concentrations of surface ozone at Nam Co Station. The higher level of surface ozone at NCO-P (Cristofanelli et al., 2010) than at Nam Co Station in the spring may also reflect this possibility. In the summer, clusters from the northern Tibetan Plateau had higher mean surface ozone levels than clusters that originated from the southern Tibetan Plateau. The air masses that arrived at Nam Co Station from the northern Tibetan Plateau and northwestern China by horizontal wind transport likely resulted in the higher ozone concentrations at Nam Co Station in the summer.

Using PSCF, we have identified air masses associated with higher surface ozone at Nam Co Station in different seasons (Fig. 12) and throughout the measurement periods (Fig. S8). The Himalayas region to the south of Nam Co Station and South Asian countries including Nepal, India Pakistan, Bangladesh and Bhutan had high PSCF weight values in both the spring and summer. The large areas of northwestern China, including the northern Tibetan Plateau, were the additional regions with potentially high PSCF weight values in the summer. The PSCF values for both the southern and northern Tibetan Plateau in the autumn were smaller than those in the spring and summer. In the autumn, the inland Tibetan Plateau appeared to have a larger impact on the study site than regions more on the edge of the Tibetan Plateau. In the winter, no obvious region was identified as a potential source region, which was likely due to low surface ozone mixing ratios in all these areas. PSCF probably picked up the contribution from STE as a signal from the south in the spring and from the north in the summer, and the transport of pollution from the Indo-Gangetic Plain and Himalayan foothills was also probably picked up by PSCF.

**5.4 Implication for measurement and study of surface ozone in the inland Tibetan Plateau and beyond**

The changes in the atmospheric environment of the Tibetan Plateau are of great concern due to its rapid responses and feedbacks to regional and global climate changes. The Tibetan Plateau covers vast areas with varying topography; however, comprehensive monitoring sites are limited and sporadically distributed. Analysis of the atmospheric composition at Waliguan in the north and Everest in the south of the Tibetan Plateau has shown that they are representative of high-altitude background sites for the Tibetan Plateau. It is noteworthy that the Tibetan Plateau, as a whole, is primarily regulated by the interplay of the Indian summer monsoon and the westerlies, and the atmospheric environment over the Tibetan Plateau is heterogeneous. Mount Everest is representative of the Himalayas on the southern edge of the Tibetan Plateau and is the sentinel of South Asia where anthropogenic atmospheric pollution has been increasingly recognized as disturbing the high mountain regions (Decesari et al., 2010; Maione et al., 2011; Putero et al., 2014). In addition, Mount Everest has been identified as a hot spot for stratospheric- tropospheric exchange (Cristofanelli et al., 2010; Škerlak et al., 2014) where the surface ozone is elevated from the baseline during the spring due to frequent stratospheric intrusions. Waliguan, in the northern Tibetan Plateau, is

occasionally influenced by regional polluted air masses (Zhu et al., 2004; Xue et al., 2011; Zhang et al., 2011) and the impacts of anthropogenic emissions on Waliguan occurr mainly in the summer (Xue et al., 2011). Nam Co Station, in the inland Tibetan Plateau, is distant from both South Asia and northwestern China; it has been found to be influenced by episodic long-range transport of air pollution from South Asia (Xia et al, 2011; Lüthi et al., 2015), evidenced by the study of aerosol and precipitation chemistry at Nam Co Station (Cong et al., 2007; Cong et al., 2010). As for surface ozone, Nam Co Station is less directly influenced by stratospheric intrusions than NCO-P and is minimally influenced by local anthropogenic emission. Nam Co Station showed distinct seasonal and diurnal variation patterns compared with those sites in the Himalayas and the northern Tibetan Plateau presented earlier. Our measurements of surface ozone at Nam Co Station are essential datasets of the inland Tibetan Plateau. More long-term measurements are needed to enable a better spatial coverage and a comprehensive understanding of regional surface ozone variations and underlying influential mechanisms.

## 6 Summary

Surface ozone mixing ratios and meteorological parameters were continuously measured from January 2011 to October 2015 at Nam Co Station in the inland Tibetan Plateau. The inter-annual mixing ratios of surface ozone were stable with an average of 47.6±11.6 ppb throughout the monitoring period. The surface ozone mixing ratios at Nam Co Station were high in the spring and low in the winter. The diurnal cycle indicated that the ozone mixing ratio continued to increase after sunrise until sunset and was higher in the daytime than at night.

The mixing ratio of surface ozone at Nam Co Station is mainly controlled by various natural factors. Downward transport of air masses, air masses from the southern Tibetan Plateau in the spring and from the northern Tibetan Plateau in the summer contributed to the elevated monthly concentrations of ozone at the surface. Diurnal peaks of surface ozone in the afternoon were associated with high SWD, high PBLH and high wind speed. The analysis suggests that the maximum contribution of stratospheric intrusions to variability in surface ozone at Nam Co Station is approximately 20%. Further analysis of tropopause folding suggests that Nam Co Station is affected by air masses associated with stratospheric intrusions transported from the southern and northern Tibetan Plateau, mainly during the spring and the summer, respectively.

Surface ozone at Nam Co Station showed distinct seasonal and diurnal variation patterns as compared with other sites in the Himalayas and the northern Tibetan Plateau. The monthly maximum of surface ozone at Nam Co Station, which is in the inland Tibetan Plateau occured later in the year compared with the sites in the southern Tibetan Plateau and the southern ridge of the Himalayas, but earlier in the year than at the sites in the northern Tibetan Plateau.

Our measurements contribute to the understanding of ozone cycles and related physico-chemical and transport processes over the Tibetan Plateau. More long-term measurements of surface ozone at field sites covering the spatially extensive Tibetan Plateau are needed to improve our understanding of surface ozone variations and the underlying influential mechanisms.

**Data availability**

All the data presented in this paper can be made available for scientific purposes upon request to the corresponding
authors (Qianggong Zhang (qianggong.zhang@itpcas.ac.cn) or Shichang Kang (shichang.kang@lzb.ac.cn)).

**Acknowledgements**

This study was supported by the National Natural Science Foundation of China (41371088, and 41630754) and the Strategic Priority Research Program (B) of the Chinese Academy of Sciences (XDB03030504). The authors are grateful to NOAA for providing the HYSPLIT model and GFS meteorological files. The authors thank Yaqiang Wang who is the
developer of MeteoInfo and who provided selfless help. Finally, the authors would like to thank the editor and referees of this manuscript for their helpful comments and suggestions.

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

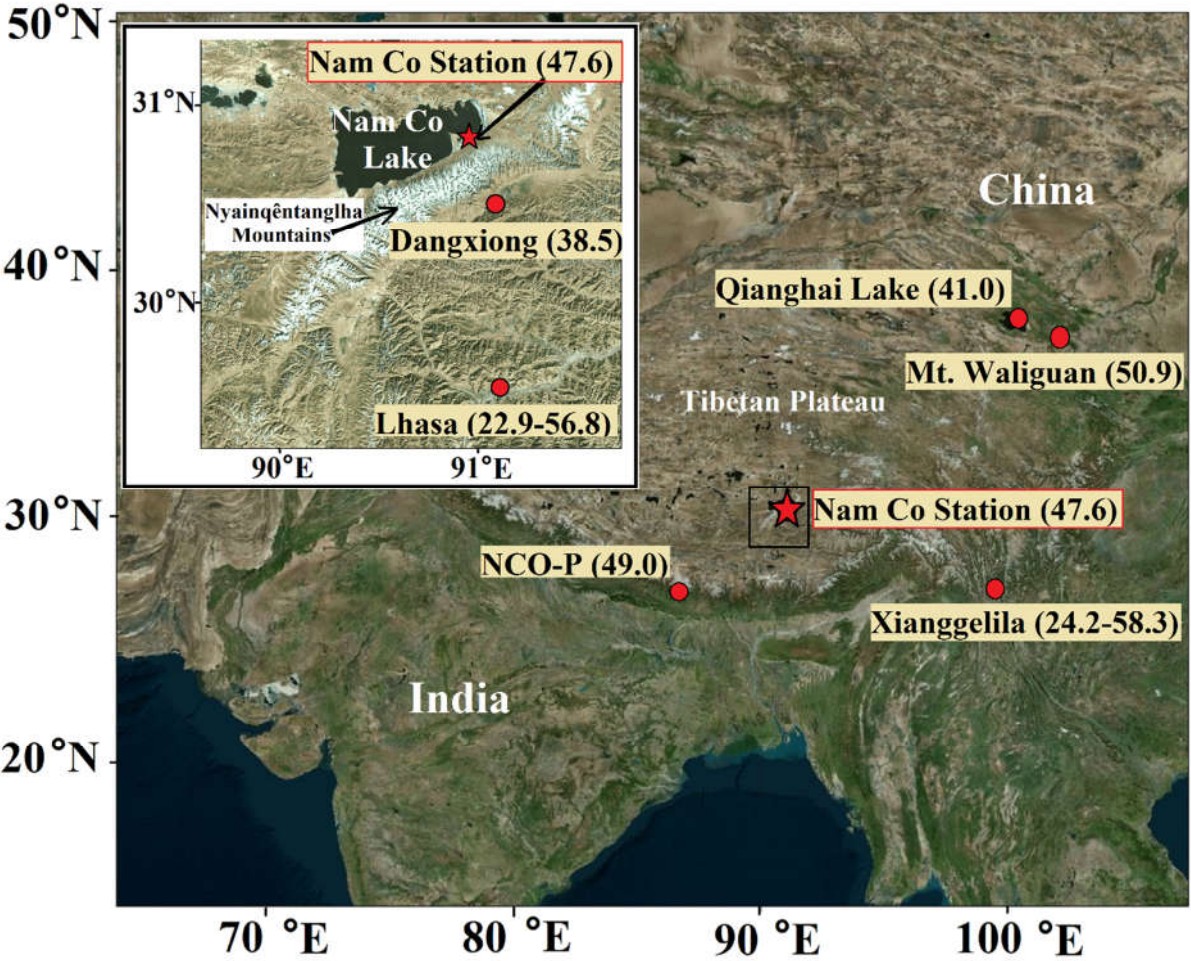

**Fig. 1. Geographical location of Nam Co Station and other sites in the Tibetan Plateau. Values in the parenthesis refers to the average or range of surface ozone in ppb as obtained from Cristofanelli et al., 2010; Lin et al., 2015; Shen et al., 2014; Xu et al., 2011; Ma et al., 2014; Ran et al., 2014.**

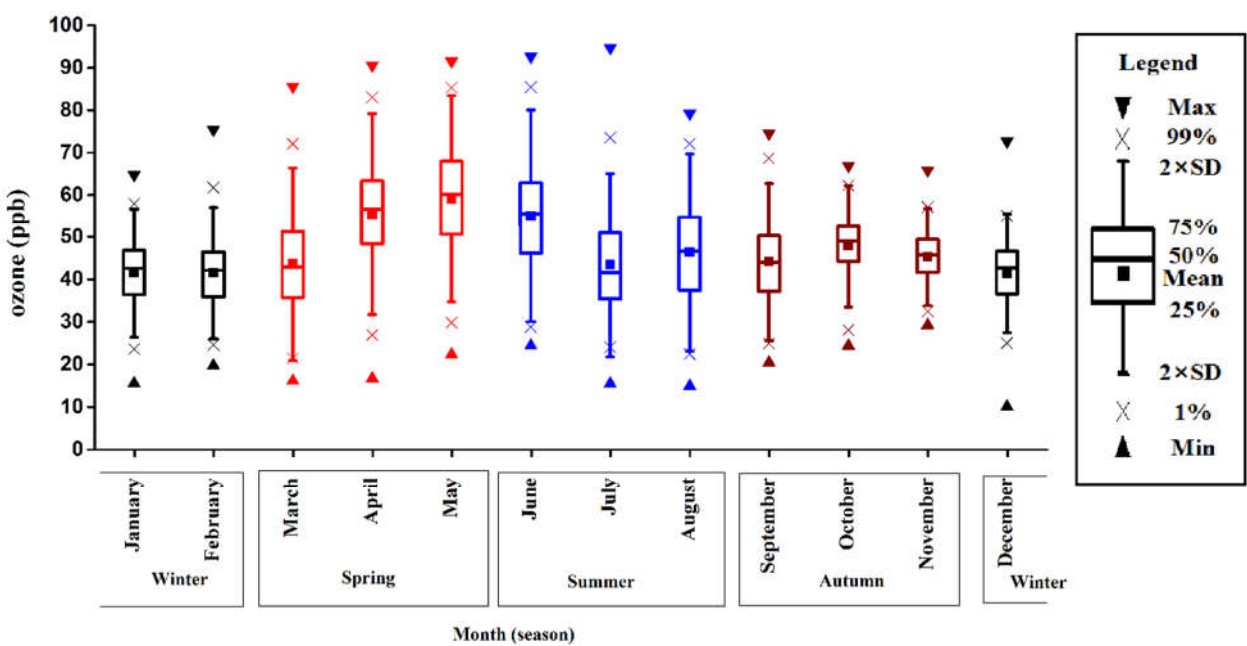

Fig. 2. Monthly average and statistical parameters of surface ozone at Nam Co Station during the whole measurement period (spring (MAM) in red; summer (JJA) in blue; autumn (SON) in dark red; winter (DJF) in black).

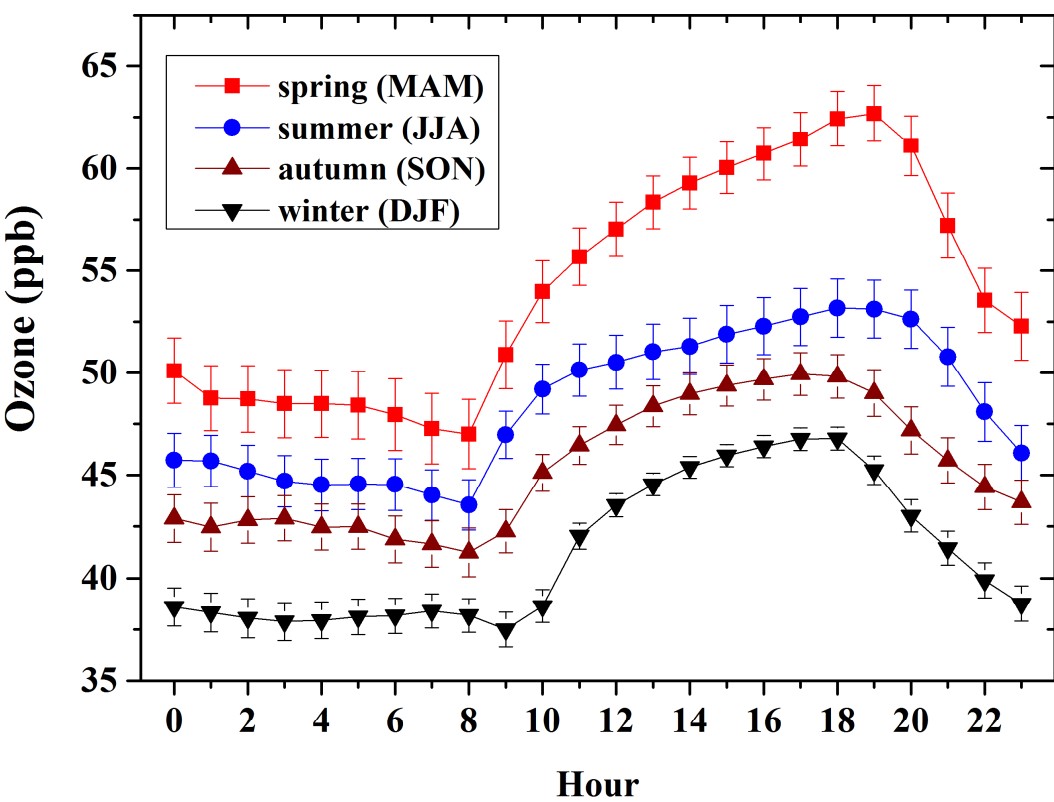

Fig. 3. Diurnal profiles of average hourly surface ozone at Nam Co Station by seasons. Error bars show the 95% confidence intervals.

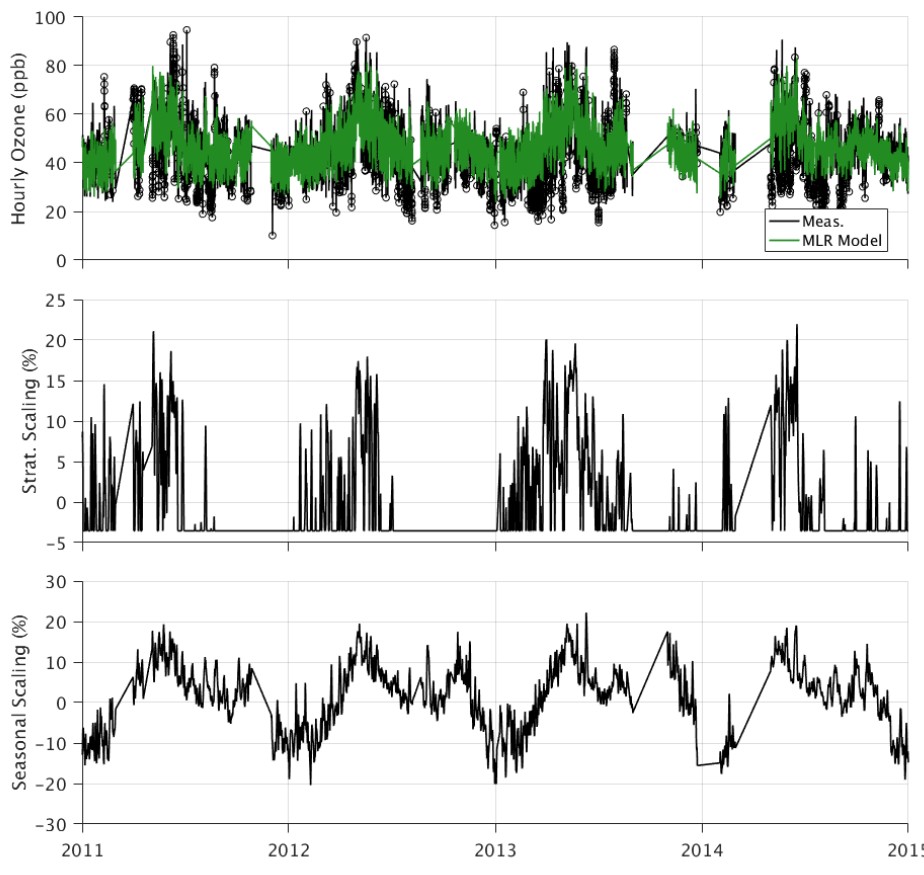

**Fig. 4. Top: Surface hourly measurements of ozone at Nam Co (black) and multi-linear regression (MLR) model fit (green). Outliers rejected by the Iteratively Reweighted Least Squares Procedure are shown as circles. Middle: Impact of the CAMx stratospheric tracer on surface ozone concentration in the regression model expressed as a**

740 **percentage change relative to the model average. Bottom: Impact of the seasonal factor (Eq. 2) on surface ozone concentration.**

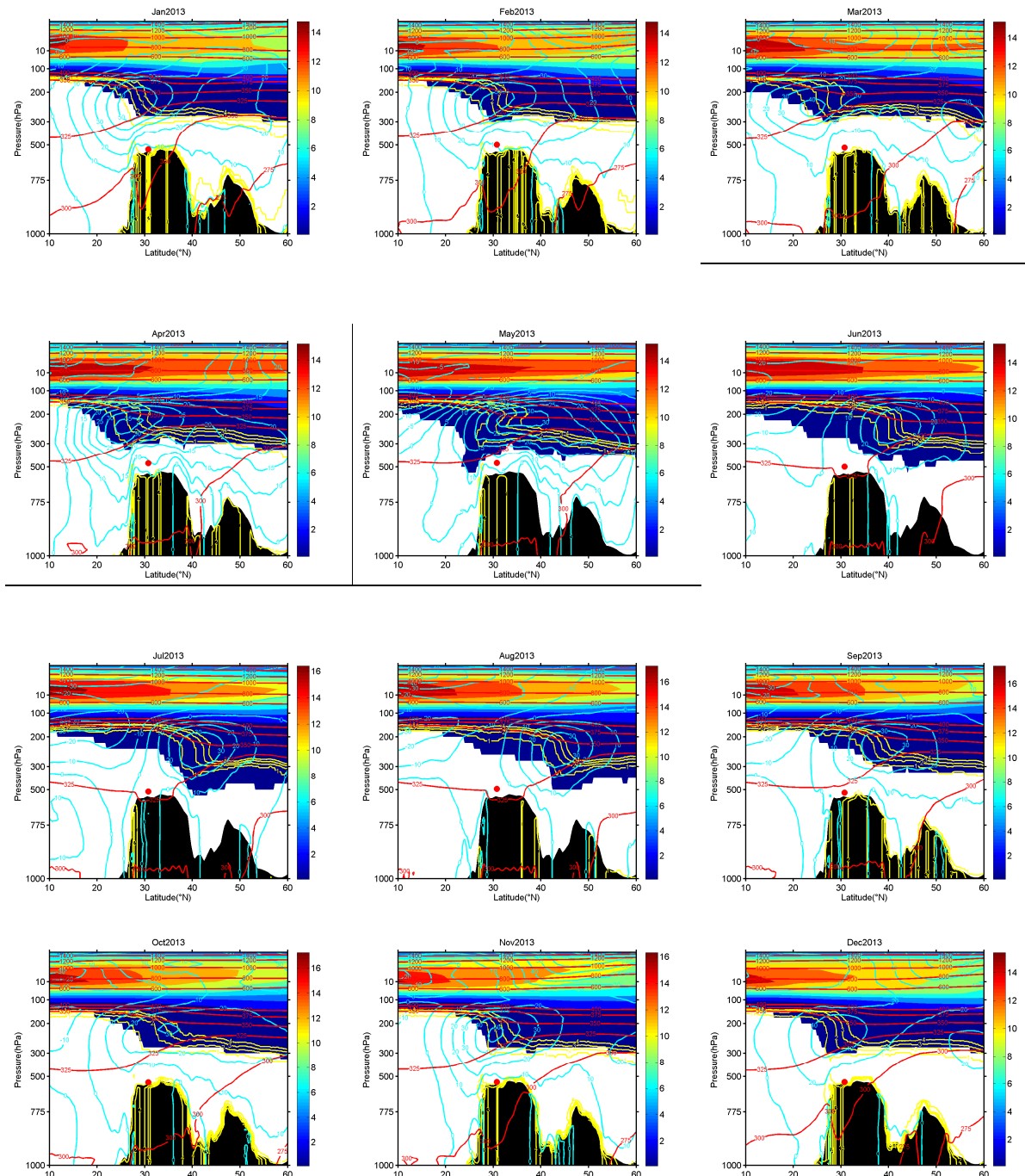

**Fig. 5. Monthly mean meridional cross-section at 91°E (over Nam Co Station ) at 20:00 UTC+8 in 2013, derived from ERA-Interim data, including zonal winds (cyan contours, m/s), potential vorticity (yellow lines, contours of 1, 2, 3, 4 potential vorticity unit), ozone (solid color, ×10⁶ kg/kg) and potential temperature (red contours, K). The**
765 **color bar shows the scale for the contour plots of ozone concentration. The area in black shows the cross section of the Tibetan Plateau terrain. The red dots show the position of the top of PBL at Nam Co Station.**

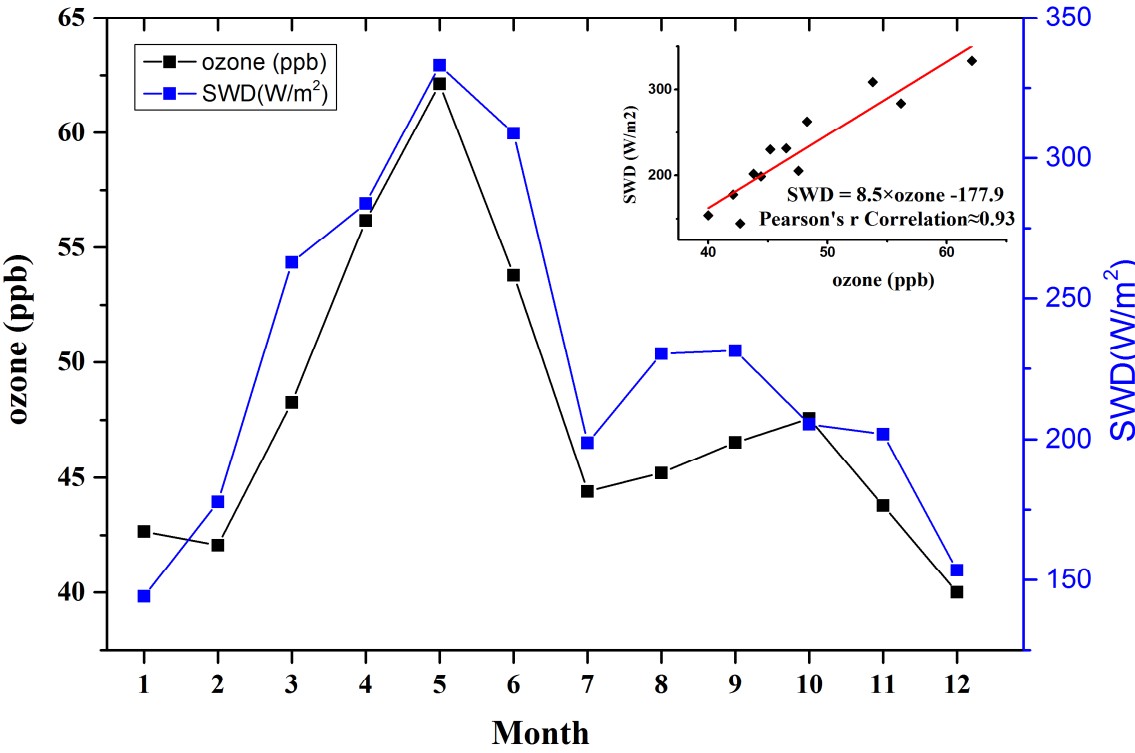

**Fig. 6. Comparison between monthly average surface ozone (black) and monthly average SWD (downward shortwave radiation, blue) at Nam Co Station in 2012.**

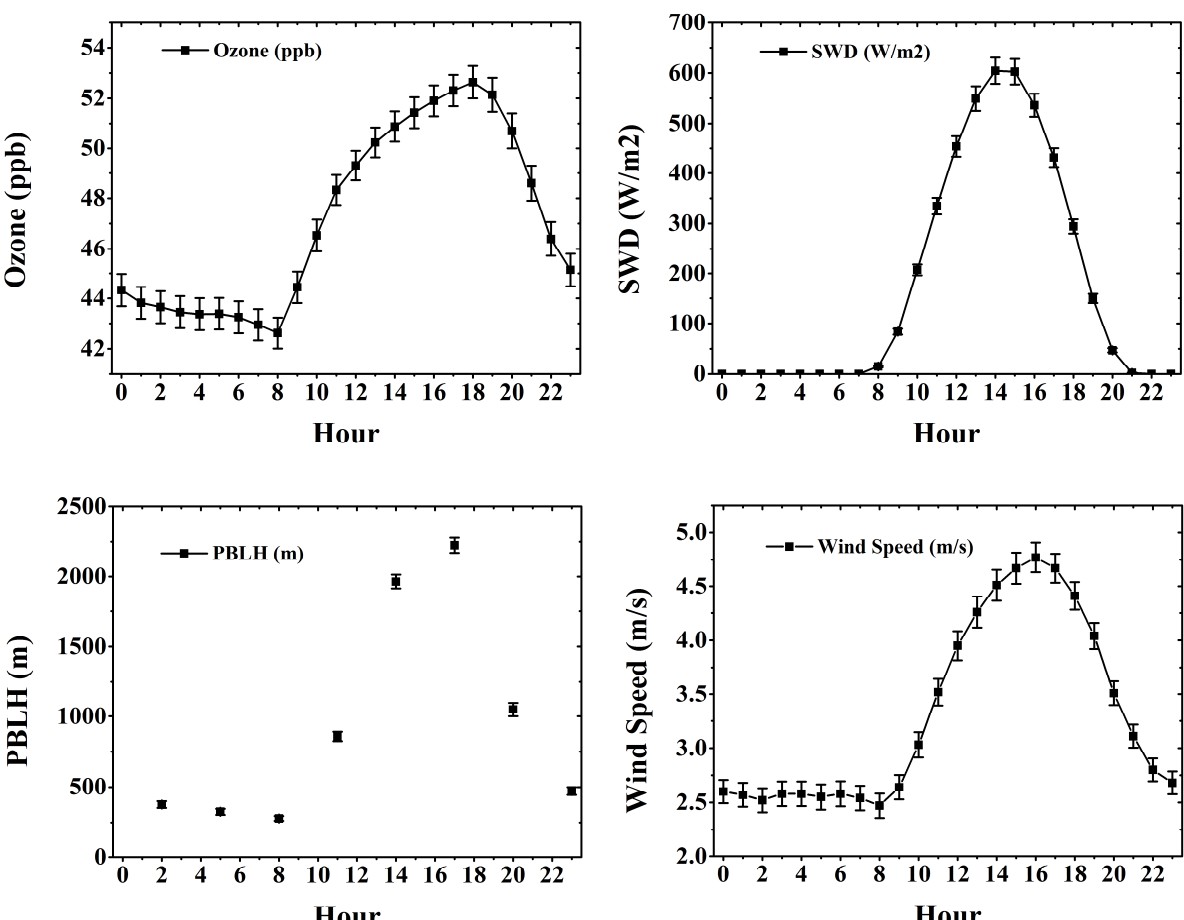

**Fig. 7 Diurnal variations of hourly average of surface ozone, SWD (downward shortwave radiation), wind speed and PBLH (planetary boundary layer height) during the whole measurement period at Nam Co Station. Error bars show the 95% confidence intervals.**

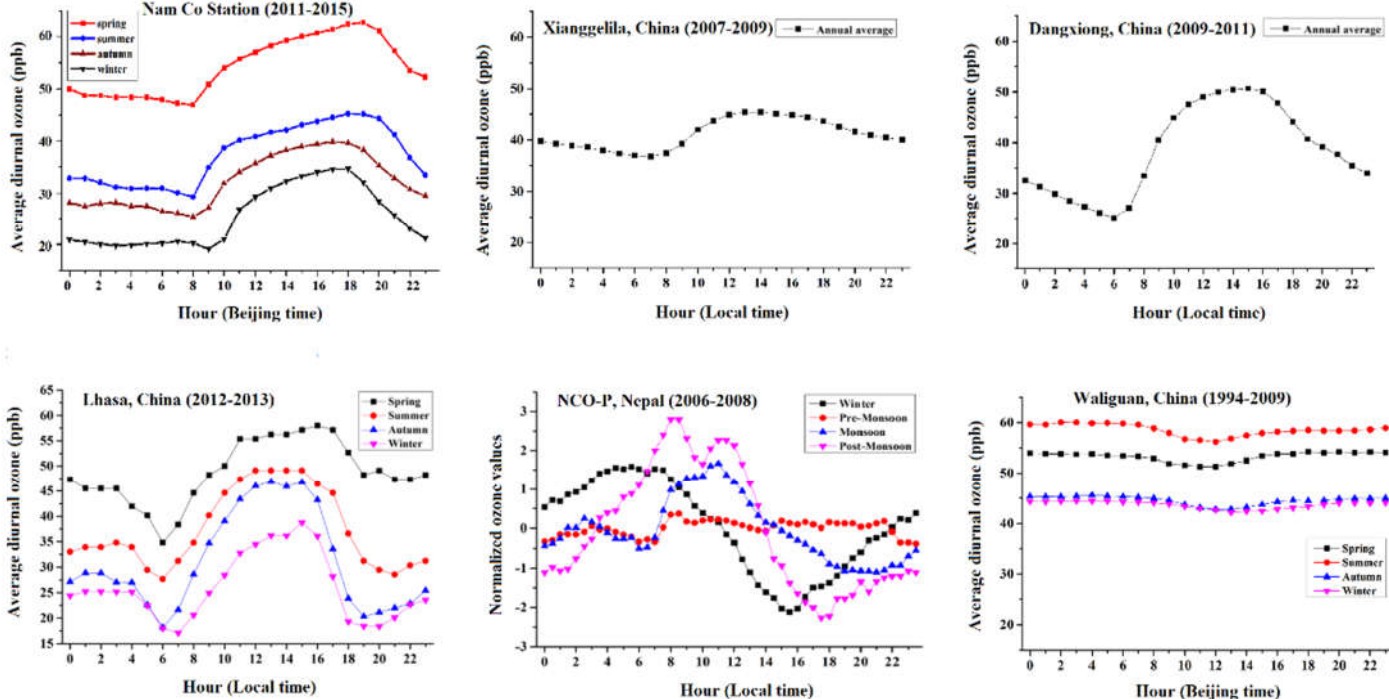

**Fig. 8. Comparison of diurnal profiles of surface ozone concentration at different sites in the Tibetan Plateau (referred to Ma et al., 2014; Lin et al., 2015; Ran et al., 2014; Cristofanelli et al., 2010; Xu et al., 2011.) Measurement years at different sites are displayed in brackets.**

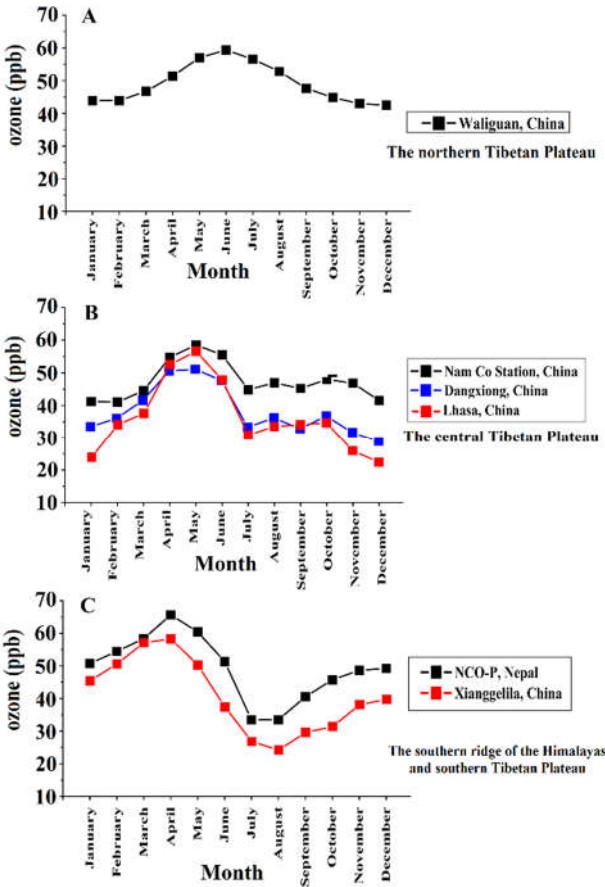

**Fig. 9. Monthly variation of surface ozone at different sites in the Tibetan Plateau (right, A: The northern Tibetan Plateau: Summer-maximum type; B: The central Tibetan Plateau: Spring-maximum type and C: The southern Tibetan Plateau and the southern ridge of the Himalayas: Spring-maximum type) (referred to Ma et al., 2014;**
**Lin et al., 2015; Ran et al., 2014; Cristofanelli et al., 2010; Zhu et al., 2004).**

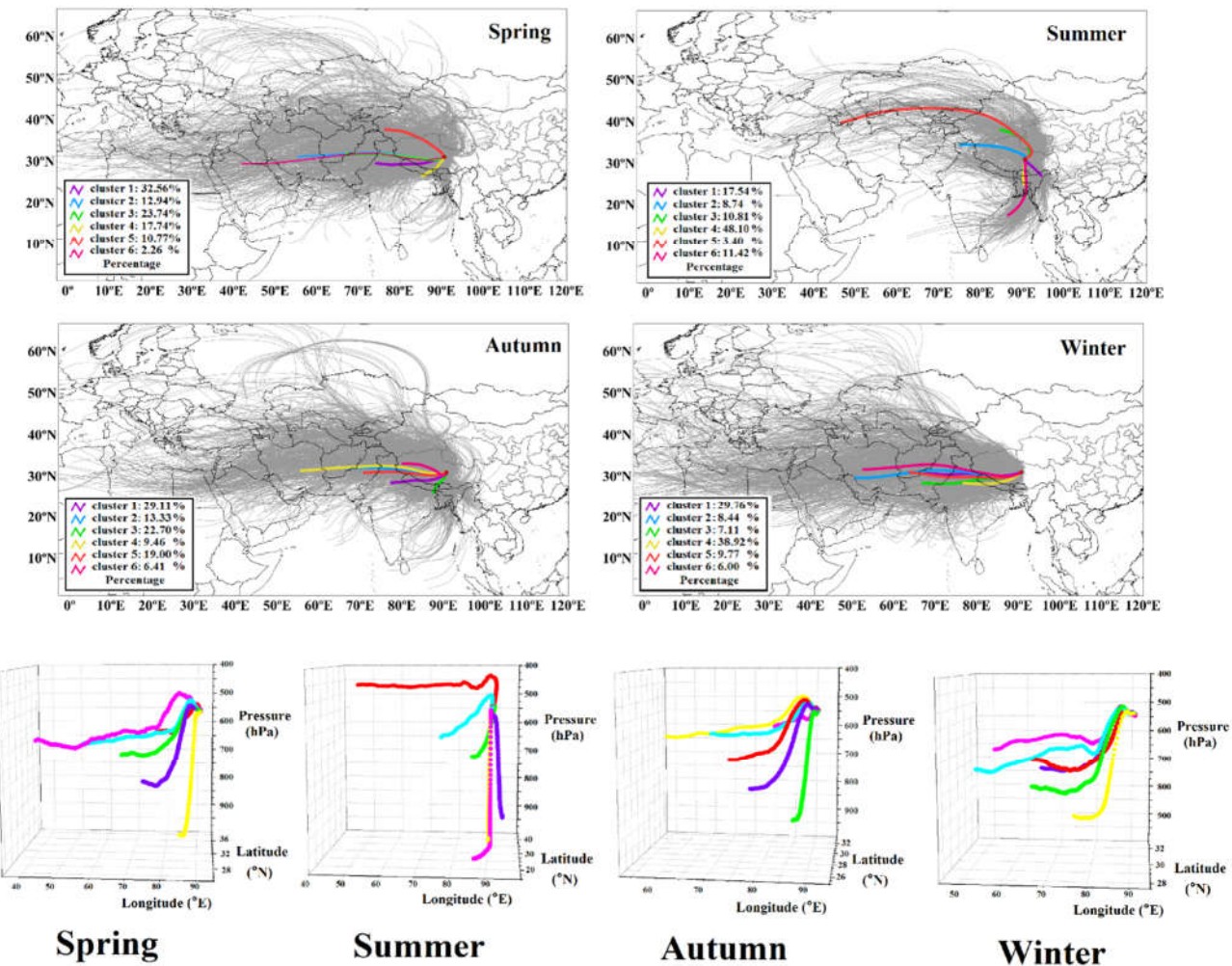

**Fig. 10.** Backward HYSPLIT trajectories for each measurement day (black lines in the maps), and mean back-trajectory for 6 HYSPLIT clusters (colored lines in the maps, 3D view shown on the right of the maps) arriving at Nam Co Station by season.

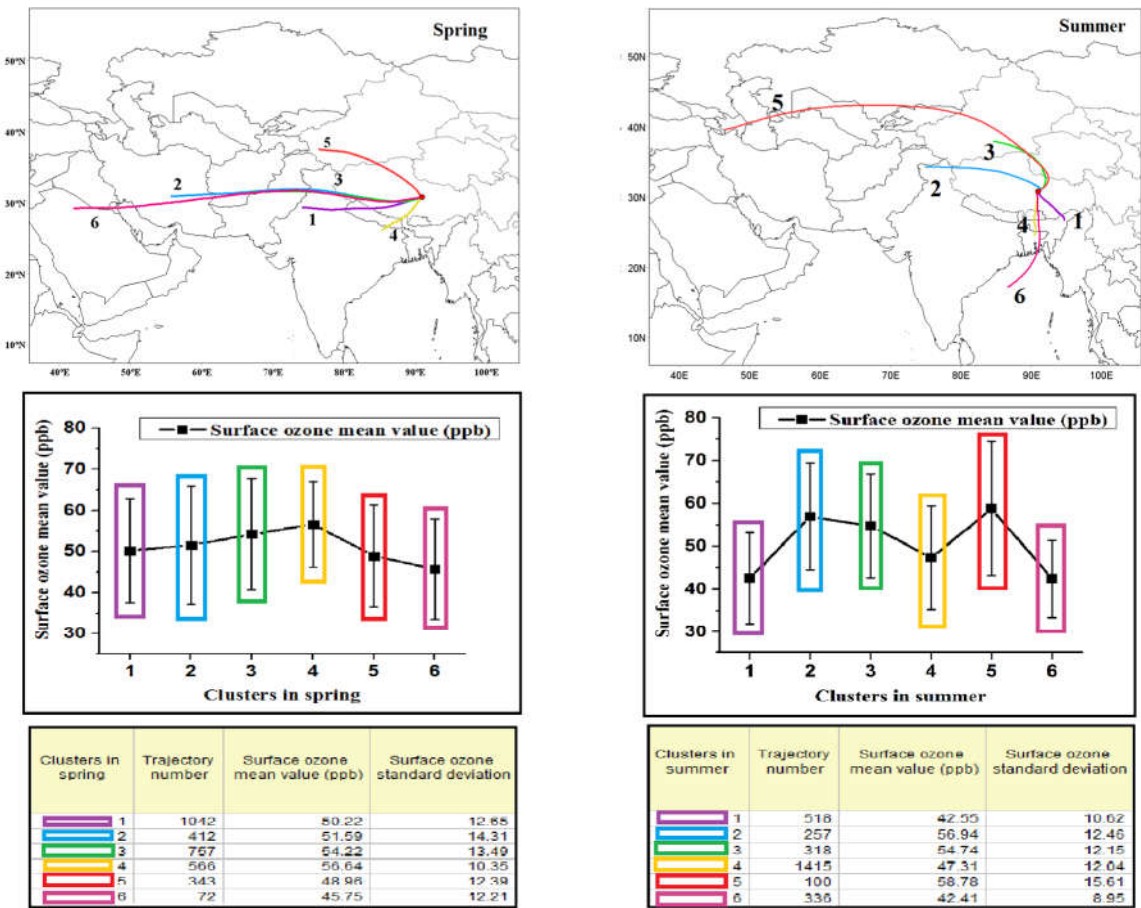

**Fig. 11. Mean trajectory of 6 HYSPLIT clusters arriving at Nam Co Station in the spring and the summer, and the range of surface ozone mixing ratios measured at Nam Co Station by cluster.**

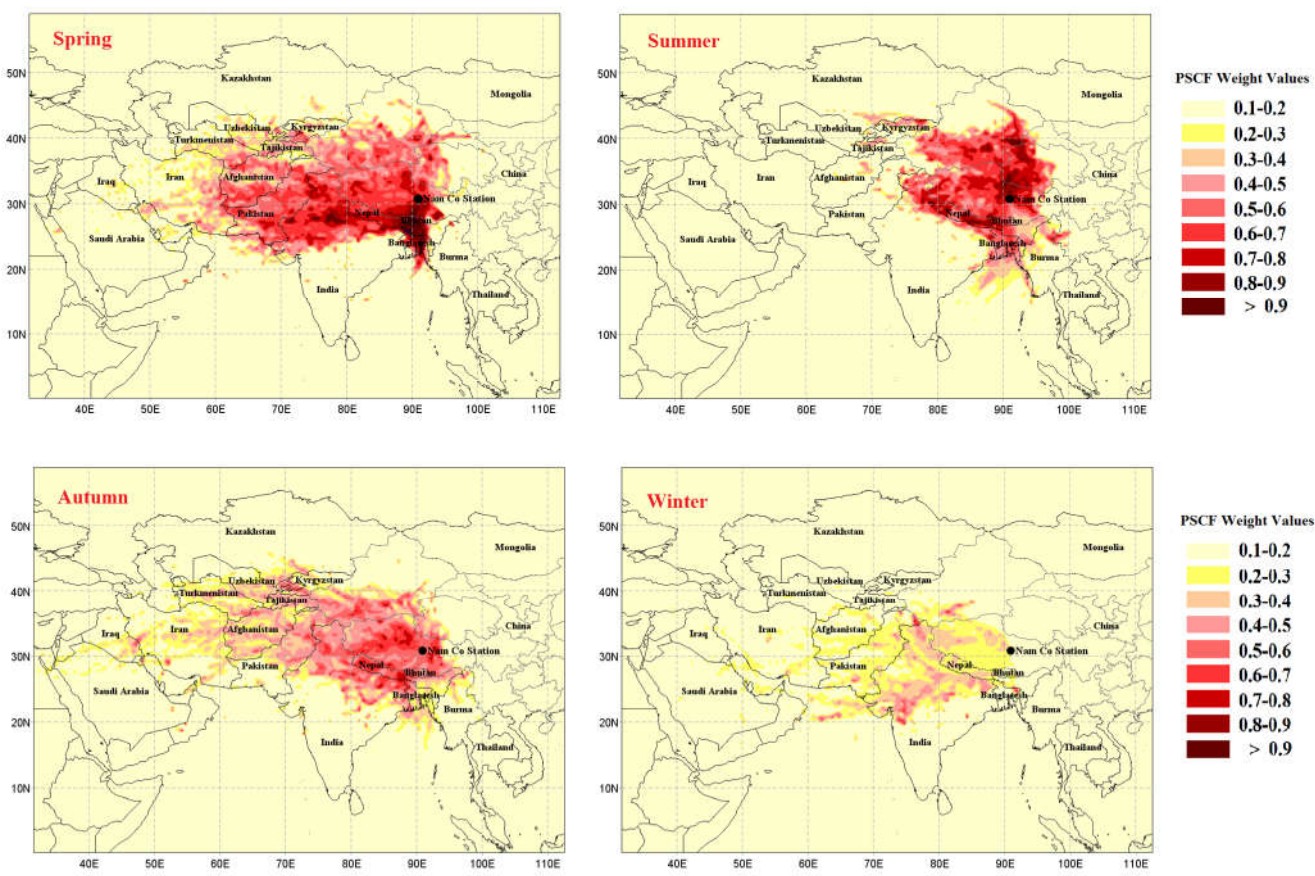

**Fig. 12. Likely source areas of air mass associated with higher surface ozone concentrations at Nam Co Station by season identified using PSCF (Potential Source Contribution Function).**

**Table 1. Statistical summary of surface ozone at Nam Co from 2011 to 2015.**

| Year (valid time during whole year %) | Ozone (ppb) | Range (ppb) |
|---|---|---|
| 2011 (75.25%) | 46.0±12.1 | 10.1-94.7 |
| 2012 (90.30%) | 48.1±11.4 | 14.3-91.5 |
| 2013 (75.90%) | 47.5±12.3 | 15.5-89.7 |
| 2014 (70.05%) | 47.5±10.6 | 14.9-90.8 |
| 2015 (66.21%) | 48.9±12.0 | 17.3-94.7 |
| Total | 47.6±11.6 | 10.1-94.7 |

**Table 2. Multi-linear regression model for hourly ozone (2011-2014) for 3 different models and the contribution of groups of input variables to the variance (%) of the ozone time series.**

**Model A: Log-transformed model of ozone concentration using CAMx stratospheric tracer to identify STE.**
**Model B: Linear-transformed model of ozone concentration using CAMx stratospheric tracer to identify STE.**
**Model C: Linear-transformed model of ozone concentration using ERA-Interim potential vorticity to identify STE.**

| A | | B | | C | |
|---|---|---|---|---|---|
| Number of all hourly data | 27310 | Number of all hourly data | 27310 | Number of all hourly data | 27310 |
| Number of IRLS hourly data | 26005 | Number of IRLS hourly data | 25934 | Number of IRLS hourly data | 25985 |
| r (all hourly data calculated) | 0.77 | r (all hourly data calculated) | 0.75 | r (all hourly data calculated) | 0.75 |
| r (IRLS hourly data calculated) | 0.81 | r (IRLS hourly data calculated) | 0.79 | r (IRLS hourly data calculated) | 0.80 |
| | | | | | |
| CAMx Strat Tracers | 18.2% | CAMx Strat Tracers | 12.5% | PV | 5.8% |
| WRF-FLEXPART Clusters | 6.5% | WRF-FLEXPART Clusters | 6.8% | WRF-FLEXPART Clusters | 6.4% |
| Local Winds | 31.0% | Local Winds | 28.6% | Local Winds | 29.4% |
| Seasonal Signal | 35.3% | Seasonal Signal | 44.2% | Seasonal Signal | 52.1% |
| Diurnal Signal | 7.4% | Diurnal Signal | 6.7% | Diurnal Signal | 5.7% |
| Annual Signal | 1.5% | Annual Signal | 0.7% | Annual Signal | 0.5% |
| WRF PBLH | 0.1% | WRF PBLH | 0.4% | WRF PBLH | 0.2% |
