# Peer review of "Surface ozone at Nam Co in the inland Tibetan Plateau: variation, synthesis comparison and regional representativeness"

_Atmospheric Chemistry and Physics, 2017_

## Referee Comment (RC1) · Anonymous Referee #1 · 24 Apr 2017

**Review**

**Surface ozone at Nam Co (4730 m a.s.l.) in the inland Tibetan Plateau: variation, synthesis comparison and regional representativeness**

Authors: Xiufeng Yin, Shichang Kang, Benjamin de Foy, Zhiyuan Cong, Jiali Luo, Lang Zhang, Yaoming Ma, Guoshuai Zhang, Dipesh Rupakheti, Qianggong Zhang

**Summary of paper**

The Tibetan Plateau is considered as an ideal region for studying processes of the background atmosphere. Sites in the southern, northern, and central regions of the Tibetan Plateau exhibit different patterns of variation in surface ozone. Measurements for the period January 2011 to October 2015 of surface ozone concentrations at Nam Co Station are summarized using mostly monthly averaged values. A large annual cycle was observed with maximum ozone mixing ratios occurring in the spring with minimum ratios occurring during the winter. The authors indicate that Nam Co Station represents a background region, where surface ozone receives negligible local anthropogenic emissions. The authors state that surface ozone at Nam Co Station is mainly dominated by natural processes involving photochemical reactions and potential local vertical mixing. Model results indicate that the study site is affected by the surrounding areas in different seasons and that air masses from the northern Tibetan Plateau lead to increased ozone levels in the summer. The authors believe that in contrast to the surface ozone levels measured at the edges of the Tibetan Plateau, those at Nam Co Station appear to be less affected by stratospheric intrusions and human activities, which makes Nam Co Station representative of vast background areas in the central Tibetan Plateau. By comparing measurements at Nam Co Station with those from other sites in the Tibetan Plateau and beyond, the authors' goal is to expand the understanding of ozone cycles and transport processes over the Tibetan Plateau.

**General Comments**

I would like to see another version of this manuscript after the authors have made their modifications.

A key question I have is to what extent do the authors believe that stratospheric intrusions (not necessarily originating directly above the site) influence the Nam Co station? The reason I am asking this question is that the authors state "In contrast to the surface ozone levels at the edges of the Tibetan Plateau, those at Nam Co Station are less affected by stratospheric intrusions and human activities which makes Nam Co Station representative of vast background areas in the central Tibetan Plateau." I am not sure what the authors are intending to say in this sentence. Does the sentence mean that stratospheric intrusions play an unimportant role at the site in influencing the surface ozone concentrations or do the authors mean that the Nam Co site is influenced by "aged" stratospheric intrusions but to a lesser extent than those intrusions that occur at the southern and northern portions of the Tibetan Plateau? Based on the detailed focus on stratospheric intrusions in the manuscript, I suspect that the authors believe that STE plays an important role at the Nam Co Station in enhancing surface concentrations during specific seasons

but that STE plays *less* of a role when compared to stations located at the southern and northern portions of the Tibetan Plateau. I would appreciate it if the authors would clarify this.

The authors devote a considerable amount of the manuscript to discussing the contribution from stratospheric intrusions during specific periods of the year. Using mostly monthly and annual average surface ozone mixing ratios, the authors report a large annual cycle with maximum ozone mixing ratios occurring in the spring, with minimum ratios occurring during the winter. As noted by the authors, during the spring, Nam Co was affected by aged stratospheric originating over the Himalayas rather than being influenced by transport from fresh stratospheric air masses directly above the station. In spring, the air masses that arrived at Nam Co Station were predominantly from the west and from the south, and the 3-D clusters indicated that the air masses traveled through the Himalayas before reaching Nam Co Station. The authors note that Cristofanelli et al. (2010), Putero et al. (2016) and Chen et al. (2011) found that the frequency of stratospheric intrusions in the Himalayas was high in spring, and slightly lower than during the winter. Škerlak et al. (2014) showed that the seasonal average ozone flux from the stratosphere to the troposphere in the Himalayas was the highest in spring. The authors noted that air masses transported in the spring from the Himalayas led to higher concentrations of surface ozone at Nam Co Station.

For the summer months, the authors note that were more backward trajectories coming from the northern Tibetan Plateau than in other seasons. HYSPLIT backward trajectories arriving at Nam Co Station in the summer were classified into 6 clusters. Clusters which came from the northern Tibetan Plateau had higher mean surface ozone levels than clusters which came from the southern Tibetan Plateau. The authors indicate that the air masses that arrived at Nam Co Station from the northern Tibetan Plateau and northwestern China by horizontal wind transport likely resulted in the higher ozone concentrations at Nam Co Station during the summer. However, Trajectories 2 and 3 during the summertime also contain high ozone concentrations (Fig. 11).

During the summer, according to Škerlak et al. (2014), the northern Tibetan Plateau is the hot spot of stratosphere-to-troposphere ozone flux. Do other trajectories (e.g., 2 and 3) during the summertime also exhibit possible contributions from STE? A further reading of Škerlak et al. (2014) indicates that the hotspot region of the Tibetan Plateau is most likely affected by stratospheric intrusions during the months of DJF, MAM, and JJA (page 926 of Škerlak et al., 2014). Škerlak et al. (2014) indicate that there are **intense** deep STT ozone fluxes over the Tibetan Plateau during MAM and JJA. Škerlak et al. (2014) indicate that the global hotspots, where surface ozone concentrations are most likely influenced by STE, is the Tibetan Plateau in all seasons except for SON (page 934).

As indicated above, the authors mostly used monthly and annual average surface ozone mixing ratios to characterize the ozone concentrations at the Nam Co Station. The use of monthly or annual average concentrations "smoothes" the variability associated with hourly average concentrations. Thus, if one were interested in assessing the magnitude of the ozone concentration enhancements that may be associated with STT events, he or she might wish to focus on the frequency and time of year when high hourly average concentrations occur. Although I am not suggesting that the authors have to perform an additional assessment, I think the authors, using *hourly* average concentrations, have an opportunity to include in their current

manuscript an expanded discussion on the potential importance of aged stratospheric air originating at other locations that is transported to the site. Fig. S1 provides potentially important information about the day-to-day variability of the hourly concentrations. I have reproduced Fig. S1 below. The figure illustrates the variability of the hourly average concentrations for the period from January 2011 until October 2015. As anticipated, the frequency of the highest hourly average concentrations (e.g., 70 ppb to > 90 ppb) occurs during the springtime and early summertime (Fig. S1). This agrees with the authors' observations based on the monthly average concentrations. However, unlike the pattern described based on the monthly averages, high hourly average concentrations are also occurring during the winter and summertime for some of the years. During the months of SON, the frequency of high hourly average concentrations is much lower than those values exhibited during the DJF, MAM, and JJA seasons. Thus, there appears to be different patterns observed when using the monthly average concentration results with those using the hourly average concentration results.

Investigating the pattern for when the highest hourly average concentrations occur, it appears that this pattern is similar to the one described by Škerlak et al. (2014), which indicated stratospheric intrusion hotspots in the Tibetan Plateau during the months of DJF, MAM, and JJA. If the authors wish to, they have the opportunity to expand their discussion in their manuscript to comment on the degree to which the observed enhanced hourly average ozone concentrations may be associated at the Nam Co Station with STE.

[Figure]

Fig. S1. Variation of surface ozone at Nam Co Station from January 2011 to October 2015. Hourly mean mixing ratios of surface ozone are in blue dots; monthly mean mixing ratios of surface ozone are in black dots; average mixing ratio of surface ozone during whole measurement period in red dash line.

**Specific Line-by-Line Comments**

1.  Title: I would suggest that the title be slightly modified as follows: Surface ozone at Nam Co in the inland Tibetan Plateau: variation, synthesis comparison and regional representativeness.

2. Lines 24-25: The authors state "Model results indicate that the study site is affected by the surrounding areas in different seasons and that air masses from the northern Tibetan Plateau lead to increased ozone levels in the summer." I think the authors are not necessarily indicating that there is an increase during the summer at Nam Co due to air masses from the northern Tibetan Plateau but that the air masses from the northern Tibetan Plateau *contribute* to the enhancement of ozone levels measured at the site. The word "increase" gives the impression that relative to the spring, the summer monthly averages are higher. The monthly average levels at the site are lower than those observed during the spring and therefore, I am suggesting a slight change in the wording.

3. Lines 34-35: I would suggest references that represent comprehensive summaries of human health and vegetation effects, such as LRTAP Convention (2015), REVIHAAP (2013), and US EPA (2013).

4. Line 45-46: The sentence: "In this situation, background sites can represent areas with surface ozone concentrations that are under the control of largely uniform synoptic systems and are minimally affected by local anthropogenic sources." What does "in this situation" refer to?

5. Lines 141-143: The sentence "In cells with high PSCF values are associated with the arrival of air parcels at the receptor site that have pollutant mixing ratios that exceed the criterion value" does not appear to be complete. Should the sentence start with "Cells with high PSCF…"?

6. Line 155: The sentence states "The mean surface ozone mixing ratio at Nam Co Station during the entire observational period was $47.6 \pm 11.6$ ppb…." I am not suggesting any change in this sentence but I do want to point out that the authors on Lines 33 and 34 state that "High levels of surface ozone are currently a major environmental concern because of the harm ozone poses to health and vegetation." This is a correct statement. However, researchers who assess human health and vegetation effects focus on the occurrence of high, as well as mid-level hourly average concentrations, and normally do not focus on high annual average concentrations. Annual, seasonal, or monthly average ozone concentrations are not necessarily the best metrics to use when assessing either human health or vegetation effects. While monthly and annual average concentrations are used for assessing the performance of global modeling results, these metrics are not necessarily relevant for assessing human health and vegetation effects.

7. Page 156: Table 1 indicates that the data capture was as follows: 2011 (75.25%), 2012 (90.30%), 2013 (75.90%), 2014 (70.05%), and 2015 (66.21%). Was the 66.21% data capture observed in 2015 related to the entire 12 months or was this value the data capture for the period January – October 2015?

8. Lines 182-183: The authors state "The transition between high levels during the daytime and low levels during the nighttime was fast." I would appreciate it if the authors could please explain why the transition was fast.

9. Lines 186-187: The authors state "Relatively large diurnal amplitudes were observed in spring, with much smaller diurnal amplitudes observed during summer, autumn and winter." Can the authors offer an explanation for this observation? Could this observation be associated with STE making it to the ground during the spring more frequently than during the other seasons?

10. Lines 194-196: The authors state "[35]S results (Lin et al., 2016) also support this result by showing that in the spring; Nam Co was affected by aged stratospheric air originating

over the Himalayas rather than being affected by transport from fresh stratospheric air masses directly above Nam Co Station." Should the ";" be placed with a "," to make a complete sentence?

11. Lines 188-200: The authors state "A multiple linear regression model was used to quantify the contributions of various factors (including temperature, clear sky solar radiation, potential vorticity, wind speed, humidity, annual cycle, interannual variation and WRF-FLEXPART trajectory clusters) to the measured maximum daily 8-hour average surface ozone." If in the authors' multiple linear regression model the variables (i.e., temperature, clear sky solar radiation, potential vorticity, wind speed, humidity, annual cycle, interannual variation and WRF-FLEXPART trajectory clusters) were not independent, what would be the effect on the outcome of the results using the model?

12. Lines 209-211: The authors state" Specific humidity was the second largest contributor (20%; Table 2) with a negative coefficient indicating that higher surface ozone was associated with drier conditions possibly due to transport of continental air masses; or impacts from air masses aloft." If the Nam Co Station were influenced by "aged" stratospheric intrusions, would the lower humidity still be associated with the "aged" transported air from the stratosphere originating over the Himalayas after several days? Perhaps a short comment in the manuscript might be in order.

13. Lines 212-214: The authors state "The negative coefficient indicates that air masses transported from the south to Nam Co were associated with lower surface ozone. For the whole measurements period, it seems that transport of surface ozone is not the main influencing factor to the daily surface ozone variations in the multiple linear regression model." However, in Lines 287-290, the authors indicate that "Backward trajectories and PSCF were utilized to identify the source of surface ozone at Nam Co Station and to assess the regional representativity of surface ozone at Nam Co. In spring, the air masses that arrived at Nam Co Station were predominantly from the west and from the south, and the 3-D clusters indicated that the air masses traveled through the Himalayas before reaching Nam Co Station (Fig. 10)." If the air masses traveled through the Himalayas during the spring before reaching the Nam Co Station, at times would not the air masses represent "aged" stratospheric intrusions and wouldn't these air masses influence the daily surface ozone variation? Is there a difference in the conclusions reached using the multiple linear regression model versus the back trajectory and the PSCF analyses? Perhaps I am missing something here.

14. Lines 256-258: The authors state "This type has a plateau of high surface ozone in spring and summer and a minimum in winter. Sites of this type occur in regions with strong ozone precursor emissions in the summer (such as the central European continent) or in regions where stratospheric intrusion occurs frequently in summer." Could the authors please provide examples for specific regions of the world where stratospheric intrusions frequently occur during the summer. Perhaps the results from Škerlak et al. (2014) might be a good source.

15. Lines 271-273: The authors state "Sites in the central Tibetan Plateau including Nam Co Station showed maximum ozone during late spring-early summer and relatively low levels in the remainder of year (Fig. 9B), corresponding to the Spring-maximum type. Compared with the surface ozone levels at Nam Co Station, those at Lhasa and Dangxiong were much lower." This conclusion is based upon the use of monthly average

concentrations. Is there any indication that the use of the frequency of high hourly average concentrations might provide a different pattern?

16. Lines 313-314: The authors state "The atmospheric environment of the Tibetan Plateau and its relationship to regional and global change are of universal concern due to the rapid responses and feedbacks specific to the "Third Pole". I would appreciate it if the authors would please expand on this sentence to explain what they mean.

17. Line 324-327: The authors state "Waliguan, in the northern Tibetan Plateau, is occasionally influenced by regional polluted air masses (Zhu et al., 2004; Xue et al., 2011; Zhang et al., 2011). Its mountainous landform facilitates mountain-valley breezes and may sometimes pump up local anthropogenic emissions especially during the winter (Xue et al., 2011)." I was under the impression that local anthropogenic sources are small near Mt. Waliguan. Mt. Waliguan is far from major cities, such as Xining (90 km) and Lanzhou (260 km) in the eastern sector. I would appreciate it if the authors would further elaborate concerning the enhancement at Mt. Waliguan from local anthropogenic emissions.

18. Lines 332-335: The authors state "During the summer, surface ozone concentrations at Nam Co Station are higher than the northern hemisphere average, which suggests that there are impacts of long-range transport. Nam Co is less influenced by stratospheric intrusions than NCOP on the slopes of Mount Everest, and it is minimally influenced by local anthropogenic emission as evidenced by the constant long-term variation of surface ozone and consistent diurnal variation regardless of season, as discussed above." What is the influence of stratospheric intrusions on Nam Co during the summer? Škerlak et al. (2014) appear to indicate that it is important during the summer. If the surface ozone concentrations during the summer at Nam Co Station are higher than the northern hemisphere average, could the suggested long-range transport be associated with "aged" air masses from the stratosphere that are being transported to the site? I think it would help the reader to clarify what the authors mean by " there are impacts of long-range transport."

19. Line 340: The summary needs to be expanded. It is very minimal at this time.

20. Lines 348-349: The authors state " Synthesis comparison indicated that Nam Co is less influenced by stratospheric intrusions and anthropogenic disturbances than sites along the rim of the Tibetan Plateau." I would appreciate it if the authors could please clarify this sentence. Should the sentence read "While the Nam Co Station is less influenced by stratospheric intrusions and anthropogenic disturbances than sites along the rim of the Tibetan Plateau, the site does exhibit during specific months large contributions associated with transported "aged" air masses associated with stratospheric intrusions." I do not wish to impose this interpretation on the authors, but rather elicit from them if this is what they are attempting to say. If not, could they please provide a concise sentence that clearly describes their conclusion on the importance of stratospheric intrusions associated with long-range transport in enhancing the surface ozone concentrations at Nam Co. I think this would help the reader.

21. Supplement: Fig. S1. I would suggest improving the readability of the title of the x-axis (Year-Month-Day-Hour). It seems to not be clear on my copy. Does the first symbol in the time series identified as 2011-01-01 in Fig. S1 represent the January average or just the 2011-01-01 point? I am not sure what the first dot represents. The meaning of the first dot is confusing.

**References**

LRTAP Convention, 2015.  Chapter III:  Mapping Critical levels for Vegetation, of the Manual on Methodologies and Criteria for Modelling and Mapping Critical Loads and Levels and Air Pollution Effects, Risks and Trends.  Available at: http://icpvegetation.ceh.ac.uk/publications/documents/Ch3-MapMan-2016-05-03_vf.pdf.

REVIHAAP. 2013. Review of evidence on health aspects of air pollution – REVIHAAP Project technical report. World Health Organization (WHO) Regional Office for Europe. Bonn. Available at: http://www.euro.who.int/__data/assets/pdf_file/0004/193108/REVIHAAP-Final-technical-report-final-version.pdf.

US EPA. 2013. Integrated Science Assessment of Ozone and Related Photochemical Oxidants (Final Report). EPA/600/R-10/076F, available at: http://www.epa.gov/ttn/naaqs/standards/ozone/s_o3_2008_isa.html.

---

## Referee Comment (RC2) · Anonymous Referee #2 · 30 Apr 2017

GENERAL COMMENTS This work by Yin et al. presents an overview of about 5 years of continuous near-surface ozone observations at the Nam Co station which is located in the central Tibetan Plateau. The scope of the paper is rather ambitious: to characterize the typical variability of near-surface O3 at this measurement site, to compare it with other sites in the Tibetan Plateau (and beyond) and to demonstrate that this site is representative for the whole Tibetan Plateau. The presented data-set is of great interest (and I suggest to share it in the framework of international initiatives like WMO/GAW or TOAR/JOIN). However, the paper is a little bit confusing and for a great part relies too much in other studies, resembling more a "review" than a research paper. Moreover, some important conclusions were based too much on qualitative assertions. As

an instance, in my opinion, the authors failed in demonstrating that: "The unique geographical characteristics make Nam Co Station more representative of the baseline of surface ozone in the extensive inland of Tibetan Plateau than other existing monitoring sites", as they report in the Summary. More analyses/comparisons are needed to assess this point! My impression is that the authors mixed together several different analyses without a well-defined scientific track. For instance, at least two different model (FLEXPART- WRF and HYSPLIT) were used with the same aim (characterize O3 variability as function of air-mass transport) but without any critical comparison or integration. The fact that O3 is positively correlated with some meteorological parameters is not of great scientific novelty and (the most important point) I suspect that the linear model results were significantly affected/biased by the use of daily average values (at least for ozone). The discussion about the role of STE is simply based on a subjective (mainly visual) analysis of O3 variability with stratospheric "tracers" (not specific analyses or tool have been used). For these reasons, I suggest to resubmit the paper after than some essential modifications have been made. In the following I provide some suggestion to help authors towards this aim.

SPECIFIC COMMENTS Line 43-45: I think that this sentence is meaningless.

Line 55: this is wrong. At NCO-P the highest contributions from STE is in WINTER. This is clearly stated by Cristofanelli et al. ACP (2010) and Putero et al. ACP (2016). The pre-monsoon (spring) O3 peaks was strongly affected by the transport of pollution from the lower troposphere (Himalayas foothills and Indo-Gangetic Plains). See e.g. Putero et al. Atmospheric Pollution (2013); Bonasoni et al., ACP (2010).

Line 69-70: this sentence is too generic. Specify what kind of ozone-related climatic and environmental effect can be assessed and by which methodology.

Line 84: remove the capital letter from "The"

SECTION 2 Line 95: how did you evaluate change in sensitivity? By which frequency the analyser was calibrated? The calibrator 49iPS was calibrated against which refer-
ence instrument?

Section 2.4: Which is the time resolution of the inputs to the MLR Model (hourly, daily)? How did you consider the FLEXPART trajectory cluster in the regression analysis? Why did you normalize the input parameters? Why did you exclude outliers? The last three sentence are rather obscure to me (from line 126). Please, provide a clear step-by-step description of the methodology. By only considering the maximum 8-hour average ozone concentration, you discharge all the information about variability at hourly scale (which is rather important)...and this is the reason why you find out a great role of radiation! At least, this must be clearly stated in the revised manuscript.

Line 100: I think that I would be better and more useful to refer the measurements to the "local time" instead of "Beijing time".

Line 110: please provide more info about the HYSPLIT simulation set-up. Which meteorological gridded data-set has been used to calculate back-trajectories (GFS)? By which time resolution did you calculate back-trajectories (Once a day? Every hour?)? How did you take into account uncertainties due to the complex topography surrounding the Plateau? Also provide more info about the cluster methodology and provide a description of the algorithm. Provide web access indication to the TRAJplot software. I think that both NOAA (for providing GDAS and HYSPLIT) and TRAJPlot developers must be acknowledged in this paper. I guess WRF-FLEXPART is much more accurate in reproducing air-mass origin and transport to Nam CO. However, please provide more technical details about the model set-up. It is not clear to me which is the reason to use HYSPLIT when WRF-FLEXPART is available. Please, explain. Did you compare the results obtained with FLEXPART and HYSPLIT? Line 117: "Six clusters were found...". Does this sentence refer to HYSPLIT or FLEXPART? Not clear ....

Section 2.5: What model did you use for this analysis (HYSPLIT or WRF-FLEXPART)? Did you consider some altitude/pressure level thresholds of back-trajectory points to allow the PSCF calculation? If not, hardly you can relate the obtained results with
surface emissions....The W values are a key parameter for the interpretation of the obtained results. How did you define them? Did you perform a sensitivity study by changing the weighting factor?

**SECTION 3**

Line 158: please attribute the origin of these anomalous events

Line 161: for the period 2006 – 2011 Putero et al (2013) found an average O3 of 48.7 ppb at NCOP, while Cristofanelli et al. (2010) over two year investigation pointed out an average value of 49 ppb. Thus, I would say that average value at Nam Co and NCO-P are comparable. Please correct.

Line 162: different factors influence background O3 levels, i.e. altitudes, latitude, site classification (mountain, coastal, marine). The authors must better address this comparison taking into account all these factors.

Line 166: So, did you consider months with at least a 60% data coverage. Please specify this point rather than indicating the number of hours.

Section 3.3: would remove Fig 3 and leave only Fig 4 (where diurnal variability are also more evident). However, for each hourly average you must add an error bar denoting the 95% confidence level of the mean average value. At this point, a description of typical local wind variability (wind speed and direction) must be added to evaluate possible influence of diurnal wind breeze on O3 variability.

Section 4.1: This analysis of stratospheric intrusion is too raw. I would like to see a more specific investigation (see e.g. Cristofanelli et al., 2010; Putero et al., 2016; Trickl et al., ACP, 2010). The authors only described in a very qualitative and oversimplified way (basically by "visual" inspection) the time series of stratospheric air markers (any statistical analysis or selection methodology is applied). Moreover, the assumption that stratospheric intrusion can be directly related to the daily maximum of ozone is wrong. Due to mixing and dilution processes, stratospheric air-masses are often characterized
by O3 values which are even lower than those due to photochemistry. Moreover, these events are often characterized by short time duration (even lower than 1 day), thus simply comparing time series of stratospheric tracers with a daily time resolution can mask the real influence of STE. The final sentence: "Nam Co was affected by aged stratospheric air originating over the Himalayas rather than being affected by transport from fresh stratospheric air masses directly above Nam Co Station ", it's not clear to me. Quantify "aged".

Section 4.2: I suggest to perform this analysis also on a seasonal basis. Since most of the used predictors are characterized by significant seasonal cycles, this would provide more hints about the role of single factors in driving O3 variability. Figure S4 it's not clear at all. What is the scale reported on the right bar?

Line 210: "impacts from air masses aloft". Be more specific!

Line 213: "why these air-masses are depleted in O3". I suspect simply because they were related to southern air-mass advection during the monsoon. Please provide a description of the seasonal frequency of occurrence of air-mass transport patterns reported by Fig. S4. You stated that: "For the whole measurements period, it seems that transport of surface ozone is not the main influencing factor to the daily surface ozone variations in the multiple linear regression model". I'm not convinced. As showed by other works (see Di Carlo, JGR, 2007). The role of dynamic is important at hourly time-scale. By analysing data as daily averages you ruled out by default these contributions! By comparing the time series of O3 observations with the regression model (Fig. 5), it is rather clear than the model was not able to reproduce the spring peak. To my opinion, this is a clear hint toward an important contribution of transport and dynamics.

Section 4.3: If data analyzed are daily averages, the correlation coefficient here provided (R: 0.77) does not describe the "local" (in-situ) role of photochemistry. This must be described by analysing the hourly data-set as you did for wind speed and PBLH. Which is the correlation coefficient between hourly ozone and hourly SWD? As sug-
gested by Fig.7, the higher correlation with wind speed and PNLH suggest that dynamics is the most important factor explaining diurnal O3 variability. I suggest to apply the linear correlation model both for daily and hourly values and to comment differences in the results.

Line 245: "the background ozone at the site": this is contradictory, the background cannot be local!

SECTION 5. It is not clear why in Figure 8 you reported "normalized O3" for NCOP. Please explain what kind of normalization was applied. At Xianggelila, Ma et a. (2014) reported that at diurnal scale O3 was strongly correlated with wind speed (as occurred also at Nam CO) and that " the transport and deposition will be the key factors influencing the diurnal variations of surface O3 at Xianggelila, a remote and clean site, rather than local photochemical processes". Also at Dangxiong, Lin et al. (2015), suggested that the correlation with high wind speed and O3 during the afternoon pointed out the important role of transport in affecting O3 more than photochemistry. I would bet that the same is true for Nam CO.

Section 5.2: In my opinion the classification of the seasonal ozone regimes I-III is oversimplified (see the nice work by Tarasova e al., 2007, ACP). I suggest the authors to skip this first part (line 243-263) and discuss the O3 variability at the Tibetan sites as a function of the characterization provided by Tarasova et al. 2007. Line 256: please provide adequate references. Line 260: I think that this sentence only refers to summer season. Please, specify. Line 275: The possible impact of NO titration to the appearance of lower ozone levels at the the Tibetan sites should be better assessed/showed. For instance, you can report diurnal variability as a function of different seasons for these sites. NCO-P is not located over the Tibetan Plateau but at the southern ridge of Himalayas. Please correct.

Line 290: Figure 10 is hard to read and clusters look very similar each other's (except than for those related to southerly circulation). What kind of cluster algorithm was
used? It looks that a large part of the information carried by the back-trajectories was missed by this clustering. Nevertheless, in agreement with this analysis, during Spring only a fraction (about 18%) of back-trajectories crossed the Himalayas. This must be clearly stated.

Line 292: Actually, Skerlak et al. (2014) reports a maximum of deep STT over the Tibetan Plateau and not only over Himalayas! In my opinion, your conclusion that O3 is higher at NCO-P due to a larger contribution from stratosphere is wrong. Looking at your Fig. 9, it looks that O3 values at NCO-P and Nam Co were well comparable on March and May. O3 was higher at NCO-P in April, but (as I reported below) the contribution of polluted air-masses in driving O3 variability at NCO-P during this season cannot be neglected!

Line 294: I think that at this point the transport of polluted air-masses from Himalaya foothills and IGP to high Himalayas must be considered (see Bonasoni et al., 2010; Putero et al., 2013; Luthi et al., 2015)! This contributed to the appearance of the premonsoon maximum at NCOP and possibly the cross-Himalaya transport can also affect Tibetan Plateau.

Line 296: which cluster was associated to the northern TP? It is not possible to recognize it from Figure 10 (please increase the fonts used for legend!)

Line 297: I read carefully Skerlak et al (2014) but I was not able to found any reference to the higher stratospheric flux over the northern Plateau in respect to the southern Plateau in autumn. Indeed, looking at their Fig. 6, this not looks to be the case.

Line 301-304: Is this confirmed also by WRF-FLEXPART clustering?

Line 305-312: were these results confirmed by the HYSPLIT clustering? I expect that WRF-FLEXPART could have much more skill than HYSPLIT (based on global meteorological fields with coarse spatial resolution)in analysing spatial "contributions" for elevated O3 values at Nam CO. However, you must attribute the seasonal variability
of the "contributions" you found by WRF-FLEXPART (by what kind of emissions, precursors are emitted over each identified regions?). Moreover, you should discuss and quantify the uncertainties related with this analysis. Also some details were missed: as an instance, for the seasonal analysis you used as O3 threshold values ,the seasonal averages or the whole period average? What happens if different threshold were applied (e.g. 75th or 90th percentiles of ozone distribution)? Probabilities higher than 1.0 were reported in the legends: I think this is inconsistent...please check!

Section 5.4: This section about representativeness of Nam CO is mostly based on an intuitive/subjective approach and from review of previous works. Even if I'm personally convinced that Nam Co is an interesting background site, the authors must perform much work if their want to unambiguously assess the spatial representativeness of the station. See for instance Henne et al., ACP, 10, 3561–3581, 2010. I do not think that a " consistent diurnal variability of ozone regardless of season" can be used as proof to claim the large spatial representativeness of the station. Moreover, it seems that the authors do not consider STE as part of the "global" background ozone: from my point of view, this is completely wrong. If not specific analyses are accrued out, I strongly recommend to eliminate this section and limit some lines of comment in the summary Section.

Line 332: please quantify the spatial scale of this "long-range" contribution

SUMMARY Line 343: "Nam Co represents a wide background region in the Tibetan Plateau". In my opinion this need more quantification efforts, since this sentence is too generic/qualitative.

Line 349: "Synthesis comparison...". The authors did not convince me about the small impact of STE.

ACKWNOLEDGMENTS You must acknowledge NOAA for providing HYSPLIT model and GFS meteorological files. I suppose that also the TrajPlot developers must be acknowledged!

**ACPD**

---

## Author Comment (AC1) · 4 Jul 2017

**Response to referee comments**

We would like to thank the referees and editor for the interest in our work and the helpful comments and suggestions to improve our manuscript. We have carefully considered all comments and the replies are listed below. The changes have been marked in the text using blue color.

**Review (Anonymous Referee #1)**

Surface ozone at Nam Co (4730 m a.s.l.) in the inland Tibetan Plateau: variation, synthesis comparison and regional representativeness

Authors: Xiufeng Yin, Shichang Kang, Benjamin de Foy, Zhiyuan Cong, Jiali Luo, Lang Zhang, Yaoming Ma, Guoshuai Zhang, Dipesh Rupakheti, Qianggong Zhang

**Summary of paper**

The Tibetan Plateau is considered as an ideal region for studying processes of the background atmosphere. Sites in the southern, northern, and central regions of the Tibetan Plateau exhibit different patterns of variation in surface ozone. Measurements for the period January 2011 to October 2015 of surface ozone concentrations at Nam Co Station are summarized using mostly monthly averaged values. A large annual cycle was observed with maximum ozone mixing ratios occurring in the spring with minimum ratios occurring during the winter. The authors indicate that Nam Co Station represents a background region, where surface ozone receives negligible local anthropogenic emissions. The authors state that surface ozone at Nam Co Station is mainly dominated by natural processes involving photochemical reactions and potential local vertical mixing. Model results indicate that the study site is affected by the surrounding areas in different seasons and that air masses from the northern Tibetan Plateau lead to increased ozone levels in the summer. The authors believe that in contrast to the surface ozone levels measured at the edges of the Tibetan Plateau, those at Nam Co Station appear to be less affected by stratospheric intrusions and human activities, which makes Nam Co Station representative of vast background areas in the central Tibetan Plateau. By comparing measurements at Nam Co Station with those from other sites in the Tibetan Plateau and beyond, the authors' goal is to expand the understanding of ozone cycles and transport processes over the Tibetan Plateau.

**General Comments**

I would like to see another version of this manuscript after the authors have made their modifications.

A key question I have is to what extent do the authors believe that stratospheric intrusions (not necessarily originating directly above the site) influence the Nam Co station? The reason I am asking this question is that the authors state "In contrast to the surface ozone levels at the edges of the Tibetan Plateau, those at Nam Co Station are less affected by stratospheric intrusions and human activities which makes Nam Co Station representative of vast background areas in the central Tibetan Plateau." I am not sure what the authors are intending to say in this sentence. Does the sentence mean that stratospheric intrusions play an unimportant role at the site in influencing the surface ozone concentrations or do the authors mean that the Nam Co site is influenced by "aged" stratospheric intrusions but to a lesser extent than those intrusions that occur at the southern and northern portions of the Tibetan Plateau? Based on the detailed focus on stratospheric intrusions in the manuscript, I suspect that the authors believe that STE plays an important role at the Nam Co Station in enhancing surface concentrations during specific seasons but that STE plays *less* of a role when compared to stations located at the southern and northern portions of the Tibetan Plateau. I would appreciate it if the authors would clarify this.

**Response:** Thank you for pointing out this critical issue. We believe that stratosphere-troposphere exchange (STE) plays an important role on surface ozone at Nam Co Station, but one that is different from the STE that happens in the southern Tibetan Plateau (in the winter and the spring) and the northern Tibetan Plateau (in the summer), Nam Co Station was affected by STE indirectly most of the time. The air masses in high ozone level can be transported to Nam Co Station horizontally after the STE in the southern Tibetan Plateau and the northern Tibetan Plateau in different seasons.

As a result of the reviews, we have refined the analysis of potential vorticity as a tracer for stratospheric air and we have also expanded the regression analysis to include tracers for stratospheric ozone transport using an air quality model. In the ACPD manuscript, we had used Potential Vorticity near the surface (500 hPa) to test for stratospheric incursions. However, this did not lead to a clear signal in the regression analysis. Based on new research, we have now found that if we use PVU at the 350 hPa level we detect an influence on the ozone time series. If we use PVU at 350 hPa above the Himalayas then this signal is even clearer. The description of the regression analysis has been expanded and the

results updated accordingly.

An even better match for stratospheric incursions was obtained when we used ERA-Interim ozone concentrations aloft as boundary and initial conditions for the CAMx air quality model. Chemistry was turned off to obtain a passive tracer of stratospheric air at the measurement site. This gave a signal in the regression analysis that is even stronger than the new PVU analysis. The text was expanded and the results updated in the manuscript as follows (lines 124 – 131):

"A tracer for stratospheric ozone incursions at the measurement site was obtained using the CAMx (Comprehensive Air-quality Model with eXtensions) v6.30 model (Ramboll Environ, 2016). The model initial and boundary conditions were obtained from ERA-Interim ozone fields, retaining only concentrations above 80 ppb and higher than 400 hPa. CAMx simulations were performed using the WRF medium and fine domains (domains 2 and 3) in nested mode for the full 4 year time series. In order to serve as a tracer for direct transport, there was no chemistry in the model and ozone was treated as a passive tracer. The resulting time series of the tracer concentration at the measurement site was used as input in the multi-linear regression model. This is similar to the procedure described in de Foy et al. (2014) to estimate the impact of the free troposphere on surface reactive mercury concentrations.".

Fig. 4 and table 2 were added to explain the new analysis. The model suggested that up to 20% of the ozone variability was due to stratospheric incursions. Meridional cross-sections over Nam Co Station (Fig. 5) illustrated the position of downward transport of stratospheric ozone in different seasons.

The authors devote a considerable amount of the manuscript to discussing the contribution from stratospheric intrusions during specific periods of the year. Using mostly monthly and annual average surface ozone mixing ratios, the authors report a large annual cycle with maximum ozone mixing ratios occurring in the spring, with minimum ratios occurring during the winter. As noted by the authors, during the spring, Nam Co was affected by aged stratospheric originating over the Himalayas rather than being influenced by transport from fresh stratospheric air masses directly above the station. In spring, the air masses that arrived at Nam Co Station were predominantly from the west and from the south, and the 3-D clusters indicated that the air masses traveled through the Himalayas before reaching Nam Co Station. The authors note that Cristofanelli et al. (2010), Putero et al. (2016) and Chen et al. (2011) found that the frequency of stratospheric intrusions in the Himalayas was high in spring, and slightly lower than during

the winter. Škerlak et al. (2014) showed that the seasonal average ozone flux from the stratosphere to the troposphere in the Himalayas was the highest in spring. The authors noted that air masses transported in the spring from the Himalayas led to higher concentrations of surface ozone at Nam Co Station.

For the summer months, the authors note that were more backward trajectories coming from the northern Tibetan Plateau than in other seasons. HYSPLIT backward trajectories arriving at Nam Co Station in the summer were classified into 6 clusters. Clusters which came from the northern Tibetan Plateau had higher mean surface ozone levels than clusters which came from the southern Tibetan Plateau. The authors indicate that the air masses that arrived at Nam Co Station from the northern Tibetan Plateau and northwestern China by horizontal wind transport likely resulted in the higher ozone concentrations at Nam Co Station during the summer. However, Trajectories 2 and 3 during the summertime also contain high ozone concentrations (Fig. 11).

During the summer, according to Škerlak et al. (2014), the northern Tibetan Plateau is the hot spot of stratosphere-to-troposphere ozone flux. Do other trajectories (e.g., 2 and 3) during the summertime also exhibit possible contributions from STE? A further reading of Škerlak et al. (2014) indicates that the hotspot region of the Tibetan Plateau is most likely affected by stratospheric intrusions during the months of DJF, MAM, and JJA (page 926 of Škerlak et al., 2014). Škerlak et al. (2014) indicate that there are **intense** deep STT ozone fluxes over the Tibetan Plateau during MAM and JJA. Škerlak et al. (2014) indicate that the global hotspots, where surface ozone concentrations are most likely influenced by STE, is the Tibetan Plateau in all seasons except for SON (page 934).

**Response:** Thank you for your comments.

As noted by Škerlak et al. (2014), surface ozone in Tibetan Plateau (considered as a whole) was most likely influenced by STE in all seasons except for autumn (SON) (page 934 in Škerlak et al., 2014). Nevertheless, when we look into different parts of Tibetan Plateau and even northwestern China, STE was not occurred synchronously. The peak of stratosphere to the troposphere ozone flux was found over the Himalayas and the southern side of the Tibetan Plateau in spring (MAM) (page 926 in Škerlak et al., 2014); while the stratosphere to the troposphere ozone flux occurred in the northern Tibetan Plateau and northwestern China is much higher than those in the southern Tibetan Plateau in summer (JJA) (Fig. 16 and page 926 in Škerlak et al., 2014).

To facilitate the understanding of STE over the Tibetan Plateau, we have added meridional cross-sections over Nam Co Station (Fig. 5) to indicate the position (altitude and longitude) of the strongest STE in the meridional cross-section (over Nam Co Station) in different months. We also added related discussion on the meridional cross-sections (lines 272 - 298):

"In order to visualize the transport of ozone from the stratosphere to the troposphere, we analyzed the upper troposphere and lower stratosphere structures of the meridional cross-section of monthly mean ERA-Interim data above Nam Co Station (Fig. 5). In the spring (Mar, Apr and May), the dynamical tropopause (identified by the isolines of 1 and 2 potential vorticity unit) exhibited a folded structure over the Tibetan Plateau. This tropopause folding can lead to a downward transport of ozone from the stratosphere to the troposphere. Tropopause folding happened in the southern Tibetan Plateau and close to Nam Co Station in the spring. Cosmogenic $^{35}$S results (Lin et al., 2016) also indicated that in the spring, Nam Co was affected by aged stratospheric air originating over the Himalayas rather than being affected by transport from fresh stratospheric air masses directly above Nam Co Station. The larger diurnal amplitude of surface ozone in the spring than other seasons (Fig. 3, mentioned in section 3.3) may be related to four factors: (1) position of STE hot spot; (2) frequency of STE; (3) PBLH at Nam Co Station and (4) solar radiation at Nam Co Station. In the spring, plots of tropopause folding suggest that STE mostly happens in the southern Tibetan Plateau which is close to Nam Co Station and that STE even happens right above Nam Co Station. Furthermore, PBLH at Nam Co Station was higher in the spring than during the rest of the year. The higher PBLH in the spring facilitated the impact of downward transport from the stratosphere to Nam Co Station. The spring also has more intense solar radiation than the summer because the Monsoon leads to increased cloudiness in the summer. The Pearson's correlation coefficient between monthly SWD and surface ozone was ~0.93 in 2012 (2012 was selected because it had a more complete dataset than the other years) (Fig. 6) indicating that monthly surface ozone variability at Nam Co Station was associated with solar radiation. This was expected as increased solar radiation promotes the photochemical production of surface ozone in the spring, which is similar to the mechanism at other background sites (Monks 2000). Consequently, more photochemical production of ozone is expected in the spring. In the summer (Jun, Jul and Aug), the jet core moved to the northern Tibetan Plateau and tropopause folding was relatively farther from Nam Co Station than those in the spring. Consequently, there was a smaller impact of stratospheric air at Nam Co Station. With tropopause

folding further north in the summer, the air masses from the northern Tibetan Plateau may contribute more to the surface ozone levels at Nam Co Station than the air masses from the southern Tibetan Plateau. In the autumn (Sep, Oct and Nov) and the winter (Der, Jan and Feb), the heights of folding were higher than those in the spring and the summer; and the PBLHs in the autumn and the winter were much lower than those in the spring and the summer. Furthermore, SWD in the autumn and the winter were weaker than those in the spring and the summer. These factors contributed to the relatively low level of surface ozone at Nam Co Station in the autumn and the winter".

As indicated above, the authors mostly used monthly and annual average surface ozone mixing ratios to characterize the ozone concentrations at the Nam Co Station. The use of monthly or annual average concentrations "smoothes" the variability associated with hourly average concentrations. Thus, if one were interested in assessing the magnitude of the ozone concentration enhancements that may be associated with STT events, he or she might wish to focus on the frequency and time of year when high hourly average concentrations occur. Although I am not suggesting that the authors have to perform an additional assessment, I think the authors, using *hourly* average concentrations, have an opportunity to include in their current manuscript an expanded discussion on the potential importance of aged stratospheric air originating at other locations that is transported to the site.

**Response:** Thank you for your comments. We agree the hourly average concentration is a better proxy for assessing enhancements induced by STE events and now present results of the multiple regression analysis using hourly ozone concentrations. The description of MLR in this study was adjusted in the manuscript as follows (lines 135 - 146):

"A Multiple Linear Regression (MLR) model was used in this study to quantify the main factors affecting hourly surface ozone concentrations. The method follows the description provided in de Foy et al. (2016b and 2016c). The inputs to the MLR model include meteorological parameters (wind speed, temperature, solar radiation and humidity), inter-annual variation factors, seasonal factors, diurnal factors, WRF boundary layer heights, WRF-FLEXPART trajectory clusters and the CAMx stratospheric ozone tracer. To obtain a normal distribution, the MLR model was applied to the logarithm of the ozone concentration offset by 10 ppb. For the WRF-FLEXPART clusters, a separate time series was constructed for each cluster, with 1 for the hours experiencing that particular cluster and 0 otherwise. The model estimated a coefficient corresponding to enhanced or decreased ozone concentrations for each cluster.

The inputs to the model were normalized linearly except for the ozone tracer which was transformed log-normally with 0 offset. Because the results of Least-Squares methods are sensitive to outliers, an Iteratively Reweighted Least Squares (IRLS) procedure was used to screen them out. Measurement times when the model residual was greater than two standard deviations of all the residuals were excluded from the analysis. This was repeated iteratively until the method converged on a stable set of outliers (de Foy et al., 2016a)."

Fig. S1 provides potentially important information about the day-to-day variability of the hourly concentrations. I have reproduced Fig. S1 below. The figure illustrates the variability of the hourly average concentrations for the period from January 2011 until October 2015. As anticipated, the frequency of the highest hourly average concentrations (e.g., 70 ppb to > 90 ppb) occurs during the springtime and early summertime (Fig. S1). This agrees with the authors' observations based on the monthly average concentrations. However, unlike the pattern described based on the monthly averages, high hourly average concentrations are also occurring during the winter and summertime for some of the years. During the months of SON, the frequency of high hourly average concentrations is much lower than those values exhibited during the DJF, MAM, and JJA seasons. Thus, there appears to be different patterns observed when using the monthly average concentration results with those using the hourly average concentration results.

Investigating the pattern for when the highest hourly average concentrations occur, it appears that this pattern is similar to the one described by Škerlak et al. (2014), which indicated stratospheric intrusion hotspots in the Tibetan Plateau during the months of DJF, MAM, and JJA. If the authors wish to, they have the opportunity to expand their discussion in their manuscript to comment on the degree to which the observed enhanced hourly average ozone concentrations may be associated at the Nam Co Station with STE.

[Figure]

Fig. S1. Variation of surface ozone at Nam Co Station from January 2011 to October 2015. Hourly mean mixing ratios of surface ozone are in blue dots; monthly mean mixing ratios of surface ozone are in black dots; average mixing ratio of surface ozone during whole measurement period in red dash line.

**Response:** Thank you for your suggestion.

Now we also investigated the STE happened by the meridional cross-section at 91°E (over Nam Co Station) monthly (Fig. 5) and the enhanced hourly average surface ozone concentrations at Nam Co Station associated with STE were analyzed by using CAMx stratospheric tracers (Table 2). Downward transport of stratospheric ozone contributed to high level of surface ozone at Nam Co Station. Following your suggestion, we have performed the Multiple Linear Regression (MLR) model by seasons using log-transforms and CAMx stratospheric tracers (Table S1). The regression model suggests that CAMx tracers contributed much more to surface ozone at Nam Co Station in the spring than during the rest of the year. The minimum impact of the CAMx tracers was during the autumn, which might be a reason for the low incidence of high hourly average concentration of surface ozone during the month of SON. MLR results indicated that although the mean contribution of the stratospheric tracer to surface ozone concentrations is only 1 ppb over the entire time series, it can reach above 20 ppb during specific events in the spring.

**Specific Line-by-Line Comments**

1. Title: I would suggest that the title be slightly modified as follows: Surface ozone at Nam Co in the inland Tibetan Plateau: variation, synthesis comparison and regional representativeness.

**Response:** Thank you for your suggestion. Title was changed according to your suggestion as follows:

"Surface ozone at Nam Co in the inland Tibetan Plateau: variation, synthesis comparison and regional representativeness".

2. Lines 24-25: The authors state "Model results indicate that the study site is affected by the surrounding areas in different seasons and that air masses from the northern Tibetan Plateau lead to increased ozone levels in the summer." I think the authors are not necessarily indicating that there is an increase during the summer at Nam Co due to air masses from the northern Tibetan Plateau but that the

air masses from the northern Tibetan Plateau ***contribute*** to the enhancement of ozone levels measured at the site. The word "increase" gives the impression that relative to the spring, the summer monthly averages are higher. The monthly average levels at the site are lower than those observed during the spring and therefore, I am suggesting a slight change in the wording.

**Response:** This sentence was changed according to your suggestion as follows (lines 25 -27):

"Model results indicate that the study site is affected by the surrounding areas in different seasons: air masses from the southern Tibetan Plateau contribute to the high ozone levels in the spring and enhanced ozone levels in the summer were associated with air masses from the northern Tibetan Plateau".

3. Lines 34-35: I would suggest references that represent comprehensive summaries of human health and vegetation effects, such as LRTAP Convention (2015), REVIHAAP (2013), and US EPA (2013).

**Response:** We added these references and sentence was rewritten as follows (lines 33 -35):

"High levels of surface ozone are currently a major environmental concern because of the harm ozone poses to health and vegetation at the surface (LRTAP, 2015; REVIHAAP, 2013; US EPA, 2013; Mauzerall and Wang, 2001; Desqueyroux et al., 2002)".

4. Line 45-46: The sentence: "In this situation, background sites can represent areas with surface ozone concentrations that are under the control of largely uniform synoptic systems and are minimally affected by local anthropogenic sources." What does "in this situation" refer to?

**Response:** We used "in this situation" to refer that "the surface ozone over the entire world can't be represented by one site or few sites. But background site can represent extended area in a relatively similar environment". It seems "in this situation" was misleading and redundant. Now we removed "in this situation" from the main text.

5. Lines 141-143: The sentence "In cells with high PSCF values are associated with the arrival of air parcels at the receptor site that have pollutant mixing ratios that exceed the criterion value" does not appear to be complete. Should the sentence start with "Cells with high PSCF…"?

**Response:** Thank you so much. We changed "In cells" to "Cells" in the manuscript and sentence was rewritten as follows (line 168 - 170) "Cells with high PSCF values are associated with the arrival of air parcels at the receptor site that have pollutant mixing ratios that exceed the criterion value".

6. Line 155: The sentence states "The mean surface ozone mixing ratio at Nam Co Station during the entire observational period was 47.6 ± 11.6 ppb…." I am not suggesting any change in this sentence but I do want to point out that the authors on Lines 33 and 34 state that "High levels of surface ozone are currently a major environmental concern because of the harm ozone poses to health and vegetation." This is a correct statement. However, researchers who assess human health and vegetation effects focus on the occurrence of high, as well as mid-level hourly average concentrations, and normally do not focus on high annual average concentrations. Annual, seasonal, or monthly average ozone concentrations are not necessarily the best metrics to use when assessing either human health or vegetation effects. While monthly and annual average concentrations are used for assessing the performance of global modeling results, these metrics are not necessarily relevant for assessing human health and vegetation effects.

Response: Thank you for pointing this out. We added the sentence as follows (lines 39 - 40):

"For global modeling, monthly and annual average concentrations of tropospheric ozone are used for assessing and improving the modeling results (Wild and Prather, 2006; Roelofs et al., 2003)".

7. Page 156: Table 1 indicates that the data capture was as follows: 2011 (75.25%), 2012 (90.30%), 2013 (75.90%), 2014 (70.05%), and 2015 (66.21%). Was the 66.21% data capture observed in 2015 related to the entire 12 months or was this value the data capture for the period January – October 2015?

Response: 66.21% in 2015 was the valid data during whole 2015 from January to December, and these valid data in 2015 was started from January 2015 to October 2015.

8. Lines 182-183: The authors state "The transition between high levels during the daytime and low levels during the nighttime was fast." I would appreciate it if the authors could please explain why the transition was fast.

Response: Thank you for your comment. The transition was probably caused by the vertical mixing and photochemical production which was induced by sunrise. We removed this sentence as we were not going to further expand this point.

9. Lines 186-187: The authors state "Relatively large diurnal amplitudes were observed in spring, with much smaller diurnal amplitudes observed during summer, autumn and winter." Can the authors offer an explanation for this observation? Could this observation be associated with STE making it to the

ground during the spring more frequently than during the other seasons?

**Response:** We added the explanation for the relatively large diurnal amplitudes in the spring in the manuscript as follows (lines 279 - 290):

"The larger diurnal amplitude of surface ozone in the spring than other seasons (Fig. 3, mentioned in section 3.3) may be related to four factors: (1) position of STE hot spot; (2) frequency of STE; (3) PBLH at Nam Co Station and (4) solar radiation at Nam Co Station. In the spring, plots of tropopause folding suggest that STE mostly happens in the southern Tibetan Plateau which is close to Nam Co Station and that STE even happens right above Nam Co Station. Furthermore, PBLH at Nam Co Station was higher in the spring than during the rest of the year. The higher PBLH in the spring facilitated the impact of downward transport from the stratosphere to Nam Co Station. The spring also has more intense solar radiation than the summer because the Monsoon leads to increased cloudiness in the summer. The Pearson's correlation coefficient between monthly SWD and surface ozone was ~0.93 in 2012 (2012 was selected because it had a more complete dataset than the other years) (Fig. 6) indicating that monthly surface ozone variability at Nam Co Station was associated with solar radiation. This was expected as increased solar radiation promotes the photochemical production of surface ozone in the spring, which is similar to the mechanism at other background sites (Monks 2000). Consequently, more photochemical production of ozone is expected in the spring".

10. Lines 194-196: The authors state "35S results (Lin et al., 2016) also support this result by showing that in the spring; Nam Co was affected by aged stratospheric air originating over the Himalayas rather than being affected by transport from fresh stratospheric air masses directly above Nam Co Station." Should the ";" be placed with a "," to make a complete sentence?

**Response:** Changed as suggested. This sentence was rewritten as follows (lines 277 - 279):

"Cosmogenic $^{35}$S results (Lin et al., 2016) also indicated that in the spring, Nam Co was affected by aged stratospheric air originating over the Himalayas rather than being affected by transport from fresh stratospheric air masses directly above Nam Co Station".

11. Lines 188-200: The authors state "A multiple linear regression model was used to quantify the contributions of various factors (including temperature, clear sky solar radiation, potential vorticity, wind speed, humidity, annual cycle, interannual variation and WRF-FLEXPART trajectory clusters) to the

measured maximum daily 8-hour average surface ozone." If in the authors' multiple linear regression model the variables (i.e., temperature, clear sky solar radiation, potential vorticity, wind speed, humidity, annual cycle, interannual variation and WRF-FLEXPART trajectory clusters) were not independent, what would be the effect on the outcome of the results using the model?

**Response:** We use block-bootstrapping to estimate the uncertainty in the results, including the impact of covariation in the inputs (de Foy et al., 2015). The results are presented for groups of variables arranged by the time scale of the variability and the type of inputs. These are mostly orthogonal to each other, although some of them have an inherent correlation. For example, the diurnal variation terms have an r2 of 0.21 with the boundary layer height and 0.19 with the local winds. Because we nonetheless wish to estimate the different contributions of these terms, we keep them separate in the analysis. Likewise, the CAMx stratospheric tracer and the seasonal time series have an r2 of 0.17. Because the stratospheric impacts are greater in the spring than during the rest of the year, the correlation between the two time series is inescapable. The block-bootstrapping method can be used to estimate the corresponding uncertainty in the results. Figure S4 shows the covariation of the results of the MLR analysis. The correlation coefficient squared (r2) of the contribution from the diurnal terms with the local winds is 0.08 and for the boundary layer height it is 0.06. This suggests that the correlation between the time series does not have a large impact on the results. For stratospheric tracer and the seasonal time series, the r2 is 0.5 which suggests that the correlation of the time series has a stronger impact on the estimation of the contribution of each term to the ozone variance in the measurements. A larger estimate of the contribution of the stratospheric tracer will lead to a lower estimate of the seasonal term and vice versa. This is reflected in the larger uncertainty in the estimates, as shown in Fig. S4.

12. Lines 209-211: The authors state" Specific humidity was the second largest contributor (20%; Table 2) with a negative coefficient indicating that higher surface ozone was associated with drier conditions possibly due to transport of continental air masses; or impacts from air masses aloft." If the Nam Co Station were influenced by "aged" stratospheric intrusions, would the lower humidity still be associated with the "aged" transported air from the stratosphere originating over the Himalayas after several days? Perhaps a short comment in the manuscript might be in order.

**Response:** Both continental air masses and air masses aloft can lead to low specific humidity, so we try to find another stratospheric incursion indicator. Now we used CAMx tracer instead of specific

humidity to identify the impact from the stratospheric incursion and CAMx tracer was a better indicator of stratospheric ozone incursion (lines 263 - 267):

"We performed a separate model run where we replaced the stratospheric tracer with the potential vorticity time series at 350 hPa above the Himalayas. The model found the best fit using the Kolmogorov-Zurbenko seasonally filtered time series of potential vorticity. The model had a slightly lower correlation coefficient, and lower contribution of the potential vorticity tracer (5.8%) than the model using the CAMx stratospheric tracer. This suggests that the CAMx stratospheric tracer was a better indicator of stratospheric ozone incursions than the time series of potential vorticity".

13. Lines 212-214: The authors state "The negative coefficient indicates that air masses transported from the south to Nam Co were associated with lower surface ozone. For the whole measurements period, it seems that transport of surface ozone is not the main influencing factor to the daily surface ozone variations in the multiple linear regression model." However, in Lines 287-290, the authors indicate that "Backward trajectories and PSCF were utilized to identify the source of surface ozone at Nam Co Station and to assess the regional representativity of surface ozone at Nam Co. In spring, the air masses that arrived at Nam Co Station were predominantly from the west and from the south, and the 3-D clusters indicated that the air masses traveled through the Himalayas before reaching Nam Co Station (Fig. 10)." If the air masses traveled through the Himalayas during the spring before reaching the Nam Co Station, at times would not the air masses represent "aged" stratospheric intrusions and wouldn't these air masses influence the daily surface ozone variation? Is there a difference in the conclusions reached using the multiple linear regression model versus the back trajectory and the PSCF analyses? Perhaps I am missing something here.

**Response:** We considered the transport by cluster in MLR and it was the secondary factor. The MLR results suggested that lower levels of surface ozone were associated with air masses came from the south (it was possibly related to the pollution emitted from Dangxiong and Lhasa) and higher levels of surface ozone were identified when air masses were from the north.

PSCF results was not separate from the stratospheric tracer and it is possible that PSCF picked up the contribution from STE as a signal from the south in the spring and from the north in the summer. PSCF results are different from MLR but not inconsistent.

14. Lines 256-258: The authors state "This type has a plateau of high surface ozone in spring and summer and a minimum in winter. Sites of this type occur in regions with strong ozone precursor emissions in the summer (such as the central European continent) or in regions where stratospheric intrusion occurs frequently in summer." Could the authors please provide examples for specific regions of the world where stratospheric intrusions frequently occur during the summer. Perhaps the results from Škerlak et al. (2014) might be a good source.

**Response:** Thank you for pointing this out. Regions including the Pamirs, Tian Shan, north-central US, Anatolia, northern side of the Tibetan Plateau, the east and west coasts of Australia, the northern Tasman Sea and Wilkes Land in East Antarctica were the places had stratospheric intrusions frequently during the summer (JJA) (Škerlak et al., 2014). But now we removed this part as suggestion from referee #2.

15. Lines 271-273: The authors state "Sites in the central Tibetan Plateau including Nam Co Station showed maximum ozone during late spring-early summer and relatively low levels in the remainder of year (Fig. 9B), corresponding to the Spring-maximum type. Compared with the surface ozone levels at Nam Co Station, those at Lhasa and Dangxiong were much lower." This conclusion is based upon the use of monthly average concentrations. Is there any indication that the use of the frequency of high hourly average concentrations might provide a different pattern?

**Response:** It is a good suggestion. We are looking forward to having collaborations with the researchers who work on the surface ozone measurement at Lhasa and Dangxiong. But now, we can only get the monthly average concentrations of surface ozone at Lhasa and Dangxiong from their publications and we were unable to investigate the pattern by using the frequency of high hourly average concentrations of surface ozone at these three sites now. We will try our best to investigate this in future.

16. Lines 313-314: The authors state "The atmospheric environment of the Tibetan Plateau and its relationship to regional and global change are of universal concern due to the rapid responses and feedbacks specific to the "Third Pole". I would appreciate it if the authors would please expand on this sentence to explain what they mean.

**Response:** The sentence has been rewritten as follows to make it clear and concise (lines 393 - 394):

"The changes of the atmospheric environment of the Tibetan Plateau are of universal concern due

to its rapid responses and feedback to regional and global climate changes".

17. Line 324-327: The authors state "Waliguan, in the northern Tibetan Plateau, is occasionally influenced by regional polluted air masses (Zhu et al., 2004; Xue et al., 2011; Zhang et al., 2011). Its mountainous landform facilitates mountain-valley breezes and may sometimes pump up local anthropogenic emissions especially during the winter (Xue et al., 2011)." I was under the impression that local anthropogenic sources are small near Mt. Waliguan. Mt. Waliguan is far from major cities, such as Xining (90 km) and Lanzhou (260 km) in the eastern sector. I would appreciate it if the authors would further elaborate concerning the enhancement at Mt. Waliguan from local anthropogenic emissions.

**Response:** Xue et al. (2011) stated "further analysis of backward trajectories for the recent 10 years indicated that WLG was frequently (∼50% of air masses) influenced by the air from the east, suggesting an important role of anthropogenic emissions in central and eastern China in shaping the summertime surface ozone and other atmospheric trace constituents at WLG and over the Tibetan Plateau." Zhang et al. (2011) stated "pollution episodes at WLG were characterized by significantly enhanced mixing ratios and large and erratic variations. This apparently reflects influence of regional emission sources on WLG"; "in summer, the most elevated CO mixing ratios are associated with cluster 3 which passed through the urbanized area southeast of WLG (e.g. Lanzhou city, the central region and southeast of Gansu province) " and "compared to the JFJ, air masses identified at WLG as polluted contained more CO relative to the background values and displayed large and irregular fluctuations suggesting greater influence from regional emission sources". Xue et al. (2013) stated at Waliguan, "the daytime upslope flow of boundary-layer air and nighttime downslope flow of free tropospheric air resulted in a reversed diurnal variation of trace gases at WLG. This unusual phenomenon could be explained by transport of anthropogenic pollution during the night. Transport of anthropogenic pollution from the northeast/east, where Xining and Lanzhou are located, is likely responsible for the enhanced levels of CO and VOCs during the nighttime at WLG." Refer to the description in these publications, Waliguan can be affected by the polluted air masses from regional emission sources and the enhanced levels of CO and VOCs during the nighttime at WLG was associated with upslope flow (night wind in mountain–valley breezes).

We adjusted this sentence as follows (lines 403 - 406):

"Waliguan, in the northern Tibetan Plateau, is occasionally influenced by regional polluted air

masses (Zhu et al., 2004; Xue et al., 2011; Zhang et al., 2011). Its mountainous landform facilitates mountain-valley breezes and may sometimes pump up anthropogenic emissions especially during the winter (Xue et al., 2011)".

18. Lines 332-335: The authors state "During the summer, surface ozone concentrations at Nam Co Station are higher than the northern hemisphere average, which suggests that there are impacts of long-range transport. Nam Co is less influenced by stratospheric intrusions than NCOP on the slopes of Mount Everest, and it is minimally influenced by local anthropogenic emission as evidenced by the constant long-term variation of surface ozone and consistent diurnal variation regardless of season, as discussed above." What is the influence of stratospheric intrusions on Nam Co during the summer? Škerlak et al. (2014) appear to indicate that it is important during the summer. If the surface ozone concentrations during the summer at Nam Co Station are higher than the northern hemisphere average, could the suggested long-range transport be associated with "aged" air masses from the stratosphere that are being transported to the site? I think it would help the reader to clarify what the authors mean by " there are impacts of long-range transport."

**Response:** Thanks for your comment. We add meridional cross-sections over Nam Co Station (Fig. 5) to indicate the position (altitude and longitude) of the strongest STE in the meridional cross-section (over Nam Co Station). In summer, the hot spot of STE was in the northern Tibetan Plateau and air masses from this region elevated surface ozone concentration at Nam Co Station in summer which was also showed in Fig. 11. Air masses with high concentration of ozone in stratosphere were probably first transported to the northern Tibetan Plateau then transported horizontally to Nam Co Station.

We added the description for this point in the manuscript as follows (lines 290 -294):

"In the summer (Jun, Jul and Aug), the jet core moved to the northern Tibetan Plateau and tropopause folding was relatively farther from Nam Co Station than those in the spring. Consequently, there was a smaller impact of stratospheric air at Nam Co Station. With tropopause folding further north in the summer, the air masses from the northern Tibetan Plateau may contribute more to the surface ozone levels at Nam Co Station than the air masses from the southern Tibetan Plateau".

(lines 378 - 381):

"In the summer, clusters from the northern Tibetan Plateau had higher mean surface ozone levels

than clusters which came from the southern Tibetan Plateau. The air masses that arrived at Nam Co Station from the northern Tibetan Plateau and northwestern China by horizontal wind transport likely resulted in the higher ozone concentrations at Nam Co Station in the summer".

19. Line 340: The summary needs to be expanded. It is very minimal at this time.

**Response:** The summary has been expanded to including major results and conclusions. Parts of summary were rewritten as follows (lines 420 - 436):

"The baseline of surface ozone is mainly controlled by various natural factors. Downward transport of air masses, air masses from the southern Tibetan Plateau in the spring and from the northern Tibetan Plateau in the summer contributed to the elevated monthly concentrations of ozone at the surface. Diurnal peaks of surface ozone in the afternoon were associated with high SWD, high PBLH and high wind speed. The analysis suggests that stratospheric intrusions account for around 20% of the variability in surface ozone concentrations at Nam Co Station. Further analysis of tropopause folding suggest that Nam Co Station is affected by "aged" air masses associated with stratospheric intrusions transported from the southern and northern Tibetan Plateau, mainly during the spring and the summer, respectively.

Synthesis comparison of ozone variability at regional and hemispheric scales revealed that the seasonality of surface ozone at Nam Co Station is most similar to other background sites in the Northern Hemisphere, albeit with slightly higher fluctuations in the summer season due to infrequent occurrences of air mass transport from Northwest China. Surface ozone at Nam Co showed distinct seasonal and diurnal variation patterns as compared with those sites in the Himalayas and the northern Tibetan Plateau. The monthly maximum of surface ozone at Nam Co Station was later in the year than the sites in the southern Tibetan Plateau and the southern ridge of the Himalayas, but earlier than the sites in the northern Tibetan Plateau.

Our measurements provide a baseline of tropospheric ozone at a remote site in the Tibetan Plateau, and contribute to the understanding of ozone cycles and related physico-chemical and transport processes over the Tibetan Plateau. More long-term measurements of surface ozone at field sites covering the spatially extensive Tibetan Plateau are needed to improve our understanding of surface ozone variations and the underlying influence mechanisms".

20. Lines 348-349: The authors state " Synthesis comparison indicated that Nam Co is less

influenced by stratospheric intrusions and anthropogenic disturbances than sites along the rim of the Tibetan Plateau." I would appreciate it if the authors could please clarify this sentence. Should the sentence read "While the Nam Co Station is less influenced by stratospheric intrusions and anthropogenic disturbances than sites along the rim of the Tibetan Plateau, the site does exhibit during specific months large contributions associated with transported "aged" air masses associated with stratospheric intrusions." I do not wish to impose this interpretation on the authors, but rather elicit from them if this is what they are attempting to say. If not, could they please provide a concise sentence that clearly describes their conclusion on the importance of stratospheric intrusions associated with long-range transport in enhancing the surface ozone concentrations at Nam Co. I think this would help the reader.

**Response:** Thanks for your comment. We rewrote this sentence as follows (lines 423 - 426):

"The analysis suggests that stratospheric intrusions account for around 20% of the variability in surface ozone concentrations at Nam Co Station. Further analysis of tropopause folding suggest that Nam Co Station is affected by "aged" air masses associated with stratospheric intrusions transported from the southern and northern Tibetan Plateau, mainly during the spring and the summer, respectively".

21. Supplement: Fig. S1. I would suggest improving the readability of the title of the x-axis (Year-Month-Day-Hour). It seems to not be clear on my copy. Does the first symbol in the time series identified as 2011-01-01 in Fig. S1 represent the January average or just the 2011-01-01 point? I am not sure what the first dot represents. The meaning of the first dot is confusing.

**Response:** Thanks for your suggestion. We added a new version of Fig. S1 in manuscript as follows:

[Figure]

Fig. S1. Variation of surface ozone at Nam Co Station from January 2011 to October 2015. Hourly mean mixing ratios of surface ozone are in blue dots; monthly mean mixing ratios of surface ozone are in black squares; average mixing ratio of surface ozone during whole measurement period in red dash line.

**References**

[revised manuscript text omitted]

---

## Author Comment (AC2) · 4 Jul 2017

GENERAL COMMENTS

This work by Yin et al. presents an overview of about 5 years of continuous near-surface ozone observations at the Nam Co station which is located in the central Tibetan Plateau. The scope of the paper is rather ambitious: to characterize the typical variability of near-surface O3 at this measurement site, to compare it with other sites in the Tibetan Plateau (and beyond) and to demonstrate that this site is representative for the whole Tibetan Plateau. The presented data-set is of great interest (and I suggest to share it in the framework of international initiatives like WMO/GAW or TOAR/JOIN). However, the paper is a little bit confusing and for a great part relies too much in other studies, resembling more a "review" than a research paper. Moreover, some important conclusions were based too much on qualitative assertions. As an instance, in my opinion, the authors failed in demonstrating that: "The unique geographical characteristics make Nam Co Station more representative of the baseline of surface ozone in the extensive inland of Tibetan Plateau than other existing monitoring sites", as they report in the Summary. More analyses/comparisons are needed to assess this point! My impression is that the authors mixed together several different analyses without a well-defined scientific track. For instance, at least two different model (FLEXPART- WRF and HYSPLIT) were used with the same aim (characterize O3 variability as function of air-mass transport) but without any critical comparison or integration. The fact that O3 is positively correlated with some meteorological parameters is not of great scientific novelty and (the most important point) I suspect that the linear model results were significantly affected/biased by the use of daily average values (at least for ozone). The discussion about the role of STE is simply

based on a subjective (mainly visual) analysis of O3 variability with stratospheric "tracers" (not specific analyses or tool have been used). For these reasons, I suggest to resubmit the paper after than some essential modifications have been made. In the following I provide some suggestion to help authors towards this aim.

**Response:** Thanks for the comments. In this revised version, we added more analysis including Multi Linear Regression which was calculated using hourly data as you suggested, Meridional cross-sections over Nam Co Station derived from ERA-Interim data and a tracer for stratospheric ozone incursions which was obtained using the CAMx.

As a result of the reviews, we have refined the analysis of potential vorticity as a tracer for stratospheric air and we have also expanded the regression analysis to include tracers for stratospheric ozone transport using an air quality model. In the ACPD manuscript, we had used Potential Vorticity near the surface (500 hPa) to test for stratospheric incursions. However, this did not lead to a clear signal in the regression analysis. Based on new research, we have now found that if we use PVU at the 350 hPa level we detect an influence on the ozone time series. If we use PVU at 350 hPa above the Himalayas then this signal is even clearer. The description of the regression analysis has been expanded and the results updated accordingly.

An even better match for stratospheric incursions was obtained when we used ERA-Interim ozone concentrations aloft as boundary and initial conditions for the CAMx air quality model. Chemistry was turned off to obtain a passive tracer of stratospheric air at the measurement site. This gave a signal in the regression analysis that is even stronger than the new PVU analysis. The text was expanded and the results updated in the manuscript as follows (lines 124 – 131):

"A tracer for stratospheric ozone incursions at the measurement site was obtained using the CAMx (Comprehensive Air-quality Model with eXtensions) v6.30 model (Ramboll Environ, 2016). The model initial and boundary conditions were obtained from ERA-Interim ozone fields, retaining only concentrations above 80 ppb and higher than 400 hPa. CAMx simulations were performed using the WRF medium and fine domains (domains 2 and 3) in nested mode for the full 4 year time series. In order to serve as a tracer for direct transport, there was no chemistry in the model and ozone was treated as a passive tracer. The resulting time series of the tracer concentration at the measurement site was used as

input in the multi-linear regression model. This is similar to the procedure described in de Foy et al. (2014) to estimate the impact of the free troposphere on surface reactive mercury concentrations.".

Fig. 4 and table 2 were added to explain the new analysis. The model suggested that up to 20% of the ozone variability was due to stratospheric incursions. Meridional cross-sections over Nam Co Station (Fig. 5) illustrated the position of downward transport of stratospheric ozone in different seasons.

SPECIFIC COMMENTS

Line 43-45: I think that this sentence is meaningless.

**Response:** Thank you for your comment. Now we removed this sentence.

Line 55: this is wrong. At NCO-P the highest contributions from STE is in WINTER. This is clearly stated by Cristofanelli et al. ACP (2010) and Putero et al. ACP (2016). The pre-monsoon (spring) O3 peaks was strongly affected by the transport of pollution from the lower troposphere (Himalayas foothills and Indo-Gangetic Plains). See e.g. Putero et al. Atmospheric Pollution (2013); Bonasoni et al., ACP (2010).

**Response:** Thank you for pointing out. We rewrote this sentence as follows (lines 56 - 57):

"At NCO-P and Xianggelila, surface ozone maximum was observed in spring (Cristofanelli et al., 2010; Ma et al., 2014)".

Line 69-70: this sentence is too generic. Specify what kind of ozone-related climatic and environmental effect can be assessed and by which methodology.

**Response:** The sentence has been rephrased for specific meaning as follows (lines 68 - 71):

"This study expands the understanding of baseline and variations in the surface ozone concentration and the transport processes that influence tropospheric ozone in the inland Tibetan Plateau. The long-term measurements of surface ozone; together with other reported surface ozone time series over the Tibetan Plateau represent valuable datasets for evaluating long-term regional-scale ozone trends".

Line 84: remove the capital letter from "The"

**Response:** Thanks. Changed as suggested in line 85.

SECTION 2 Line 95: how did you evaluate change in sensitivity? By which frequency the analyser was calibrated? The calibrator 49iPS was calibrated against which reference instrument?

**Response:** Now we modified the description in the main text as (lines 97 - 98):

"Yearly instrument calibrations are performed against the Standard Reference Photometer (SRP) maintained by the WMO World Calibration Centre in Switzerland (EMPA)".

Section 2.4: Which is the time resolution of the inputs to the MLR Model (hourly, daily)? How did you consider the FLEXPART trajectory cluster in the regression analysis? Why did you normalize the input parameters? Why did you exclude outliers? The last three sentence are rather obscure to me (from line 126). Please, provide a clear step-by step description of the methodology. By only considering the maximum 8-hour average ozone concentration, you discharge all the information about variability at hourly scale (which is rather important). . .and this is the reason why you find out a great role of radiation! At least, this must be clearly stated in the revised manuscript.

**Response:** Thank you for your suggestion. We agree the hourly average concentration is a better proxy for assessing enhancements induced by STE events and now present results of the mupltiple regression analysis using hourly ozone concentrations. The MLR analysis estimates the impact of the WRF-FLEXPART clusters on the ozone levels at the measurement site. For the WRF-FLEXPART clusters, a separate time series was constructed for each cluster, with 1 for the hours experiencing that particular cluster and 0 otherwise. The model estimated a coefficient corresponding to enhanced or decreased ozone concentrations for each cluster.

Least-Squares methods are sensitive to outliers. This is why we remove them from the analysis in order to have a more robust analysis. There are many approaches to this problem described in the literature under the heading of "robust estimation" for example. The specific method used here is described in (de Foy et al., 2016a).

It is common practice to normalize the input parameters for a regression analysis. This improves the stability and the robustness of the estimates. Please refer to the statistics literature for more detail.

The description of MLR in this study was adjusted in the manuscript as follows (section 2.4, lines

135 - 146) :

[revised manuscript text omitted]

Line 100: I think that I would be better and more useful to refer the measurements to the "local time" instead of "Beijing time".

**Response:** Thanks for your suggestion. We hope to keep the data displayed in UTC+8 (Beijing Time) in this study because all the measurements in this study were recorded in UTC+8 and all the models in this study were also calculated in UTC+8.

Line 110: please provide more info about the HYSPLIT simulation set-up. Which meteorological gridded data-set has been used to calculate back-trajectories (GFS)? By which time resolution did you calculate back-trajectories (Once a day? Every hour?)? How did you take into account uncertainties due to the complex topography surrounding the Plateau? Also provide more info about the cluster methodology and provide a description of the algorithm. Provide web access indication to the TRAJplot software. I think that both NOAA (for providing GDAS and HYSPLIT) and TRAJPlot developers must be acknowledged in this paper. I guess WRF-FLEXPART is much more accurate in reproducing air-mass origin and transport to Nam CO. However, please provide more technical details about the model set-up. It is not clear to me which is the reason to use HYSPLIT when WRF-FLEXPART is available. Please, explain. Did you compare the results obtained with FLEXPART and HYSPLIT?

**Response:** We used TrajStat for clusters calculation and reference was listed in the main text; now we added a reference in the main text (line 116) (Sirois and Bottenheim, 1995), and the descriptions of backward trajectory clusters methodology and the algorithm in this article were very detailed. We revised the description of HYSPLIT (lines 109 - 116):

"Gridded meteorological data for backward trajectories in HYSPLIT were obtained from Global Data Assimilation System (GDAS-1) by the U.S. National Oceanic and Atmospheric Administration (NOAA) with 1°×1° latitude and longitude horizontal resolution and vertical levels of 23 from 1000 hPa to 20 hPa (http://www.arl.noaa.gov/gdas1.php). The backward trajectories arrival height was set at 500 m (500 m, 1000 m and 1500 m were tested as arrival height and there was no obvious difference in results)

above the surface and the total run times was 120 hours for each backward trajectory and in time interval of 3 hours during whole measurement period. The vertical motion was calculated using the default model selection, which used the meteorological model's vertical velocity fields. Angle distance (Sirois and Bottenheim, 1995) was selected to calculate clusters in this study".

The set-up of WRF-FLEXPART can refer to de Foy et al. (2016a) which was mentioned in the main text (lines 119 - 120).

HYSPLIT and WRF-FLEXPART were both widely used. FLEXPART and HYSPLIT were used for different purposes in this study. We used WRF-FLEXPART to generate inputs to MLR model which is better than HYSPLIT and we used HYSPLIT to generate the trajectories be used in PSCF calculation.

Line 117: "Six clusters were found. . .". Does this sentence refer to HYSPLIT or FLEXPART? Not clear . . ..

**Response:** Sentence was rewritten as follows (lines 122 - 123):

"Six clusters were found to represent the dominant flow patterns to the Nam Co Station by using WRF-FLEXPART".

Section 2.5: What model did you use for this analysis (HYSPLIT or WRF-FLEXPART)? Did you consider some altitude/pressure level thresholds of back-trajectory points to allow the PSCF calculation? If not, hardly you can relate the obtained results with surface emissions. . ..The W values are a key parameter for the interpretation of the obtained results. How did you define them? Did you perform a sensitivity study by changing the weighting factor?

**Response:** In this study, PSCF was calculated by using trajectories which were calculated by HYSPLIT. The top of the model was set to 10000 m.

W values was set as follows: $Wij \begin{cases} 1.00 & n_{ij} > 3N_{ave} \\ 0.70 & 3N_{ave} > n_{ij} > 1.5N_{ave} \\ 0.42 & 1.5N_{ave} > n_{ij} > N_{ave} \\ 0.05 & N_{ave} > n_{ij} \end{cases}$

where Nave represents the mean nij of all grid cells. The weighted PSCF values were obtained by multiplying the original PSCF values by the weighting factor. We used several values of W and found that by using the values listed in our manuscript, the most information can be kept in PSCF result.

SECTION 3

Line 158: please attribute the origin of these anomalous events

**Response:** Thanks for your suggestion. Here we just want to show general characteristics of ozone levels. In this paper, we focus on ozone levels and its temporal changes over the long-term monitoring period, diagnoses of specific ozone elevation can be meaningful but is beyond the scope of the current study. We plan to investigate ozone anomalous events and using more data from sites around Nam Co in the near future.

Line 161: for the period 2006 – 2011 Putero et al (2013) found an average O3 of 48.7 ppb at NCOP, while Cristofanelli et al. (2010) over two year investigation pointed out an average value of 49 ppb. Thus, I would say that average value at Nam Co and NCO-P are comparable. Please correct.

**Response:** Thank you for pointing this out. Now we rewrote this sentence as (lines 186 - 189):

"The mean surface ozone mixing ratio at Nam Co Station was within the reference range reported for the Himalayas and Tibetan Plateau, and it was higher than the ratios for the two nearest urban sites: Lhasa (Ran et al., 2014) and Dangxiong (Lin et al., 2015); and comparable to of two sites on the edge of the Tibetan Plateau: Waliguan Station (Xu et al., 2011) and NCO-P (5079 m) (Cristofanelli et al., 2010) (see Fig. 1 for station locations).".

Line 162: different factors influence background O3 levels, i.e. altitudes, latitude, site classification (mountain, coastal, marine). The authors must better address this comparison taking into account all these factors.

**Response:** We agree. As stated in Vingarzan (2004), "The ozone concentration in any given area results from a combination of formation, transport, destruction and deposition", here we would just like to make a simple comparison to show the baseline of surface ozone at Nam Co in a global context. A comprehensive comparison in terms of altitudes, latitudes and site classification is more informative but is beyond the scope of the current paper. In addition, the range of 20-45 ppb is actually surface ozone baseline at background sites over the mid-latitudes in the Northern Hemisphere as indicated in Vingarzan (2004). We rephrased the sentences to make it clear and concise (lines 189 - 192):

"Surface ozone mixing ratios at Nam Co as well as other sites over the Tibetan Plateau were

generally higher than the range of 20-45 ppb measured at background sites in the mid-latitudes of the Northern Hemispheres. This was in agreement with the higher concentrations typically seen at sites located in the free troposphere (Vingarzan, 2004)".

Line 166: So, did you consider months with at least a 60% data coverage. Please specify this point rather than indicating the number of hours.

**Response:** Thanks for your suggestion. We specified this point as follows (line 194):

"Every month considered in this study had more than 400 hours of available data (valid data for each month >56%)".

Section 3.3: would remove Fig 3 and leave only Fig 4 (where diurnal variability are also more evident). However, for each hourly average you must add an error bar denoting the 95% confidence level of the mean average value.

**Response:** Thanks for your suggestion.

Fig. 3 was removed now and we adjusted Fig. 4 as you suggested (now it is Fig. 3).

[Figure]

Fig. 3. Diurnal profiles of average hourly surface ozone at Nam Co Station by seasons. Error bars are 95% confidence levels.

At this point, a description of typical local wind variability (wind speed and direction) must be added to evaluate possible influence of diurnal wind breeze on O3 variability.

**Response:** Wind rose at Nam Co Station during the day (a) and at night (b) was now added as Fig. S7. Description of local wind variability was added as follows (lined 308 - 309):

"There was a lake-land breeze influencing Nam Co Station and the wind speed in the daytime was

[Figure]

Fig. S7 Wind ross at Nam Co Station during the day (a) and at night (b).

Section 4.1: This analysis of stratospheric intrusion is too raw. I would like to see a more specific investigation (see e.g. Cristofanelli et al., 2010; Putero et al., 2016; Trickl et al., ACP, 2010). The authors only described in a very qualitative and oversimplified way (basically by "visual" inspection) the time series of stratospheric air markers (any statistical analysis or selection methodology is applied). Moreover, the assumption that stratospheric intrusion can be directly related to the daily maximum of ozone is wrong. Due to mixing and dilution processes, stratospheric air-masses are often characterized by O3 values which are even lower than those due to photochemistry. Moreover, these events are often characterized by short time duration (even lower than 1 day), thus simply comparing time series of stratospheric tracers with a daily time resolution can mask the real influence of STE. The final sentence: "Nam Co was affected by aged stratospheric air originating over the Himalayas rather than being affected by transport from fresh stratospheric air masses directly above Nam Co Station ", it's not clear to me. Quantify "aged". Section 4.2: I suggest to perform this analysis also on a seasonal basis. Since most of the used predictors are characterized by significant seasonal cycles, this would provide more hints about the role of single factors in driving O3 variability. Figure S4 it's not clear at all. What is the scale reported on the right bar? Line 210: "impacts from air masses aloft". Be more specific! Line 213: " why these airmasses are depleted in O3". I suspect simply because they were related to southern air-mass advection during the monsoon. Please provide a description of the seasonal frequency of occurrence of air-mass transport patterns reported by Fig. S4. You stated that: "For the whole measurements period, it seems that transport of surface ozone is not the main influencing factor to the daily surface ozone variations in the multiple linear regression model". I'm not convinced. As showed by other works (see Di Carlo, JGR, 2007). The role of dynamic is important at hourly timescale. By analysing data as daily averages you ruled out by default these contributions! By comparing the time series of O3 observations with the regression model (Fig. 5), it is rather clear than the model was not able to reproduce the spring peak. To my opinion, this is a clear hint toward an important contribution of transport and dynamics. Section 4.3: If data analyzed are daily averages, the correlation coefficient here provided (R: 0.77) does not describe the "local" (in-situ) role of photochemistry. This must be described by analysing the hourly data-set as you did for wind speed and PBLH. Which is the correlation coefficient between hourly ozone and hourly SWD? As suggested by Fig.7, the higher correlation with wind speed and PNLH suggest that dynamics is the most important factor explaining diurnal O3 variability. I suggest to apply the linear correlation model both for daily and hourly values and to comment differences in the results.

**Response:** Thanks for your suggestion. This was also a concern of Referee #1. We now present results using hourly data from the regression model. In the ACPD paper, we had found that potential vorticity near the surface did not correspond to higher ozone concentrations in the regression model. However, new research has found that potential vorticity aloft (350 hPa) did correspond to higher ozone. This suggests that PVU aloft could be used as a tracer of STE's. We also performed extensive new simulations of the impact of ERA-Interim stratospheric ozone at the surface using an air quality model. This provided time series of stratospheric tracers at the surface which were found to contribute up to 20% of the ozone variability at the site. Thanks to these new results, we have rewritten section 4, please see our response to your comments above.

Line 245: "the background ozone at the site": this is contradictory, the background cannot be local!

**Response:** Thanks for your suggestion. This sentence was changed as follows (lines 328 -333):

"The seasonal variation of surface ozone mixing ratios at different sites around the world is influenced by many factors including: stratospheric intrusion, photochemical production, long-range

transport of ozone or its precursors, local vertical mixing and even deposition (Vingarzan, 2004; Ordónez et al., 2005; Tang et al., 2009; Reidmiller et al., 2009; Cristofanelli et al., 2010; Langner et al., 2012; Ma et al., 2014; Lin et al., 2015; Ran et al., 2014; Xu et al., 2011; Macdonald et al., 2011; Pochanart et al., 2003; Derwent et al., 2016; Lin et al., 2014; Tarasova et al., 2009; Gilge et al., 2010; Wang et al., 2011; Wang et al., 2009; Zhu et al., 2004; Zhang et al, 2015; Nagashima et al., 2010)".

SECTION 5. It is not clear why in Figure 8 you reported "normalized O3" for NCOP. Please explain what kind of normalization was applied.

**Response:** We made Fig. 8 based on Cristofanelli et al. (2010) who reported the diurnal cycle of normalized ozone values. Cristofanelli et al. (2010) investigated the average diurnal variation of normalized O3 values obtained by subtracting daily means from the actual 30-min ozone concentrations.

At Xianggelila, Ma et a. (2014) reported that at diurnal scale O3 was strongly correlated with wind speed (as occurred also at Nam CO) and that "the transport and deposition will be the key factors influencing the diurnal variations of surface O3 at Xianggelila, a remote and clean site, rather than local photochemical processes". Also at Dangxiong, Lin et al. (2015), suggested that the correlation with high wind speed and O3 during the afternoon pointed out the important role of transport in affecting O3 more than photochemistry. I would bet that the same is true for Nam CO.

**Response:** We investigated the relationship between surface ozone and wind speed. The correlation coefficient between surface ozone and wind speed was 0.95 which indicating that high level of surface ozone was associated with high wind speed. It is important to note that local wind speed also correlates with time of day and with the evolution of the boundary layer height. In our regression model, we include all these factors and estimate the uncertainty due to the covariance by carrying out a block-bootstrapping analysis. The regression analysis suggests that local winds account for 31% of the ozone variability at the site (line 240, table 2).

Section 5.2: In my opinion the classification of the seasonal ozone regimes I-III is oversimplified (see the nice work by Tarasova e al., 2007, ACP). I suggest the authors to skip this first part (line 243-263) and discuss the O3 variability at the Tibetan sites as a function of the characterization provided by Tarasova et al. 2007. Line 256: please provide adequate references. Line 260: I think that this sentence only refers to summer season. Please, specify.

Response: Thanks for your suggestion. Now we removed "Type of seasonal variation of surface ozone in the Northern Hemisphere" from the manuscript.

Line 275: The possible impact of NO titration to the appearance of lower ozone levels at the the Tibetan sites should be better assessed/showed. For instance, you can report diurnal variability as a function of different seasons for these sites. NCO-P is not located over the Tibetan Plateau but at the southern ridge of Himalayas. Please correct.

Response: Thanks for your suggestion. We will try our best to look into the diurnal variability as a function of different seasons for these sites in future. The description of the location of NCO-P was changed as suggested (lines 353 - 355):

"In the southern Tibetan Plateau and the southern ridge of the Himalayas, Xianggelila and NCO-P each had a single surface ozone peak in spring (pre-monsoon) and a minimum in summer (monsoon) with a difference between the two exceeding 30 ppb".

Line 290: Figure 10 is hard to read and clusters look very similar each other's (except than for those related to southerly circulation).

Response: Thank you for pointing out. Now we revised Figure 10 to make it easier to read.

[Figure]

Fig. 10. Backward HYSPLIT trajectories for each measurement day (black lines in the maps), and mean back-trajectory for 6 HYSPLIT clusters (colored lines in the maps, 3D view shown on the right of the maps) arriving at Nam Co Station by season.

What kind of cluster algorithm was used? It looks that a large part of the information carried by the back-trajectories was missed by this clustering. Nevertheless, in agreement with this analysis, during Spring only a fraction (about 18%) of back-trajectories crossed the Himalayas. This must be clearly stated.

**Response:** Angle distance (Sirois and Bottenheim, 1995) was selected to calculate clusters in this study. In spring, cluster 1 (32.56%) and cluster 4 (17.74%) were the clusters crossed the Himalayas in

different pathways.

Line 292: Actually, Skerlak et al. (2014) reports a maximum of deep STT over the Tibetan Plateau and not only over Himalayas! In my opinion, your conclusion that O3 is higher at NCO-P due to a larger contribution from stratosphere is wrong. Looking at your Fig. 9, it looks that O3 values at NCO-P and Nam Co were well comparable on March and May. O3 was higher at NCO-P in April, but (as I reported below) the contribution of polluted air-masses in driving O3 variability at NCO-P during this season cannot be neglected! Line 294: I think that at this point the transport of polluted air-masses from Himalaya foothills and IGP to high Himalayas must be considered (see Bonasoni et al., 2010; Putero et al., 2013; Luthi et al., 2015)! This contributed to the appearance of the premonsoon maximum at NCOP and possibly the cross-Himalaya transport can also affect Tibetan Plateau.

**Response:** We thought the ozone values at NCO-P in March, April and May was much higher than those at Nam Co Station accordingly.

Polluted air-masses can contribute to the surface ozone variability at NCO-P and contribution can also contribute to the elevated level of surface ozone at Nam Co Station. Now we added the description for this as follows (lines 366 - 368):

"The contribution of polluted air masses in driving ozone variability at the southern ridge of the Himalayas was remarkable in the spring and it may also have an effect on the level of surface ozone at Nam Co Station through transport".

Line 296: which cluster was associated to the northern TP? It is not possible to recognize it from Figure 10 (please increase the fonts used for legend!)

**Response:** Thank you for pointing out. Cluster 2, 3 and 5 were associated to the northern Tibetan Plateau. Now we made Figure 10 easier to read.

Line 297: I read carefully Skerlak et al (2014) but I was not able to found any reference to the higher stratospheric flux over the northern Plateau in respect to the southern Plateau in autumn. Indeed, looking at their Fig. 6, this not looks to be the case.

**Response:** Thanks for your comment. Now meridional cross-sections over Nam Co Station (Fig. 5) were added to indicate the position (altitude and longitude) of the strongest STE in the meridional crosssection (over Nam Co Station) in different months (line 272 - 298):

"In order to visualize the transport of ozone from the stratosphere to the troposphere, we analyzed the upper troposphere and lower stratosphere structures of the meridional cross-section of monthly mean ERA-Interim data above Nam Co Station (Fig. 5). In the spring (Mar, Apr and May), the dynamical tropopause (identified by the isolines of 1 and 2 potential vorticity unit) exhibited a folded structure over the Tibetan Plateau. This tropopause folding can lead to a downward transport of ozone from the stratosphere to the troposphere. Tropopause folding happened in the southern Tibetan Plateau and close to Nam Co Station in the spring. Cosmogenic [35]S results (Lin et al., 2016) also indicated that in the spring, Nam Co was affected by aged stratospheric air originating over the Himalayas rather than being affected by transport from fresh stratospheric air masses directly above Nam Co Station. The larger diurnal amplitude of surface ozone in the spring than other seasons (Fig. 3, mentioned in section 3.3) may be related to four factors: (1) position of STE hot spot; (2) frequency of STE; (3) PBLH at Nam Co Station and (4) solar radiation at Nam Co Station. In the spring, plots of tropopause folding suggest that STE mostly happens in the southern Tibetan Plateau which is close to Nam Co Station and that STE even happens right above Nam Co Station. Furthermore, PBLH at Nam Co Station was higher in the spring than during the rest of the year. The higher PBLH in the spring facilitated the impact of downward transport from the stratosphere to Nam Co Station. The spring also has more intense solar radiation than the summer because the Monsoon leads to increased cloudiness in the summer. The Pearson's correlation coefficient between monthly SWD and surface ozone was ~0.93 in 2012 (2012 was selected because it had a more complete dataset than the other years) (Fig. 6) indicating that monthly surface ozone variability at Nam Co Station was associated with solar radiation. This was expected as increased solar radiation promotes the photochemical production of surface ozone in the spring, which is similar to the mechanism at other background sites (Monks 2000). Consequently, more photochemical production of ozone is expected in the spring. In the summer (Jun, Jul and Aug), the jet core moved to the northern Tibetan Plateau and tropopause folding was relatively farther from Nam Co Station than those in the spring. Consequently, there was a smaller impact of stratospheric air at Nam Co Station. With tropopause folding further north in the summer, the air masses from the northern Tibetan Plateau may contribute more to the surface ozone levels at Nam Co Station than the air masses from the southern Tibetan Plateau. In the autumn (Sep, Oct and Nov) and the winter (Der, Jan and Feb), the heights of folding were higher

than those in the spring and the summer; and the PBLHs in the autumn and the winter were much lower than those in the spring and the summer. Furthermore, SWD in the autumn and the winter were weaker than those in the spring and the summer. These factors contributed to the relatively low level of surface ozone at Nam Co Station in the autumn and the winter."

Line 301-304: Is this confirmed also by WRF-FLEXPART clustering?

**Response:** While we used WRF-FLEXPART and HYSPLIT for different purpose, this is also confirmed by WRF-FLEXPART.

Line 305-312: were these results confirmed by the HYSPLIT clustering? I expect that WRF-FLEXPART could have much more skill than HYSPLIT (based on global meteorological fields with coarse spatial resolution) in analysing spatial "contributions" for elevated O3 values at Nam CO. However, you must attribute the seasonal variability of the "contributions" you found by WRF-FLEXPART (by what kind of emissions, precursors are emitted over each identified regions?). Moreover, you should discuss and quantify the uncertainties related with this analysis. Also some details were missed: as an instance, for the seasonal analysis you used as O3 threshold values, the seasonal averages or the whole period average? What happens if different threshold were applied (e.g. 75th or 90th percentiles of ozone distribution)? Probabilities higher than 1.0 were reported in the legends: I think this is inconsistent. . .please check!

**Response:** Thank you for pointing this out.

These results were calculated by PSCF which using backward trajectories calculated by HYSPLIT.

We considered the transport by cluster in MLR and it was a secondary factor. The MLR result suggested that lower levels of surface ozone were associated with air masses that came from the south (it was possibly related to the pollution emitted from Dangxiong and Lhasa) and higher levels of surface ozone were identified when air masses were from the north. PSCF results do not identify the stratospheric tracer separately and it is therefore possible that PSCF picked up the contribution from STE as a signal from the south in the spring and from the north in the summer. The fact that the MLR results account for the stratospheric tracers separately explains why we obtained PSCF results that are different but not inconsistent from the MLR model.

With respect to the seasonality of the WRF-Flexpart results we have added a new table S1 that shows the MLR results by season. These also suggest that most of the STE's occur in the spring

As we mentioned in section 2.5, in this study, PSCF was calculated basing on trajectories corresponding to concentrations that exceed the mean level of surface ozone. When we used 75th and 90th percentiles of surface ozone distribution as the threshold in PSCF, there were a lot of information being missed.

We checked legend which was automatic generated by MeteoInfo and now the legend in figure was revised.

Section 5.4: This section about representativeness of Nam CO is mostly based on an intuitive/subjective approach and from review of previous works. Even if I'm personally convinced that Nam Co is an interesting background site, the authors must perform much work if their want to unambiguously assess the spatial representativeness of the station. See for instance Henne et al., ACP, 10, 3561–3581, 2010. I do not think that a "consistent diurnal variability of ozone regardless of season" can be used as proof to claim the large spatial representativeness of the station. Moreover, it seems that the authors do not consider STE as part of the "global" background ozone: from my point of view, this is completely wrong. If not specific analyses are accrued out, I strongly recommend to eliminate this section and limit some lines of comment in the summary Section.

**Response:** We rewrote this section and named it as "Implication for measurement and study of surface ozone in the inland Tibetan Plateau and beyond". This section was rewritten as follows (lines 393 - 413):

"The changes of atmospheric environment of the Tibetan Plateau are of universal concern due to its rapid responses and feedbacks to regional and global climate changes. The Tibetan Plateau covers vast areas with varied topography; however, comprehensive monitoring sites are few and sporadically distributed. Analysis of atmospheric composition at Waliguan in the north and Everest in the south of the Tibetan Plateau have shown that they are representative of high-altitude background sites for the entire Tibetan Plateau. It is noteworthy that the Tibetan Plateau, as a whole, is primarily regulated by the interplay of the Indian summer monsoon and the westerlies; and the atmospheric environment over the Tibetan Plateau is heterogeneous. Mount Everest is representative of the Himalayas on the southern edge

of the Tibetan Plateau and is the sentinel of South Asia where anthropogenic atmospheric pollution has been increasingly recognized as disturbing the high mountain regions (Decesari et al., 2010; Maione et al., 2011; Putero et al., 2014). In addition, Mount Everest has been identified as a hotspot for stratospheric- tropospheric exchange (Cristofanelli et al., 2010; Škerlak et al., 2014) where the surface ozone is elevated from the baseline during the spring due to frequent stratospheric intrusions. Waliguan, in the northern Tibetan Plateau, is occasionally influenced by regional polluted air masses (Zhu et al., 2004; Xue et al., 2011; Zhang et al., 2011). Its mountainous landform facilitates mountain-valley breezes and may sometimes pump up anthropogenic emissions especially during the winter (Xue et al., 2011). Nam Co Station, in the inland Tibetan Plateau, is distant from both South Asia and northwestern China, it has been found to be influenced by episodic long-range transport of air pollution from South Asia (Xia et al, 2011; Lüthi et al., 2015), evidenced by the study of aerosol and precipitation chemistry at Nam Co Station (Cong et al., 2007; Cong et al., 2010). As for surface ozone, Nam Co Station is less influenced by stratospheric intrusions directly than NCO-P, and is minimally influenced by local anthropogenic emission. It showed distinct seasonal and diurnal variation patterns as compared with those sites in the Himalayas and the northern Tibetan Plateau as presented earlier. Our measurements of surface ozone at Nam Co are essential baseline data of the inland Tibetan Plateau, more long-term measurements are needed to enable a better spatial coverage and a comprehensive understanding of regional surface ozone variations and underlying influence mechanisms".

Line 332: please quantify the spatial scale of this "long-range" contribution

**Response:** Long-range transport of air pollutants referred to the atmospheric transport of air pollutants within a moving air mass for a distance greater than 100 kilometers.

SUMMARY Line 343: "Nam Co represents a wide background region in the Tibetan Plateau". In my opinion this need more quantification efforts, since this sentence is too generic/qualitative.

**Response:** We removed this sentences from the revised manuscript.

Line 349: " Synthesis comparison. . .". The authors did not convince me about the small impact of STE.

**Response:** We have tried our best to give more quantized evidence and modify our description in manuscript now.

ACKWNOLEDGMENTS You must acknowledge NOAA for providing HYSPLIT model and GFS meteorological files. I suppose that also the TrajPlot developers must be acknowledged

**Response:** Thanks for your suggestion. Added as you suggested. Acknowledgements were rewritten as follows (lines 438 - 442):

[revised manuscript text omitted]

---

## Referee Report (RR1)

**Review of the paper "Surface ozone at Nam Co in the inland Tibetan Plateau: variation, synthesis comparison and regional**

representativeness" by Yin et al.

This is my second review of the paper. I would like to congrats the authors since the manuscript is really improved since its first submission. Most of my suggestions have been accepted by the authors and most of my concerns cleared. A few minor points must be considered at this point, thus I support publication on ACP after there are considered/fixed.

Abstract, line 19: please, substitute "long-term" with "continuous"

Abstract, line 23: here the low anthropogenic contribution from China in summer and from South Asia in spring must be cited.

Pag 7, line 182: 47.6 +/- 11.6 ppb. Is it +/- 1-sigma? Please, specify.

It is noteworthy that July 2015 appeared much more higher that other July months. Maybe an interesting anomaly to investigate in a future paper!

Pag 9, line 233: "...because stratospheric intrusion contribution is characterised by a maximum in spring,..."

Figure S4 is rather obscure to me. I cannot be able to understand which is the message that the authors aim to provide by this figure and what information is actually provided. What these histograms and scatterplots represent? Please, better explain in the text and in the figure caption. Moreover, even in the SM, I would like to see some explanation about this block-bootstrapped method. I'm not used with this method (and I guess other potential readers) and a simple explanation can help in better evaluating the obtained results without searching in other papers!

Line 256: to consider the PV at 350 hPa has much more sense that considering PV at the surface, actually!

Line 273: actually these structures over Himalayas are similar to those reported by Bracci et al. (JAMC, 2012). However, looking at figure 5, the 1-2 PV isolines do not appeared much more different in spring than in winter, please check it!

Line 283: in my opinion, by this analysis hardly you can affirm that STE "occurred right above Nam Co". Please remove.

Line 285-290: It is not clear which is the "take home message" of the SWD analysis. It is the local photochemistry which is supposed to play the most important role or the hemispheric-scale contribution? Please, clarify.

Line 294: please correct "Der".

Line 295: "the height of the folding is higher than those in the spring and summer". I do not think that this info can be retrieved by Fig. 5. Maybe some works can be cite to support (e.g. Ojha, N., Pozzer, A., Akritidis,

D., and Lelieveld, J.: Secondary ozone peaks in the troposphere over the Himalayas, Atmos. Chem. Phys., 17, 6743-6757, https://doi.org/10.5194/acp-17-6743-2017, 2017.)

Line 312: and high PBLH, thus suggesting an important role of thermal transport regimes of PBL air-masses to diurnal ozone variability". Once again: it is not completely clear which is (for the authors) the driving processes for diurnal variability: (local?) photochemistry or transport?

Line 346: "...on the urban scale, due to NO titration under ambient conditions not favourable to photochemical production."

Section 5.3: I must say that I'm still convinced that your clustering is not so effective. Looks for instance to the winter season: the single trajectories spanned over a wide latitudinal range, but the cluster centroids are not able to catch this variability. This must be stressed in the discussion...

Line 378: please modify as following: "...may also reflect this possibility."

Figure 11: it is almost impossible to read the inserts. I would suggest to rearrange figure 11 (see my attachment)

Line 390: probably PSCF also picked up the contribution of the transport of pollution from the Indo-gangetic plains and Himalaya foothills...

Line 395 – 397: Actually, you showed that Mt. Waliguan, NCOP and other stations have very different seasonal and diurnal cycles. So I do not agree that they are representative of the entire TP! Please, check!

Line 425: I think that the term "aged" is confusing in this context. As shown by other works (Putero et al., 2016; Ojha et al., 2017...), STE affecting Himalayas can occur very far from the region (e.g. Mediterranean basin). So I wouldn't call STE occurring over TP as "aged"...

Line 428: ..."is similar" (please remove "most")

Line 429: please quantify "infrequent": which is the percentage of occurrence?

---

## Author Response (AR2)

**Response to referee comments**

We would like to thank the referees and editor again for the helpful comments and suggestions to improve our manuscript. We have carefully considered all comments and the replies are listed below. The changes have been marked in the text using blue color.

**Anonymous Referee #2**

This is my second review of the paper. I would like to congrats to the authors because the manuscript is really improved since its first submission. Most of my suggestions have been catched by the authors and most of my concerns cleared. A few minor points must be considered at this point, thus I support publication on ACP after there are considered/fixed.

Response: Thanks for your valuable advices and comments.

Abstract, line 19: please, substitute "long-term" with "continuous" **Response:** Thanks for your suggestion. Changed as suggested in line 19.

Abstract, line 23: here the low anthropogenic contribution from China in summer and from South Asia in spring must be cited.

**Response:** The sentence was added as follows (lines 24 - 25):

"and the anthropogenic contribution from South Asia in spring and China in summer may affect Nam Co Station occasionally".

Pag 7, line 182: 47.6 +/- 11.6 ppb. Is it +/- 1-sigma? Please, specify.
Response: 47.6±11.6 is the mean ± standard deviation. Now we specified as follows (lines 220- 221): "(mean ± standard deviation)".

It is noteworthy that July 2015 appeared much more higher that other July months. Maybe an interesting anomaly to investigate in a future paper!

Response: Thanks for your suggestion. We will do further study considering your advice.

Pag 9, line 233: "...because stratospheric intrusion contribution is characterised by a maximum in spring,..."

**Response:** Thanks for your suggestion. Now we added the sentence as follows (lines 260 - 261): "because stratospheric intrusion contributions are seasonal,".

Figure S4 is rather obscure to me. I cannot be able to understand which is the message that the authors aim to provide by this figure and what information is actually provided. What these histograms and scatterplots represent? Please, better explain in the text and in the figure caption. Moreover, even in the SM, I would like to see some explanation about this block-bootstrapped method. I'm not used with this method (and I guess other potential readers) and a simple explanation can help in better evaluating the obtained results without searching in other papers!

Response: Thanks for your suggestion. Now we added the sentences as follows (lines 275 - 284):

"The uncertainties in the model regression results are shown graphically in Fig. S4. The histograms show variation in the contribution to ozone variance for each group based on the 100 realizations of the model. Taking the CAMx tracer as an example, the model suggests that this term contributes 17.7% of the O3 variance on average, but the results range from 12 to 24% and have a standard deviation of 2.6%. The scatterplots show the covariance between the model estimates for different groups. Most groups do not covary and hence there is no correlation between the x and y axis and the correlation coefficients are low. For example, the contribution from the WRF-FLEXPART clusters does not covary with the CAMx tracer, and  $r^2 = 0.03$ . In contrast, the seasonal signal covaries with the CAMx tracer ( $r^2 = 0.5$ ) which suggests that there is an increased uncertainty in the estimates for these terms and that changes in estimates for the seasonal signal will lead to changes in the estimates for the CAMx tracer. In this case, this is because stratospheric intrusions occur in spring and hence there is an increased uncertaint the term is an increased provide the seasonal signal covaries for the terms and the terms and the terms are stratospheric intrusions occur in spring and hence there is an increased uncertainty between the terms is an increased provide the stratospheric intrusions occur in spring and hence there is an increased uncertainty in the stratospheric intrusions between the two groups".

Line 256: to consider the PV at 350 hPa has much more sense that considering PV at the surface, actually! **Response:** Thanks for your comment. PV at 350 hPa was in positive correlation with surface ozone concentration at Nam Co Station.

Line 273: actually these structures over Himalayas are similar to those reported by Bracci et al. (JAMC, 2012). However, looking at figure 5, the 1-2 PV isolines do not appeared much more different in spring than in winter, please check it!

**Response:** PV isolines is one of the parameters in these monthly mean meridional cross-section. Zonal winds and ozone concentration appeared different in the spring and the winter, especially the longitude and the altitude of the hot spots of ozone which indicating that the surface ozone concentration at Nam Co Station is more easily affected by the air mass from stratosphere in the spring than the winter.

Line 283: in my opinion, by this analysis hardly you can affirm that STE "occurred right above Nam Co". Please remove.

Response: Thanks for your comment. Now we removed this sentence.

Line 285-290: It is not clear which is the "take home message" of the SWD analysis. It is the local photochemistry which is supposed to play the most important role or the hemispheric-scale "photochemistry"? Please, clarify.

**Response:** Thanks for your comment. Local photochemistry is an important role to the monthly surface ozone variation at Nam Co Station and the intense solar radiation at Nam Co Station in the spring may lead to more photochemical production of surface ozone.

Line 294: please correct "Der". **Response:** Thanks for your comment. Changed as suggested in line 332.

Line 295: "the height of the folding is higher than those in the spring and summer". I do not think that this info can be retrieved by Fig. 5. Maybe some works can be cite to support (e.g. Ojha, N., Pozzer, A., Akritidis, D., and Lelieveld, J.: Secondary ozone peaks in the troposphere over the Himalayas, Atmos. Chem. Phys., 17, 6743-6757, https://doi.org/10.5194/acp-17-6743-2017, 2017.)

**Response:** Thanks for your comment. Changed as suggested as (lines 330 – 333):

"Ojha et al. (2017) found that the potential vorticity layer in the summer was weaker than during the late winter and in the spring in the central Himalayan region which is to the south of Nam Co Station. In the autumn (Sep, Oct and Nov) and the winter (Dec, Jan and Feb), the mixing heights at Nam Co Station were much lower than those in the spring and the summer".

Line 312: and high PBLH, thus suggesting an important role of thermal transport regimes of PBL airmasses to diurnal ozone variability". Once again: it is not completely clear which is (for the authors) the driving processes for diurnal variability: (local?) photochemistry or transport?

**Response:** Thank you for pointing this out. Both vertical mixing and photochemical production are important to the diurnal variation of surface ozone at Nam Co Station.

Line 346: "...on the urban scale, due to NO titration under ambient conditions not favourable to photochemical production."

Response: Thank you for pointing this out. Now we rewrote this sentence as (lines 378 - 380):

"It is possible that titration of ozone by NOx at the urban sites as well as the differences in altitude and meteorology may lead to differences between the surface ozone concentrations at Nam Co Station and those at Lhasa and Dangxiong".

Section 5.3: I must say that I'm still convinced that your clustering is not so effective. Looks for instance to the winter season: the single trajectories spanned over a wide latitudinal range, but the cluster centroids are not able to catch this variability. This must be stressed in the discussion...

**Response:** Thanks for your comment. We calculated the backward trajectories every 3 hours during the measurement period. Comparing the limited trajectories in the periphery, there were much more trajectories in the main body of trajectories which were grouped into clusters.

Line 378: please modify as following: "...may also reflect this possibility."

**Response:** Thank you for pointing this out. Now we modified this sentence as (lines 411 - 412):

"The higher level of surface ozone at NCO-P (Cristofanelli et al., 2010) than at Nam Co Station in the spring may also reflect this possibility".

Figure 11: it is almost impossible to read the inserts. I would suggest to rearrange figure 11 (see my attachment)

Response: Thanks for your comment. Now we rearranged figure 11 as you suggested.

Line 390: probably PSCF also picked up the contribution of the transport of pollution from the Indogangetic plains and Himalaya foothills...

**Response:** Thank you for pointing this out. Now we modified this sentence as (lines 424 - 425):

"and the transport of pollution from the Indo-Gangetic Plain and Himalayan foothills was also probably picked up by PSCF".

Line 395 – 397: Actually, you showed that Mt. Waliguan, NCOP and other stations have very different seasonal and diurnal cycles. So I do not agree that they are representative of the entire TP! Please, check!

**Response: Thanks for your comment. Now we removed "entire" from this sentence.**

Line 425: I think that the term "aged" is confusing in this context. As shown by other works (Putero et al., 2016; Ojha et al., 2017...), STE affecting Himalayas can occur very far from the region (e.g. Mediterranean basin). So I wouldn't call STE occurring over TP as "aged"... **Response:** Thanks for your comment. Now we removed "aged" from this sentence.

Line 428: ..."is similar" (please remove "most") **Response:** Thanks for your comment. Now we removed "most" from this sentence.

Line 429: please quantify "infrequent": which is the percentage of occurrence?

**Response:** Thanks for your comment. Cluster 5 in the summer represents the backward trajectories from or passed through the Northwest China and it occurs only 3.4% during whole the summer. Cluster 3 in the summer also represents the trajectories from parts of the Northwest China and it occurs 10.81% during whole the summer.

**Co-Editor Decision: Reconsider after minor revisions (Editor review)**

Comments to the Author:

Dear authors,

Thank you for submitting your revised manuscript, which has been reviewed by two referees. After the revision the manuscript reads much better. However, further improvement is necessary to meet the ACP quality requirements. Please take into account referee #2's comments in your next revision. In addition I also have many points for your consideration. Please submit your revised manuscript with your point-by-point responses to referee and editor comments.

Sincerely,

Xiaobin Xu

**Response:** Thank you so much for your comment.

Co-editor comments:

(1) the English language needs to be polished further. There are many grammatical errors (e.g., incorrect use of "the", plural and singular, etc.). I strongly suggest that you either have the manuscript edited a native English-speaking expert or select the English language copy-editing.

**Response:** Thanks for your suggestion. Now we polished the manuscript by native English-speaking expert.

(2) Line 25, I think you mean "Model results indicate that the study site is affected 'differently' by the surrounding areas in different seasons."

**Response:** Thank you for pointing this out. Now we rewrote this sentence as (lines 26 - 27):

"Model results indicate that the study site is affected differently by the surrounding areas in different seasons".

(3) Line 38, change "globally" to "global".

Response: Changed as you suggested in line 40.

(4) Lines 44-45, "... photochemistry was identified as the dominant source of tropospheric ozone at some sites,". I think photochemistry is a dominant source of tropospheric ozone over all sites of the world, not only "some sites".

Response: Thank you for pointing this out. Now we removed "at some sites" from this sentence.

(5) Line 46, what do you mean by "largely uniform synoptic systems"? A synoptic system or a large scale weather system has its structure characterized by many parameters, pressure, winds, temperature, etc. The spatial distributions of these parameters are not necessarily uniform. I think it is better to change "largely uniform synoptic systems" to " synoptic systems".

Response: Changed as you suggested. Now we removed "largely uniform" from this sentence.

(6) Lines 61-63, "The paucity of long-term surface ozone observations in the Tibetan Plateau, ... and can potentially lead to inaccurate simulation of surface ozone variation over the Tibetan Plateau". The accuracy of surface ozone simulation is determined by the capability of the model, the accuracy and fineness of emission inventory, the quality of meteorological data, etc., and should not be directionally influenced by the paucity of long-term surface ozone observations.

Response: Thank you for pointing this out. Now we modified this sentence as (lines 64 - 65):

"The paucity of long-term surface ozone observations in the Tibetan Plateau, especially in the inland region, limits our understanding of the regional background ozone level and the factors that influence it in the Tibetan Plateau".

(7) Lines 70-71, I do not think this sentence should be placed here.**Response:** Thanks for your comment. Now we removed this sentence.

(8) Lines 74-75, it is better to move this sentence to line 49 (before "Due to...").Response: Thanks for your comment. Changed as you suggested (lines 50 - 51).

(9) Lines 78-80, check the language and logic of this sentence.

**Response:** Thank you for pointing this out. Now we modified this sentence as (lines 78 - 79):

"Nam Co Station was established in September 2005 to monitor atmospheric conditions and enabled research of the atmospheric environment in the inland Tibetan Plateau (Kang et al., 2011)".

(10) Line 90, change "tourist" to "tourism".**Response:** Changed as you suggested in line 90.

(11) Line 104, was the radiation measurement system calibrated? Was it stable?Response: The radiation measurement system was calibrated and it was relatively stable.

(12) Line 113, are you sure there was no obvious difference among the backward trajectories for 500 m, 1000 m and 1500 m? This is usually not the case. Please check again.

**Response:** We calculated the backward trajectories for these three heights and it seems that the trajectories for different arrival heights had similar features and the pathways were similar. The trajectories calculated at 1500 m height traveled longer distances than those in 500 and 1000 m, but in the same direction.

(13) Lines 139-140, what do you mean by "offset by 10 ppb"? Add 10 ppb to all hourly ozone value? Why can a normal distribution be obtained by doing so?

**Response:** Thanks for your comment. We have added an explanation as follows (lines 163 - 164):

"We used the natural logarithm of  $O_3$  concentrations offset by 10 ppb to approximate a normal distribution while reducing the long tail in the transformed variable that would be caused by taking the logarithm of low  $O_3$  concentrations".

(14) Line 194, some of the data gaps in Figs. 4 and S5 seem to be over a month. Are you sure you have >56% of data for EACH month?

Response: Thanks for your comment. Each month reported in this study had valid data more than 56%.

(15) What do you want to say with "the overall trends" here?

**Response:** Thank you for pointing this out. Now we modified this sentence as (lines 233 - 235):

"The surface ozone concentrations at Nam Co Station experienced similar annual cycles during each of the 5 years of measurements with slight variations (Fig. S1)".

(16) Lines 212-214, Fig. 3 shows the daily minima in spring, summer and autumn are all at 8:00, contradicting to what you state here.

Response: Thank you for pointing this out. Now we modified this sentence as (lines 251 - 253):

"Mixing ratios went from low levels at night to high levels during the daytime. The diurnal profile was generally characterized by a later shift from low to high concentrations in the winter than in the rest of the year, most likely as a result of the later time of sunrise".

(17) Section 4, please mathematically show your regression model. Without this it is very difficult to understand your method and results. (18) Lines 223-224, what do you actually mean with "For the boundary layer height quintiles were used"? (19) Lines 239-240, the WRF-FLEXPART produced six transport clusters. Why did you only obtain one contribution?

**Response:** Thanks for your comments. For these three comments, now we added the description of Multiple Linear Regression as (lines 136 - 192):

[revised manuscript text omitted]

**Response:** Thanks for your comment. The sentence was based on an out-of-date figure, we apologize for this error. Now we rewrote this sentence (lines 266 - 267) and we have also better described the scaling factor in section 2.4 thanks to your comments:

"The regression model suggests that both stratospheric transport and seasonal variation can lead to enhancements in hourly concentrations of up to 20% of the baseline level of ozone".

(21) Lines 301-305, "This is probably due to the time delay between the maximum ozone concentration and the maximum solar radiation" (lines 302-303). What do you mean with this sentence? The fact is that there is a delay between both. "Hourly average SWD showed a positive correlation with hourly average surface ozone (correlation coefficient=0.77) which indicated that the potential of local ozone formation by photochemical production during the daytime contributed to the peak in the afternoon (Wang et al., 2006)". It is too speculative and contradicting your points that are following. Local ozone formation may result in a good SWD-ozone correlation. However, a good SWD-ozone correlation may not necessarily be caused by local ozone formation. Wang et al. (2006) reported in situ photochemical production of ozone under the Waliguan spring and summer conditions that were very different from the Nam Co conditions. So, this citation does not support your speculation. I am not saying that there is no local formation of ozone at Nam Co. Theoretically local formation exists always where solar radiation is received. And ozone is also removed by local chemical and physical processes. If there is not net ozone production, you cannot attribute your SWD-ozone correlation to local ozone formation. Even if there is a net ozone production, you cannot simply do so with detailed analysis because the SWD-ozone correlation may be caused by some other mechanisms. In fact, you are saying that the wind speed and PBLH are the main factors (lines 305-306). I strongly suggest that you delete this speculation and keep consistency.

**Response:** Thank you for pointing this out. Changed as you suggested. Now we removed the discussion of SWD-ozone correlation and modified the sentence as (lines 344 - 345):

"In addition, local photochemical production may also contribute to the higher concentration of surface ozone at Nam Co Station in the daytime".

(22) Lines 345-346, "It is possible that the local NOx emissions in these two urban regions reduce the average ozone on the urban scale". Again, this is groundless. NOx is the key precursor of ozone and promotes the photochemical production of ozone under favorable meteorological conditions. The meteorological conditions in Lhasa favor the ozone production as reported in Ran et al. (2014). Ozone at Dangxiong is controlled by large-scale physic-chemical processes, particularly the vertical mixing (Lin et al., 2015). Therefore, the differences between ozone values at Nam Co and those at Lhasa and Dangxiong should not be due to differences in NOx emissions and may be caused some other factors, such differences in altitude, meteorology, etc.

Response: Thanks for your comment. Now we rewrote this sentence as (lines 378 - 380):

"It is possible that titration of ozone by NOx at the urban sites as well as the differences in altitude and meteorology may lead to differences between the surface ozone concentrations at Nam Co Station and those at Lhasa and Dangxiong".

(23) Lines 388-389, "In the winter, no obvious region was identified, which was likely due to low surface ozone mixing ratios in all these areas." Not clear.

**Response:** Thank you for pointing this out. Now we modified this sentence as (lines 422 - 423): "In the winter, no obvious region was identified as a potential source region".

(24) Line 393, "universal concern"?! Please do not use inflated language.

Response: Thank you for pointing this out. Now we changed "universal" to "great" in line 427.

(25) Line 394, "varied" or "varying"?

Response: Thank you for pointing this out. Now we changed "varied" to "varying" in line 428.

(26) Line 404-406, "Its mountainous landform facilitates mountain-valley breezes and may sometimes pump up anthropogenic emissions especially during the winter (Xue et al., 2011)". This is wrong! The impacts of anthropogenic emissions on the Waliguan station occur mainly in summer. And the Xue et al. (2011) paper is about a summer study.

**Response:** Thanks for your comment. Now we rewrote this sentence as (lines 438 - 439): "and the impacts of anthropogenic emissions on Waliguan occurr mainly in the summer".

(27) Lines 423-424, "The analysis suggests that stratospheric intrusions account for around 20% of the variability in surface ozone concentrations at Nam Co Station" is a misleading statement. Fig. 4 shows that stratospheric ozone makes at maximum a contribution of 20% to the variation of surface ozone at

Nam Co. Most of the time, the stratospheric contribution is either zero or very low. I think you should state "...that the maximum contribution of stratospheric intrusions to surface ozone at Nam Co Station is about 20%" or give an average contribution.

**Response:** Thanks for your comment. Changed as you suggested. Now we rewrote this sentence as (lines 457 - 458):

"The analysis suggests that the maximum contribution of stratospheric intrusions to variability in surface ozone at Nam Co Station is approximately 20%".

(28) Please rewrite the third paragraph of Section 6 (Summary). This paragraph should be based on the major results in Section 5. I do not think this is the case in the current version.

**Response:** Thanks for your comment. Now we rewrote this paragraph as (lines 461- 464):

"Surface ozone at Nam Co Station showed distinct seasonal and diurnal variation patterns as compared with other sites in the Himalayas and the northern Tibetan Plateau. The monthly maximum of surface ozone at Nam Co Station, which is in the inland Tibetan Plateau was later in the year than at the sites in the southern Tibetan Plateau and the southern ridge of the Himalayas, but earlier in the year than at the sites in the northern Tibetan Plateau".

(29) Line 466, "Our measurements provide a baseline of tropospheric ozone at a remote site in the Tibetan Plateau". Be careful, your measurements of surface ozone at any site cannot provide a baseline (or whatever) of tropospheric ozone, which should be described by tropospheric column content and vertical distribution of ozone globally or over a region.

**Response:** Thanks for your comment. Changed as you suggested. Now we rewrote this sentence as (lines 465 - 466):

"Our measurements contribute to the understanding of ozone cycles and related physico-chemical and transport processes over the Tibetan Plateau".

(30) Fig. 4, the "scaling factor" is not explained and not clear.

Response: Thanks for your comment. Now we added sentences as (lines 264 - 267):

"In the log-transformed model, the time series for each group corresponds to a scaling factor that is applied to the baseline ozone concentration. This is shown in Fig. 4 for the stratospheric tracer and for the seasonal scaling. The regression model suggests that both stratospheric transport and seasonal variation can lead to enhancements in hourly concentrations of up to 20% of the baseline level of ozone".

(31) Fig. 5, what are the red dots on the sub-plots? The location of Nam Co? If so, there are relative positions should not be varying.

**Response:** The red dots show the position of the top of boundary layer height at Nam Co Station. We also clarify it in the caption of Fig. 5.

(32) Fig. 10, the color selection of this figure is poor. Some of the trajectory clusters are hardly differentiated. I suggest that you use light grey for individual trajectories and well contrasted colors for the clusters.

Response: Thanks for your comment. Changed as you suggested.

(33) Fig. 12, the PSFC data look very strange. Why are the differences among season so large? See for example the difference between spring and winter? How did you obtain the weighted values? Were there calculated on a yearly or seasonal basis?

**Response:** We used the mean concentration of surface ozone as the threshold in PSCF calculation. Due to low surface ozone concentration in winter, less data were used in PSCF calculation and grid cells in the winter had less air parcels passed through. Then the PSCF values multiplied smaller weighting value and resulted in the difference of weighted PSCF among different seasons. The weighted PSCF values were obtained by multiplying the original PSCF values by the weighting factor. There were calculated seasonally but the weighting function was unified.

(34) Tables 2 and S1, it is not so straightforward what you are presenting in the tables. Please study the ACP instructions for the requirements on tables.

Response: Thanks for your comment. Changed as you suggested.

**Surface ozone at Nam Co in the inland Tibetan Plateau: variation, synthesis comparison and regional representativeness**

Xiufeng Yin1, 2, 3, Shichang Kang1, 3, 4, Benjamin de Foy5, Zhiyuan Cong2, 4, Jiali Luo6, Lang Zhang2, Yaoming Ma2, 3, 4, Guoshuai Zhang2, Dipesh Rupakheti2, 3, Qianggong Zhang2, 4

[revised manuscript text omitted]